# Decoding necrosome assembly: harmonizing signal amplification and attenuation through optimal RIP3 stoichiometry

Xiang Li [1,10] ✉, Yating Cao[2,10], Fei Xu[3,10], Yiting Zhang[2], Yue Kong[2], Chengjie Lan[2], Rongfeng Zhu[4], Cheng Lin[1], Chuan-Qi Zhong [2], Zhilong Liu[1], Hong Qi [5], Yichuan Huang[1], Yunshan Xiao[6], Gui-Quan Sun[5,7], Jianwei Shuai[8] ✉ & Xin Chen [2,9] ✉

Necrosome assembly is essential for necroptosis, a process implicated in neurodegeneration, ischemic injury, and inflammatory diseases. Yet the spatiotemporal rules governing this assembly remain elusive. Leveraging quantitative STORM and mathematical modeling, we define an approximately 3:1 ratio of RIP3 to RIP1 in necrosomes as the optimal stoichiometry for necroptosis, enabling signal amplification and a threshold response. Surprisingly, excessive RIP3 oligomerization attenuates signaling, acting as an intrinsic size control mechanism. RIP3 assembly is dynamically regulated: it is constrained by stimulation and RIP1, promoted by RIP3 itself, and unexpectedly limited by downstream MLKL. A complementary balance between necrosome quantity and RIP3 assembly degree ensures efficient MLKL phosphorylation. In contrast, Caspase-8 assembly is limited by c-FLIP and recruited linearly by RIP1, while its distinct behavior from RIP3 underlies the biphasic necroptotic response to RIP1. These findings uncover the flexible, multi-strategic nature of signalosomes and offer valuable insights for therapeutic and synthetic biology.

With the growing recognition of structural diversity of signalosomes, higher-order assemblies have emerged across a wide spectrum of biological landscapes, generally involving the polymerization of receptors, adaptors, and effectors into large structures with defined, rigid shapes or dynamic, liquid-like condensates[1–3]. These supramolecular complexes enhance signal transduction, amplify cellular responses, and facilitate enzyme activation, while maintaining a dynamic and tightly regulated structure[4–6]. Among them, amyloid-like assemblies are a well-known subclass, essential for the assembly of RIP1-RIP3 in necrosomes[7].

Necrosome assembly, the central signaling event driving necroptosis, has been primarily studied in the context of TNF signaling[8]. Upon TNF treatment and inhibition of apoptosis, a cascade of events triggers the assembly of necrosomes (Fig. 1a). TNF receptor 1 (TNFR1) recruits TNF receptor-associated death domain (TRADD) and RIP1, which subsequently dissociates from the membrane-bound complex and interacts with RIP3. Our prior research indicates that TRADD does not participate in necrosomes formation[9], while MLKL is additionally recruited into necrosomes. The RIP homology-interacting motif (RHIM) domains of

[1]Department of Physics, Fujian Provincial Key Lab for Soft Functional Materials Research, Xiamen University, Xiamen, China. [2]State Key Laboratory of Cellular Stress Biology, School of Life Sciences, Faculty of Medicine and Life Sciences, Xiamen University, Xiamen, China. [3]Department of Physics, Anhui Normal University, Wuhu, China. [4]Institute of Chemical Biology, Shenzhen Bay Laboratory, Shenzhen, China. [5]Complex Systems Research Center, Shanxi University, Taiyuan, China. [6]Department of Obstetrics, School of Medicine, Xiamen University, Xiamen, China. [7]School of Mathematics, North University of China, Taiyuan, Shanxi, China. [8]Oujiang Laboratory (Zhejiang Lab for Regenerative Medicine, Vision and Brain Health), Wenzhou Institute, University of Chinese Academy of Sciences, Wenzhou, China. [9]Xiamen Cardiovascular Hospital of Xiamen University, School of Medicine, Fujian Branch of National Clinical Research Center for Cardiovascular Diseases, Xiamen, China. [10]These authors contributed equally: Xiang Li, Yating Cao, Fei Xu. ✉e-mail: xianglibp@xmu.edu.cn; shuaijw@wiucas.ac.cn; xchen@xmu.edu.cn

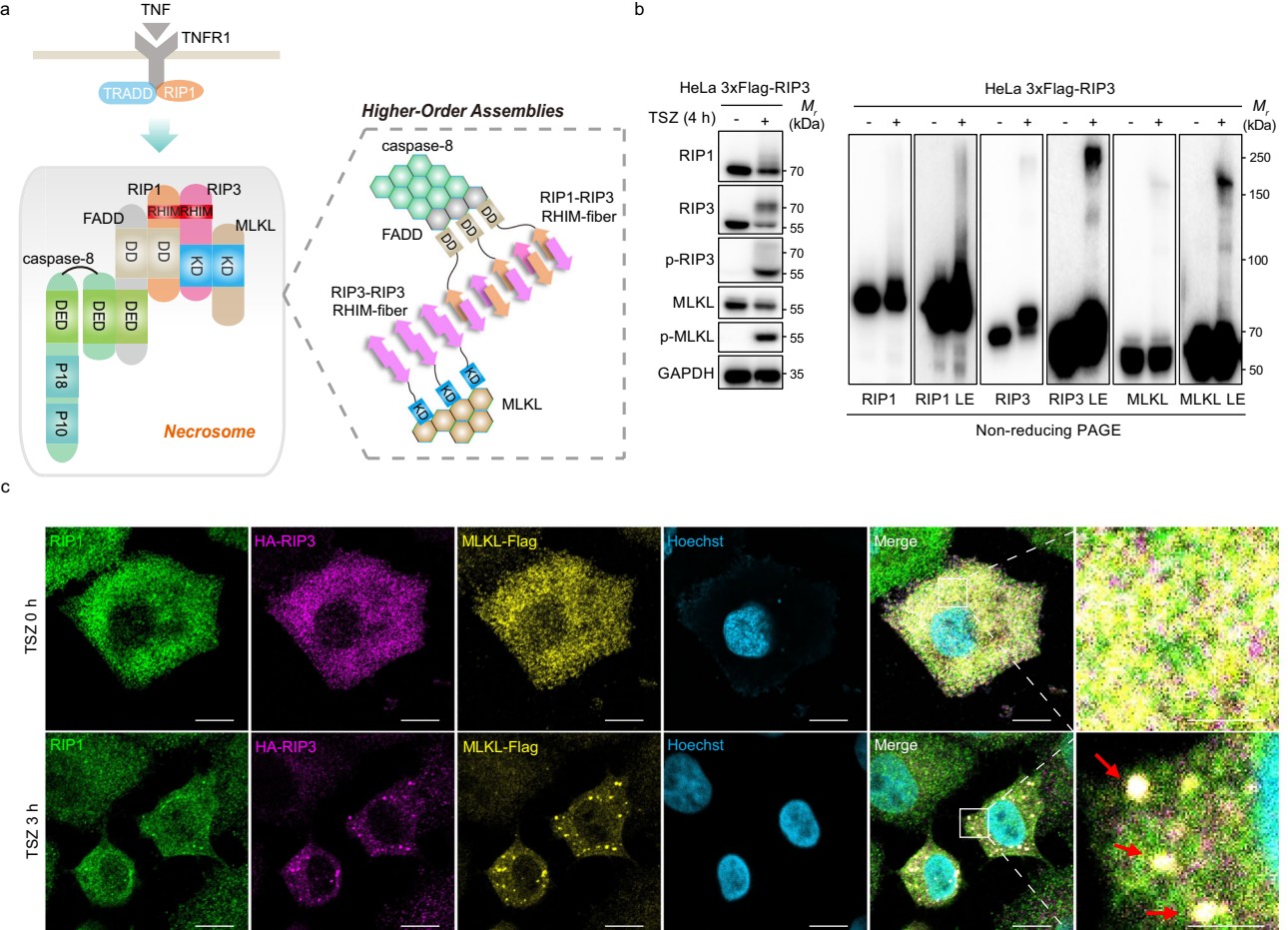

**Fig. 1 | Necrosomes are higher-order assemblies comprising RIP1, RIP3, and MLKL. a** Schematic of TNF-induced supramolecular signaling leading to necrosome formation. **b** Immunoblot of the indicated proteins in lysates from RIP3-expressing HeLa cells, untreated or treated with TSZ (TNF + Smac mimetic + zVAD-fmk) for 4 h. LE, long exposure. **c** Confocal images of *MLKL*-knockout (KO) HeLa cells reconstituted with HA-RIP3 and MLKL-Flag and stimulated with DMSO or TSZ for 3 h. Fixed cells were immunolabeled with antibodies against RIP1 (green), HA (magenta), and Flag (yellow). DNA was counterstained with Hoechst 33342 (blue). Enlarged regions (white boxes) depict cytosolic necrosomes (red arrows), demonstrating co-localization of all three proteins. Data in (**b**, **c**) are representative of two and three independent experiments, respectively. Scale bars: 10 μm (overviews) and 2 μm (insets). Source data are provided as a Source Data file.

RIP1 and RIP3 form filamentous structures reminiscent of β-amyloids. The structural organization is intricate, with both RIP1 and RIP3 adopting a meandering β-strand architecture, comprising four short β-segments separated by turns, which collectively form the serpentine fold. The heteroamyloid complex formed by RIP1–RIP3 serves as a nucleation site for RIP3 recruitment and polymerization via the same β-amyloid interaction[10]. Meanwhile, RIP1 utilizes its death domains (DD) to recruit FAS-associated death domain (FADD), which further interacts with caspase-8 via the death effector domain (DED). DED-filaments serve as scaffolds for caspase-8 clustering onto FADD, resulting in the formation of a supramolecular complex[11,12]. Upon activation, RIP3 phosphorylates its substrate, mixed-lineage kinase domain-like protein (MLKL). Our recent study suggests that RIP3 oligomers, particularly those sized tetramer or above[13], facilitate MLKL polymerization within necrosomes[14], driving membrane permeabilization and subsequent necroptotic cell death[15–19].

Necrosome is a supramolecular signaling complex that serves as the central execution module of necroptosis. Necrosome assembly not only ensures the elimination of infected or damaged cells but also promotes the release of damage-associated molecular patterns (DAMPs), thus amplifying inflammatory responses[20]. Importantly, aberrant necrosome assembly or dysregulation of its components has been implicated in a wide spectrum of pathological conditions, including ischemia-reperfusion injury, neurodegenerative diseases, inflammatory bowel disease, and certain types of cancer[21]. As such, dissecting the principles of necrosome assembly is critical for understanding the molecular logic of necroptosis and for identifying therapeutic targets in inflammation-associated diseases. Yet, decoding the spatial organization and functional logic of necrosomes remains challenging.

Traditional approaches such as immunoprecipitation-mass spectrometry (IP-MS) provide ensemble-level stoichiometry but average out nanoscale heterogeneity. Single-molecule localization microscopy (SMLM), including stochastic optical reconstruction microscopy (STORM), enables nanoscale visualization of signalosomes, but technical uncertainties such as dye photo-physics and antibody variability hinder absolute molecule counting[22]. Therefore, our understanding of necrosome assembly is far from complete: Why do RIP1 and RIP3 form such specific configurations in necrosomes? How do cells terminate signaling from these higher-order assemblies? What controls the size of these higher-order complexes, and are there specific or universal mechanisms underlying this process?

In this study, we applied an integrative strategy combining super-resolution STORM imaging, mass spectrometry, and mathematical modeling. By introducing an internal MAP7-based calibration system and an SMAP–POCA analysis pipeline, we overcame technical uncertainties in STORM quantification. We discovered that necrosomes adopt an optimal stoichiometry, with approximately three RIP3 molecules per RIP1, balancing signal amplification and attenuation.

Notably, larger RIP3 assemblies suppress necroptotic signaling under MLKL-mediated negative feedback, extending current understanding of higher-order complexes as purely amplifying platforms. Caspase-8 assembly is limited by c-FLIP and recruited linearly by RIP1, while its distinct behavior from RIP3 underlies the biphasic necroptotic response to RIP1. This flexible, multi-strategic assembly of necrosomes offers valuable insight into the regulatory logic of cell death signaling and therapeutic modulation.

## Results

### Higher-order assemblies of necrosomes are a hallmark of necroptotic cells

Higher-order structures are prevalent in signal transduction and often regulate signaling fidelity and amplification through polymerization of signaling components[23]. Prior studies primarily examined the effects of higher-order assembly of individual proteins within complexes on signal transmission, while the cooperative assembly of multiple distinct proteins into supramolecular complexes remains less well characterized[24–26]. Necrosomes offer a representative model system for investigating how higher-order assemblies involving two or more distinct proteins within a complex influence signal transduction. To confirm and analyze the structure of necrosomes, we reconstituted *MLKL*-knockout (KO) HeLa cells with MLKL-Flag and also ectopically expressed HA-RIP3 in this cell line. Cells underwent necroptosis upon treatment with TNF, Smac mimetic, and zVAD (TSZ), as indicated by robust phosphorylation of RIP3 and MLKL (Fig. 1b, left panel). Notably, RIP1, RIP3, and MLKL formed SDS-resistant, high-molecular-weight polymers that accumulated at the top of the non-reducing gel upon long exposure, consistent with their incorporation into supramolecular structures (Fig. 1b, right panel). These polymeric assemblies were also observed in various cell lines, including human HT-29 and murine L929[13], indicating a conserved necrosomes polymerization process across species. To visualize necrosome formation, we performed confocal microscopy in HeLa cells treated with DMSO or TSZ. In TSZ-treated cells, RIP1, RIP3, and MLKL formed distinct punctate structures, in contrast to their diffuse localization in control cells (Fig. 1c). These puncta frequently co-localized, as highlighted in the boxed regions, illustrating that higher-order assembly serves as a general and crucial step for necroptosis induction.

### Necrosome stoichiometry characterizes the nanoscale assemblies of RIP1, RIP3, MLKL, and caspase-8

To map the nanoscale architecture of necrosome, we employed super-resolution microscopy (STORM) with a lateral localization precision of ~12.93 nm, calculated by the LAMA algorithm[27]. As shown in the workflow (Fig. 2a), we performed single-color STORM imaging of RIP1, RIP3, and MLKL in necroptotic HeLa cells reconstituted with MLKL-Flag and HA-RIP3. To quantitatively assess these supramolecular structures, we implemented a robust SMAP + POCA analysis pipeline (Supplementary Fig. 1, see the Methods section for details), incorporating Voronoi-based clustering and morphological filtering. In untreated cells, RIP1, RIP3, and MLKL were visualized as dispersed small puncta in cytosol (TSZ 0 h, Fig. 2b). After stimulation, RIP1, RIP3 and MLKL formed larger punctate structures in round or rod shapes at early time (TSZ 2 h, Fig. 2b). These puncta grew and exhibited rod-shaped structures during necroptosis after long-time TSZ treatment (TSZ 4 h, Fig. 2b). To analyze the rod-shaped structures of RIP1, RIP3 and MLKL, we quantified the area of STORM-identified puncta (Fig. 2c). For RIP1 rods, the predominant area was ~17 × 10³ nm² at early stages (TSZ 2 h), with maximum values up to ~39 × 10³ nm². At later time points (TSZ 4 h), the predominant area increased to ~20 × 10³ nm², with maxima up to ~51 × 10³ nm². RIP3 rods exhibited larger areas than RIP1, with a predominant value of ~42 × 10³ nm² and maxima of ~103 × 10³ nm² at 4 h. MLKL rods showed areas similar to RIP1 rods during TSZ treatment. These findings indicate that RIP3 contributes

more substantially than RIP1 to rod-shaped necrosome assembly, consistent with previous reports showing that additional RIP3 homo-polymerization on the RIP1–RIP3 hetero-amyloid platform is critical for necroptosis induction[13]. Notably, the temporal increase in the fraction of RIP1-, RIP3-, or MLKL-positive rod-shaped necrosomes reflects a structural maturation process, transitioning from early oligomeric puncta to stabilized, rod-like supramolecular complexes during necroptosis (Fig. 2d).

To accurately determine necrosome stoichiometry, we utilized immunoprecipitation (IP) coupled with quantitative mass spectrometry (MS). Heavy amino acid-labeled proteins were spiked into purified IP complexes (FLAG-RIP3 and FLAG-MLKL), which were subsequently subjected to SWATH-MS and DDA analyses, allowing quantitative identification of necrosome-associated components across multiple time points (Fig. 2e). A substantial quantity of necrosome-associated proteins was detected, with our primary focus on elucidating the stoichiometry of key signal transducers (Supplementary Fig. 2). Through immunoprecipitation of RIP3 complex, we demonstrated that the ratio of caspase-8 to RIP1 was approximately 1:1 at early time (2 h) but gradually decreased over time (Fig. 2f), implying the presence of more than one RIP1 per caspase-8 within necrosomes at late stages. Similarly, the MLKL to RIP1 ratio was initially around 1:1 but increased to approximately 2:1 in the late stage (4 h–8 h). In MLKL co-immunoprecipitation assays, the RIP3 to RIP1 ratio stabilizes at about 3:1 during the late stage. The consistent ratios of RIP3 to RIP1 and MLKL to RIP1 at late stage suggest that necrosomes adopt a stable assembly mode to sustain necroptotic signaling.

Based on these quantitative measurements, we deduced an in situ necrosome stoichiometry of RIP1, caspase-8, RIP3, and MLKL approximately 2:1:6:4 in late-stage necroptosis (Fig. 2g). This assembly ratio outlines an assembly subunit that reflects a mechanistically optimized stoichiometry of the necrosome: two RIP1 molecules together with two RIP3 molecules form a heterotypic RHIM 'seed' (PDB: 5V7Z)[10]; among six RIP3 molecules, four assemble into a RIP3 tetramer, the minimal execution unit of necroptosis (PDB: 7DAC)[28], which activates MLKL (PDB: 7MON) and thereby drives four MLKL monomers to form the functional MLKL tetramer[29]. Caspase-8 participates as a non-catalytic monomeric regulator (PDB: 8YD8)[30]. Together, these features are consistent with prior reports and represent a structurally and functionally coherent scheme for necrosome assembly.

### A three-to-one or higher RIP3:RIP1 stoichiometry shapes signal amplification and threshold response

The precise mechanism by which the identified necrosome stoichiometry (Fig. 2g) governs subsequent activation remains unclear. To address this, we developed a basic mathematical model comprising two constituents, A and B, representing upstream and downstream signals, respectively (Fig. 3a and Supplementary Table 1). Within the binding complex, B is recruited and activated by A, and both A and B can form various higher-order assemblies. m and n represent the numbers of A and B molecules in the binding complex. We first considered the scenario where A remains monomeric ($m = 1$), while B assembles into different higher-order structures (Fig. 3b, upper panel). Compared to the monomeric B ($n = 1$, black line), dimeric ($n = 2$, red line) or tetrameric ($n = 4$, blue line) B substantially amplifies output signal, as measured by the level of activated B. Additionally, the input-output relationship transitions from a linear pattern when B is monomeric to nonlinear patterns as B assembles into dimeric or tetrameric structures. These findings align with the prevailing view that higher-order assembly is a pivotal initial step in achieving signal amplification and threshold behavior[1,2]. Next, we discussed the inverse condition, where B remains monomeric ($n = 1$) while A forms higher-order structures. In this case, increasing the assembly order of A ($m = 2, 4$) considerably reduces output signal compared to monomeric A ($m = 1$)

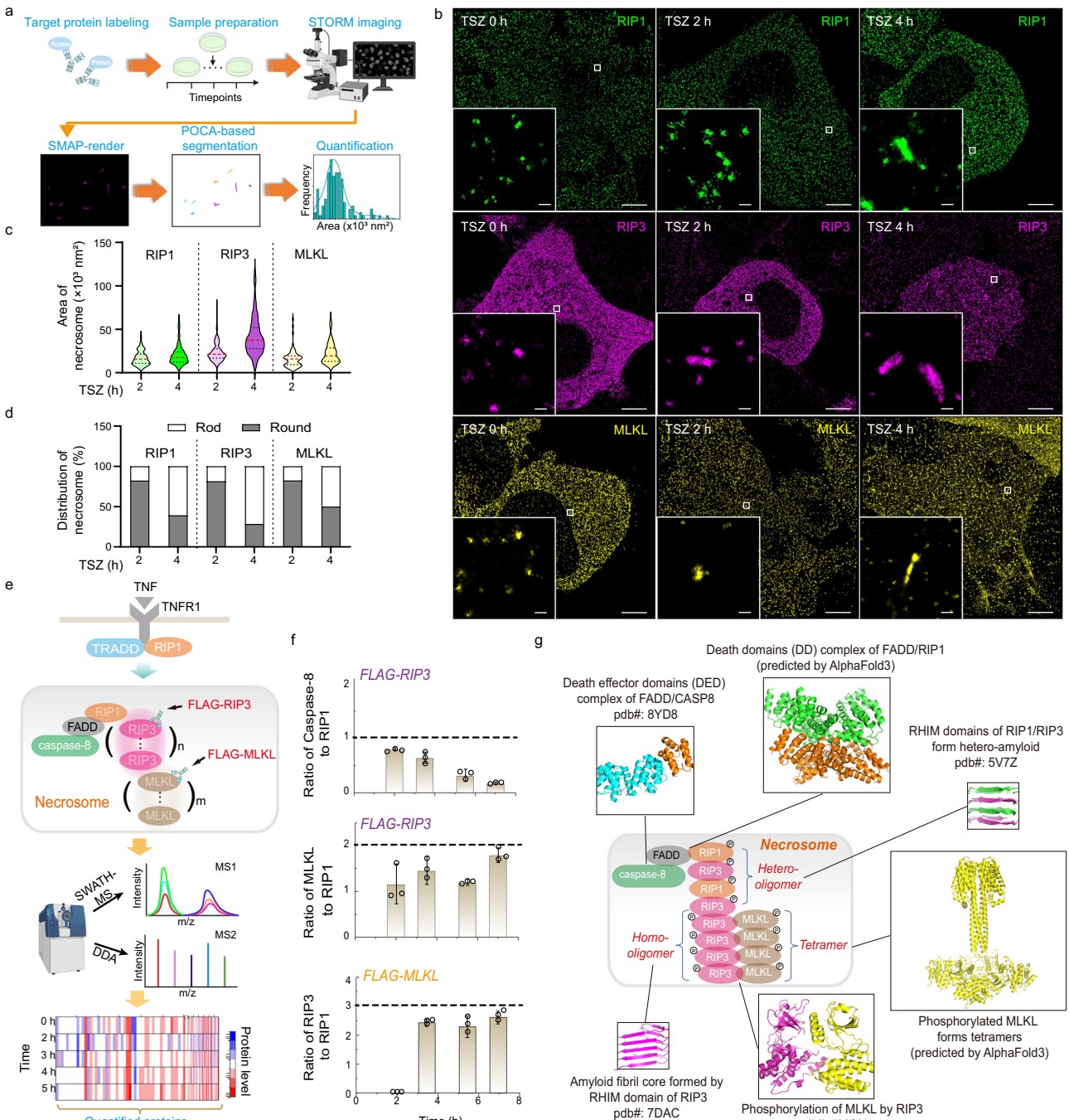

**Fig. 2 | Necrosomes form rod-shaped structures with varying molecular stoichiometry of key components. a** STORM workflow and image analysis pipeline. Detailed procedures for image acquisition and segmentation are described in the Methods. **b** Single-color STORM images of RIP1, RIP3, and MLKL in HA-RIP3-expressing HeLa cells treated with TSZ for the indicated durations. Insets show magnified views (white boxes) of RIP1, RIP3, or MLKL within necrosomes. Images are representative of two independent experiments per condition. **c** Area distributions of RIP1-, RIP3-, and MLKL-stained rods in HA-RIP3-expressing HeLa cells treated with TSZ for 2 h (*n* = 53 RIP1, 114 RIP3, 63 MLKL) or 4 h (*n* = 77 RIP1, 168 RIP3, 37 MLKL). **d** Proportions of round- versus rod-shaped RIP1, RIP3, and MLKL structures in Flag-RIP3–expressing HeLa cells treated with TSZ for 2 h or 4 h, classified based on STORM images. **e** Scheme of quantitative mass spectrometry (MS) workflow. RIP3 and MLKL complexes were immunoprecipitated with anti-Flag

agarose beads. IP samples were digested with trypsin, and peptides were analyzed using DDA and SWATH-MS. Details are described in the Methods section. Panels (**a**, **e**) were created with BioRender.com (https://BioRender.com/cril410). **f** Ratios of caspase-8, MLKL, and RIP3 to RIP1 in Flag-RIP3/Flag-MLKL-enriched necrosomes, as determined by MS. Data are presented as mean ± SD from three biological replicates per group. **g** Proposed model of the modular and cooperative architecture of the necrosome. Stoichiometric ratios (RIP1:RIP3:MLKL:Caspase-8 ≈ 2:6:4:1) were derived from IP-MS analysis. Published (PDB) and predicted (AlphaFold) structural models illustrate domain-specific interactions: DD oligomerization (TNFR, TRADD, RIP1), RHIM-driven hetero-/homo-amyloid assembly (RIP1–RIP3), and downstream effector recruitment (FADD–caspase-8; RIP3–MLKL). Scale bars: 5 μm (overviews) and 100 nm (insets). Source data are provided as a Source Data file.

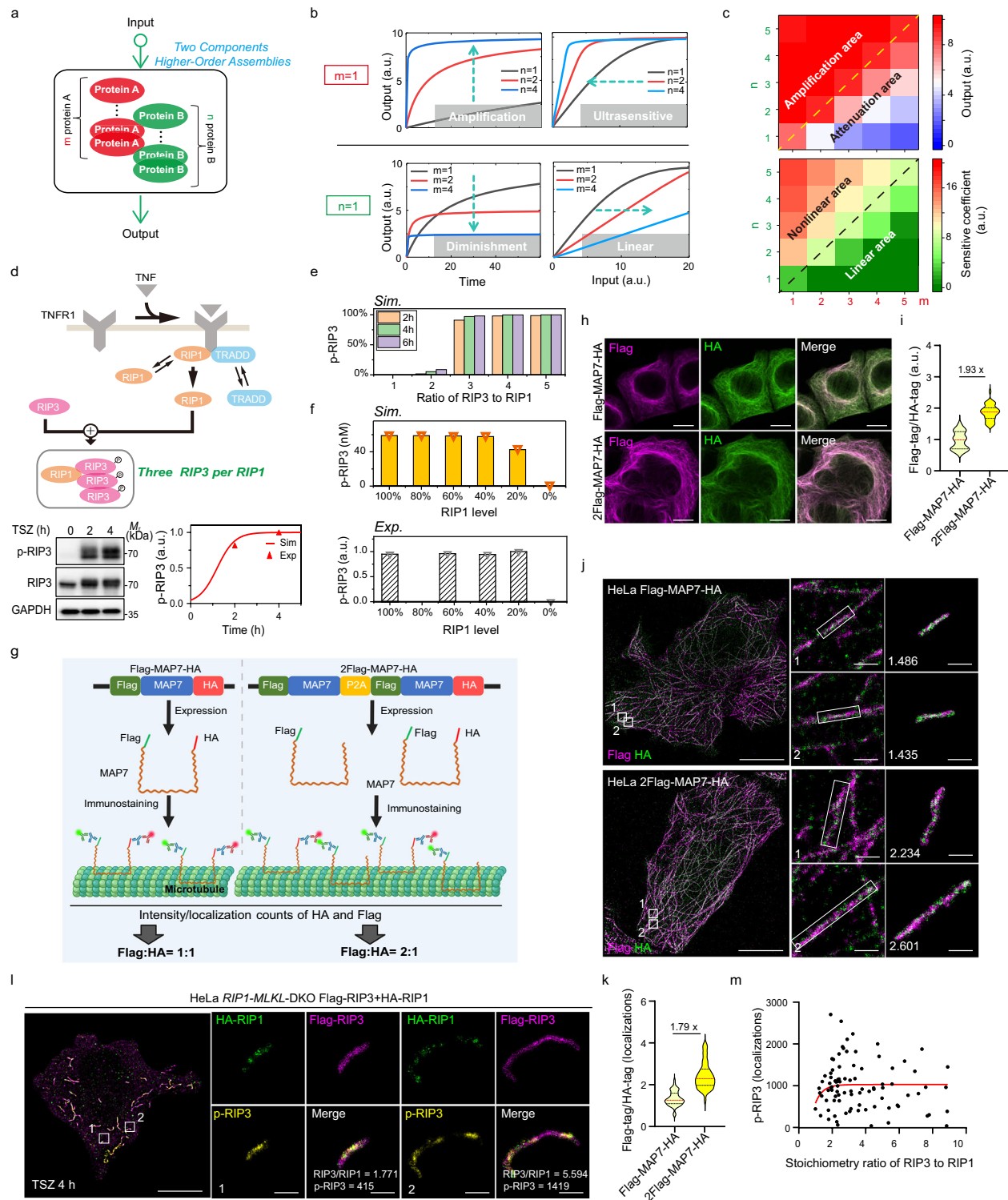

(Fig. 3b, lower panel). Moreover, as A transitions from a monomeric to dimeric or tetrameric structure, the response pattern of output to input becomes increasingly linear. A more comprehensive analysis considering various stoichiometries of A and B delineates the boundaries of signal control and response characteristics (Fig. 3c). As shown in our results, when B molecules outnumber A (n > m), signal amplification (upper panel) and nonlinear response (lower panel) predominate. Conversely, when A molecules outnumber B (n < m), signal attenuation and linear response behavior emerge. The greater the disparity between m and n, the more pronounced the corresponding output behaviors become.

To assess the functionality of rod-shaped RIP1-RIP3 complexes, we also developed a comprehensive protein interaction model of TNF-induced necroptotic signaling (Fig. 3d, Supplementary Fig. 3, and Supplementary Table 2). Given the quantified RIP3-to-RIP1 ratio of approximately 3:1 (Fig. 2f), we implemented a binding rule where three RIP3 molecules interact with one RIP1 molecule. In necrosomes, caspase-8 safeguards cells against necroptosis through inactivating RIP1 and RIP3[31,32]. To eliminate the interference of caspase-8, we calibrated the model using experimental data of RIP3 phosphorylation obtained with TSZ treatment (Fig. 3d, lower panel). We then varied the number of RIP3 molecules to examine how different stoichiometric

**Fig. 3 | Rod-shaped RIP1-RIP3 architecture in necrosomes enables signal amplification and threshold responses. a** Diagram illustrating a basic model depicting higher-order complexes assembly by two components. **b** Effects of varying the number of B (n) or A (m) molecules on output, with one component fixed at 1 within the subunit. Green arrows indicate changes in output signal and regulatory behavior with increasing input intensity as n or m varies. **c** Impact of diverse assembly stoichiometries on output (upper panel) and regulatory behavior (lower panel). The sensitivity coefficient quantifies the threshold response by measuring the reduction in stimulus intensity required to decrease the output from its maximum (20 a.u.). **d** Schematic of necrosome assembly in *MLKL*-KO HeLa cells. Immunoblots are representative of three independent experiments. p-RIP3 was quantified by band intensity. The kinetic model (Supplementary Table 2) recapitulates experimental RIP3 phosphorylation dynamics. **e** Model predictions of how RIP3:RIP1 stoichiometries in necrosomes affect necroptotic signal output. **f** Predicted effect of reduced RIP1 expression on RIP3 phosphorylation (upper panel), alongside experimental validation using literature data[9] (lower panel). p-RIP3 was quantified by band intensity and RIP1 levels were controlled by shRNA-

mediated knockdown. **g** Schematic of MAP7-based calibration system created in BioRender (https://BioRender.com/zaak4bk).HeLa cells expressing Flag-MAP7-HA or Flag-MAP7-P2A-Flag-MAP7-HA ("2Flag-MAP7-HA") are expected to display Flag:HA stoichiometries of 1:1 and 2:1, respectively. Dual-color confocal (**h**) and STORM (**j**) images of HeLa cells expressing Flag-MAP7-HA or 2Flag-MAP7-HA, immunolabeled with anti-Flag (purple) and anti-HA (green) antibodies. Quantification of fluorescence intensity (**i**; n = 40 fields per group) and localization counts (**k**; n = 45 structures per group) confirmed Flag:HA ratios matched expected values. Insets show dual-labeled tubulin with calculated Flag:HA ratios. **l** Three-color STORM of necrosomes in HA-RIP1/Flag-RIP3–expressing *RIP1-MLKL*-DKO HeLa cells treated with TSZ for 4 h, labeled for HA-RIP1 (green), Flag-RIP3 (purple), and p-RIP3 (yellow). Insets highlight necrosomes with calculated Flag:HA ratios and p-RIP3 localizations. **m** Quantification of RIP3:RIP1 stoichiometry and relative p-RIP3 levels (localizations) in individual necrosomes (n = 89). Images in (**h, j, l**) are representative of two independent experiments. Scale bars: 10 μm (**h**, overviews in **j** and **l**) and 300 nm (insets in **j** and **l**). Source data are provided as a Source Data file.

---

ratios of RIP3 to RIP1 influence RIP3 activation. Simulations showed that a subunit composed of a single RIP3 molecule (n = 1) and one RIP1 molecule (m = 1), i.e., a 1:1 RIP3:RIP1 stoichiometry, fails to induce RIP3 phosphorylation (Fig. 3e), corroborating previous experimental data showing that a RIP1-RIP3 heterodimer alone is insufficient to trigger necroptosis[13,33]. Likewise, a subunit containing two RIP3 molecules (n = 2, m = 1), corresponding to a 2:1 RIP3:RIP1 stoichiometry, barely induces RIP3 phosphorylation. Only oligomers with at least three RIP3 molecules per one RIP1 (n = 3 or greater), corresponding to a RIP3:RIP1 ratio of three to one or higher, effectively promote extensive phosphorylation (Fig. 3e). Thus, the 3:1 stoichiometric ratio of RIP3 to RIP1 defines the threshold size for initiating necroptosis. The higher abundance of RIP3 relative to RIP1 within necrosomes serves as an effective mechanism for signal amplification. As an inherent property of supramolecular complexes, RIP3 stoichiometry also elicits a threshold response in necroptotic signaling. Simulations demonstrated that when a 3:1 stoichiometric ratio is present, RIP3 phosphorylation remains relatively stable as RIP1 levels decrease from 100% to 20%, but drops sharply below this threshold (Fig. 3f, upper panel). This threshold behavior was experimentally validated in our previous study[9], showing shRNA-mediated RIP1 knockdown scarcely impacted RIP3 phosphorylation determined by immunoblot densitometry, whereas RIP1 depletion abolished RIP3 phosphorylation (Fig. 3f, lower panel).

To experimentally validate the model predictions with reliable quantitative STORM, we designed a microtubule-associated protein MAP7-based calibration system. We introduced either a Flag-MAP7-HA or a dual-tagged 2×Flag-MAP7-HA construct (via a P2A system) into HeLa cells, ensuring defined antigen stoichiometry (Fig. 3g). Confocal microscopy confirmed robust labeling and canonical microtubule morphology in both constructs (Fig. 3h). Quantification of fluorescence intensity showed an expected ~2-fold increase in Flag:HA signal in 2Flag-MAP7-HA cells compared to Flag-MAP7-HA (1.93×; Fig. 3i). Subsequent STORM imaging revealed that the average localization ratio of Flag to HA closely matched the theoretical value (1.79× vs. 2.00×; Fig. 3j, k). These results demonstrate that, despite inherent uncertainties in antibody recognition and dye photophysics, the MAP7-based reference system conceptually validates the quantitative potential of antibody-based dual-color STORM when combined with high-quality reagents and calibration controls.

Using this system, we next examined RIP1 and RIP3 stoichiometry in individual necrosomes. Three-color STORM was performed in TSZ-treated *RIP1-MLKL* double-knockout (DKO) HeLa cells reconstituted with Flag-RIP1 and HA-RIP3 (Fig. 3l). The immunostaining procedures, antibody selection (e.g., anti-Flag and anti-HA), and imaging/analysis workflows were identical to those used in MAP7 calibration, ensuring quantitative comparability. Segmentation and overlap analysis showed

a wide distribution of RIP3:RIP1 ratios across necrosomes at 4 h post-TSZ, ranging from ~1.0 to ~9.0, with an average of 3.6 (Fig. 3m and Supplementary Fig. 4). To probe necrosome signaling output, we quantified RIP3 activation within each necrosome. The experimental distribution of p-RIP3 closely mirrors the model prediction (Fig. 3e): p-RIP3 remains near baseline when RIP3:RIP1 ≤ 2, exhibits a sharp, switch-like rise as the ratio approaches ~3, and then plateaus near maximal levels for ≥3. This concordance demonstrates that the RIP3:RIP1 stoichiometric ratio functions as a critical determinant, triggering pronounced signal amplification and switch-like behavior in necrosome activation.

## Signal attenuation serves as an intrinsic size control mechanism for RIP3 stoichiometry

Current research strongly supports the critical role of supramolecular assemblies, such as the RIP3 stoichiometry in our study (Fig. 3), in amplifying signal transduction[1,2,34]. However, the expansion of higher-order complexes is not limitless and must adhere to intrinsic constraints. How do cells terminate signaling from such higher-order assemblies? Specifically, once RIP3's higher-order complex is assembled, what mechanisms control its spatial size? We expanded the basic mathematical model (Fig. 3a) to encompass three constituents: A, B, and C, representing upstream, midstream, and downstream signals, respectively (Fig. 4a and Supplementary Table 1). Upon stimulation, A recruits and activates B, which subsequently interacts with C to form a complex. Within the complex, A, B, and C are capable of forming various higher-order assemblies, with m, n, and h denoting the number of molecules of A, B, and C. We analyzed different assembly ratios to elucidate how these strategies define output responses (Fig. 4b). Three representative scenarios were selected for clarity: When both A and B are monomeric (m = 1, n = 1), yielding an A:B:C stoichiometry of 1:1:h, the output progressively intensifies as C's assembly (h) transitions to higher-order assemblies (Fig. 4b, blue line, right panel). When A is monomeric (m = 1) and C forms a trimer (h = 3), corresponding to an A:B:C stoichiometry of 1:n:3, the output initially rises and then declines as B's assembly degree increases, peaking when B approaches a dimer (n = 2) (Fig. 4b, green line, right panel). Similarly, when B is monomeric (n = 1) and C forms a trimer (h = 3), giving an A:B:C stoichiometry of m:1:3, the output reaches its maximum when A approaches a dimer (m = 2) (Fig. 4b, red line, right panel). Based on principles of maximum efficiency and optimal signaling, cells preferentially select the optimal ratio for efficient signal transduction.

Theoretically, the existence of the optimal stoichiometry of B can be intuitively recognized. As depicted in Fig. 4c, the three-component system can be decomposed into two two-component systems (TCS). In TCS-1, consisting of upstream A and midstream B, B functions as a downstream signal of A. Analysis of two-component systems shows

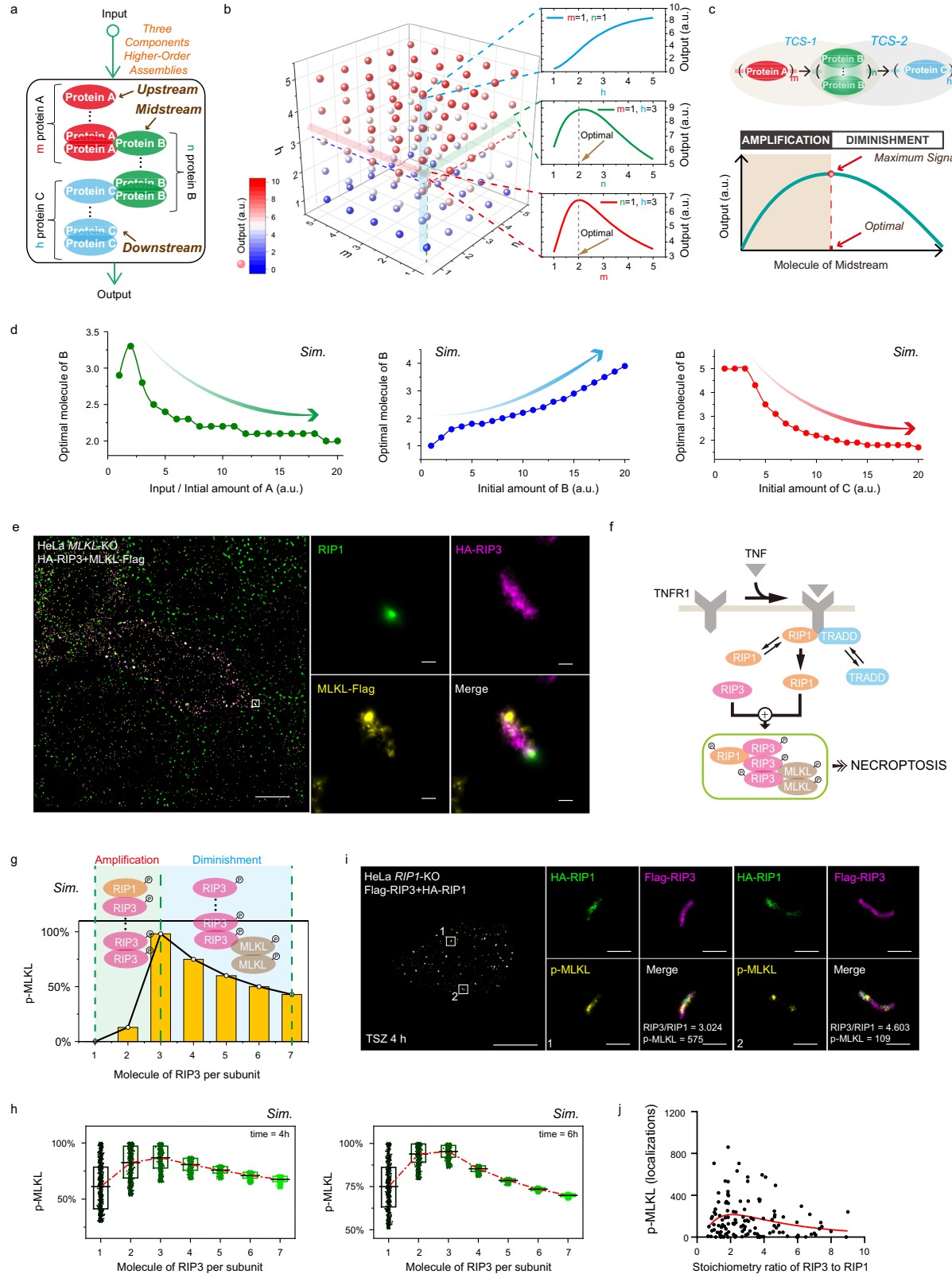

that increasing B's assembly degree enhances output signal (Fig. 3c), with B's higher-order assembly acting as a signal amplifier. However, in TCS-2, where B is midstream and C is downstream, B serves as an upstream signal to C. Here, increasing B's assembly degree reduces output intensity, with B's higher-order assembly serving as an inhibitor. Thus, excessively higher-order assembly of B serves as a mechanism for signal attenuation or even termination. This

attenuation restricts further incorporation of B into complex, functioning as an intrinsic spatial size control mechanism. While moderate higher-order assembly of B amplifies signal, achieving an optimal stoichiometry is essential to preclude excessive assembly that could attenuate or terminate signal, thereby ensuring efficient transmission.

To explore the regulation of the optimal stoichiometry, additional simulations were conducted. Input intensity and component levels

**Fig. 4 | Optimal molecule stoichiometry of RIP1-RIP3 assemblies in necrosomes. a** Diagram illustrating model of higher-order complexes formed by three components. **b** Impact of diverse assembly configurations of A, B and C on output. **c** Intuitive explanation of the optimal configuration for the midstream signal. The three-component complex can be divided into two assemblies: TCS-1, where B's higher-order assembly amplifies the signal from A; and TCS-2, where B inhibits signal transmission to C. **d** Impact of input intensity and initial levels of A, B, and C on B's optimal assembly configuration. **e** Multi-color STORM images of *MLKL*-KO HeLa cells expressing HA-RIP3 and MLKL-Flag after TSZ treatment, labeled for RIP1 (green), HA-RIP3 (purple) and MLKL-Flag (yellow). Images are representative of three independent experiments. **f** Schematic of TNF-induced RIP1-RIP3-MLKL necrosomes assembly. Details of the model are given in Supplementary Table 2. **g** RIP3 higher-order assembly's impact on MLKL phosphorylation. **h** Statistical analysis of $10^4$ random models showing the influence of RIP3 assembly on MLKL phosphorylation

efficiency, with approximately 3:1 ratio of RIP3 to RIP1 consistently yielding the highest efficiency at 4 h and 6 h. Box plots: centre line, median; box bounds, 25th and 75th percentiles (IQR); whiskers extend to the lowest and highest data points within 1.5×IQR from the lower and upper quartiles. To visualize the distribution, all simulation outcomes are overlaid as jittered dots. The overlaid solid line connects the per-condition mean of the 500 simulations to show the trend. For each condition, $n = 500$ independent stochastic simulations. **i** Three-color STORM of necrosomes in HA-RIP1/Flag-RIP3–expressing *RIP1*-KO HeLa cells treated with TSZ for 4 h, labeled for HA-RIP1 (green), Flag-RIP3 (purple), and p-MLKL (yellow). Insets highlight necrosome morphology with calculated Flag:HA ratios and p-MLKL localizations. Images are representative of two independent experiments. **j** Quantification of RIP3:RIP1 stoichiometry and relative p-MLKL levels (localizations) in individual necrosomes ($n = 122$). Scale bars: 10 μm (**i**, overview), 5 μm (**e**, overview), 300 nm (**i**, insets), and 100 nm (**e**, insets). Source data are provided as a Source Data file.

differentially influence B's configuration. Modulating B's assembly degree by input and upstream A exhibits similar trends (Fig. 4d, green dotted line). At low input intensities or A levels (<-2.5 arbitrary units (a.u.)), increases in either result in higher assembly degrees of B. Conversely, at high input intensities or A levels, further increases lead to a reduction in B's assembly degree. Increasing B's level enhances its higher assembly degree (Fig. 4d, blue dotted line). Unexpectedly, downstream C also regulates B's stoichiometry (Fig. 4d, red dotted line). At low C levels (<-3.0 a.u.), increases have no significant effect on B's stoichiometry, whereas higher C levels induce lower assembly degrees of B.

To directly assess the spatial architecture of necrosomes, we performed three-color STORM imaging of HA-RIP3 and MLKL-Flag in *MLKL*-KO HeLa cells (Fig. 4e). These experiments revealed rod-shaped complexes with co-localization of RIP1, RIP3, and MLKL, in agreement with our single- and two-color STORM results (Figs. 2b and 3l). To evaluate the functionality of rod-shaped RIP1-RIP3-MLKL complexes, we expanded our protein interaction model to incorporate MLKL (Fig. 4f and Supplementary Table 2). Given the quantified RIP1 to MLKL ratio of 1:2 within necrosomes (Fig. 2f), we postulated that each subunit comprises one RIP1 and two MLKL molecules, establishing a baseline RIP1:RIP3:MLKL stoichiometry of 1:n:2. We altered the number of RIP3 to evaluate its impact on MLKL activation. Consistent with the two-component RIP1-RIP3 results (Fig. 3e), stoichiometries of 1:1:2 or 1:2:2 (one or two RIP3 per subunit) produced little MLKL phosphorylation (Fig. 4g). Three RIP3 per RIP1 emerged as the threshold size for significant MLKL phosphorylation. Increasing RIP3 beyond this level, yielding 1:4:2 or higher, reduced phosphorylation efficiency, indicating that over-assembly is inhibitory. Within RIP1-RIP3-MLKL complexes, RIP3 serves as the midstream signal: its higher-order assembly amplifies upstream RIP1 signaling, while simultaneously inhibiting downstream MLKL signaling. Variability exists between cells. To simulate this, we exhaustively modeled the binding/unbinding strengths of all protein-protein interactions, along with the rates of protein synthesis/degradation in TNF pathway leading to necrosome formation, within reasonable physiological parameter ranges[9,35,36]. We selected $10^4$ random TNF pathway models and calculated their MLKL phosphorylation efficiency under varying RIP3 assembly ratios. The results showed significant differences in the kinetic evolution, with synchronization among models improving as the degree of RIP3 assembly increased (Supplementary Fig. 5). Statistical analysis indicated that MLKL phosphorylation efficiency peaked at a RIP3:RIP1 ratio of three to one (Fig. 4h), suggesting that this stoichiometry is prevalent in necrosomes.

To experimentally validate the optimal RIP3 stoichiometry, we conducted three-color STORM imaging in *RIP1*-KO HeLa cells expressing Flag-RIP1 and HA-RIP3, using monoclonal p-MLKL antibodies (Fig. 4i). Quantification of p-MLKL as a function of the RIP3:RIP1 ratio exhibits a unimodal distribution of p-MLKL, peaking at a RIP3:RIP1 ratio of ~3 (Fig. 4j). This trend closely matched our simulation (Fig. 4g, h),

confirming that stoichiometric balance between RIP3 and RIP1 governs optimal necrosome activation. Together, these data uncover signal attenuation as a key regulatory mechanism that defines the spatial size and activity of RIP3 supramolecular assemblies in necroptosis.

## RIP3 stoichiometry can be selectively targeted by multiple strategies

The roles of RIP1, RIP3, and MLKL in necroptosis are well-established, yet the spatial regulation of RIP3 assembly by these proteins remains unclear. We computed a two-dimensional phase diagram of MLKL activation based on the degree of RIP3 assembly within necrosomes (Fig. 5a–c). Simulation predicted that high TNF stimulation (TNF > -50 nM) most efficiently induces MLKL phosphorylation with a 3:1 stoichiometric ratio of RIP3 to RIP1 (Fig. 5a). Low TNF stimulation requires larger assemblies. At very low TNF levels (TNF <-5 nM), eight RIP3 per RIP1 are required for effective MLKL phosphorylation. Therefore, RIP3 stoichiometries for MLKL activation range from three and eight molecules per RIP1. Upstream RIP1 exhibits similar regulatory behavior towards RIP3 assembly, with lower RIP1 levels correlating with higher RIP3 assembly degrees (Supplementary Fig. 6). Additionally, a positive correlation exists between RIP3 levels and its assembly degrees (Fig. 5b). Higher RIP3 levels require larger assemblies (four to eight molecules per RIP1) for effective MLKL activation. As a downstream signal, elevated MLKL levels minimally affect RIP3 stoichiometry (Fig. 5c), while reduced MLKL levels necessitate larger RIP3 assemblies, indicating that MLKL limits the supramolecular assembly of RIP3.

To test these model predictions, we performed STORM imaging analysis of RIP3 assembly in necrosomes under various TNF concentrations in Flag-RIP3 HeLa cells. Immunoblotting confirmed that increasing TNF stimulation enhances RIP3 and MLKL phosphorylation (Fig. 5d, left panel). Consistently, low TNF produced fewer but larger RIP3 assemblies, whereas high TNF yielded more numerous but smaller assemblies, revealing a clear inverse relationship between TNF concentration and the degree of higher-order RIP3 assembly (Fig. 5d, right panel and 5e). Thus, higher stimulation correlates with smaller RIP3 assemblies but a larger count of necrosomes. We next employed doxycycline (Dox)-induced RIP3-WT expression to obtain varying RIP3 levels in cells (Fig. 5f, left panel). Increased RIP3 expression leads to elevated RIP3 and MLKL phosphorylation, as well as the formation of larger RIP3 assemblies (Fig. 5f, right panel). Statistical analysis reveals a substantial rise in both RIP3 assembly degree and the average count of necrosomes per cell with higher RIP3 levels (Fig. 5g). Thus, RIP3 abundance positively regulates both its own supramolecular assembly and necrosome quantity.

To assess the role of MLKL, we knocked down MLKL to different expression levels using MLKL-specific shRNA (Fig. 5h, left panel). While MLKL is not expected to influence RIP3 assembly, imaging shows that lower MLKL levels result in fewer but larger RIP3 assemblies (Fig. 5h, right panel and 5i), suggesting MLKL not only receives signals from

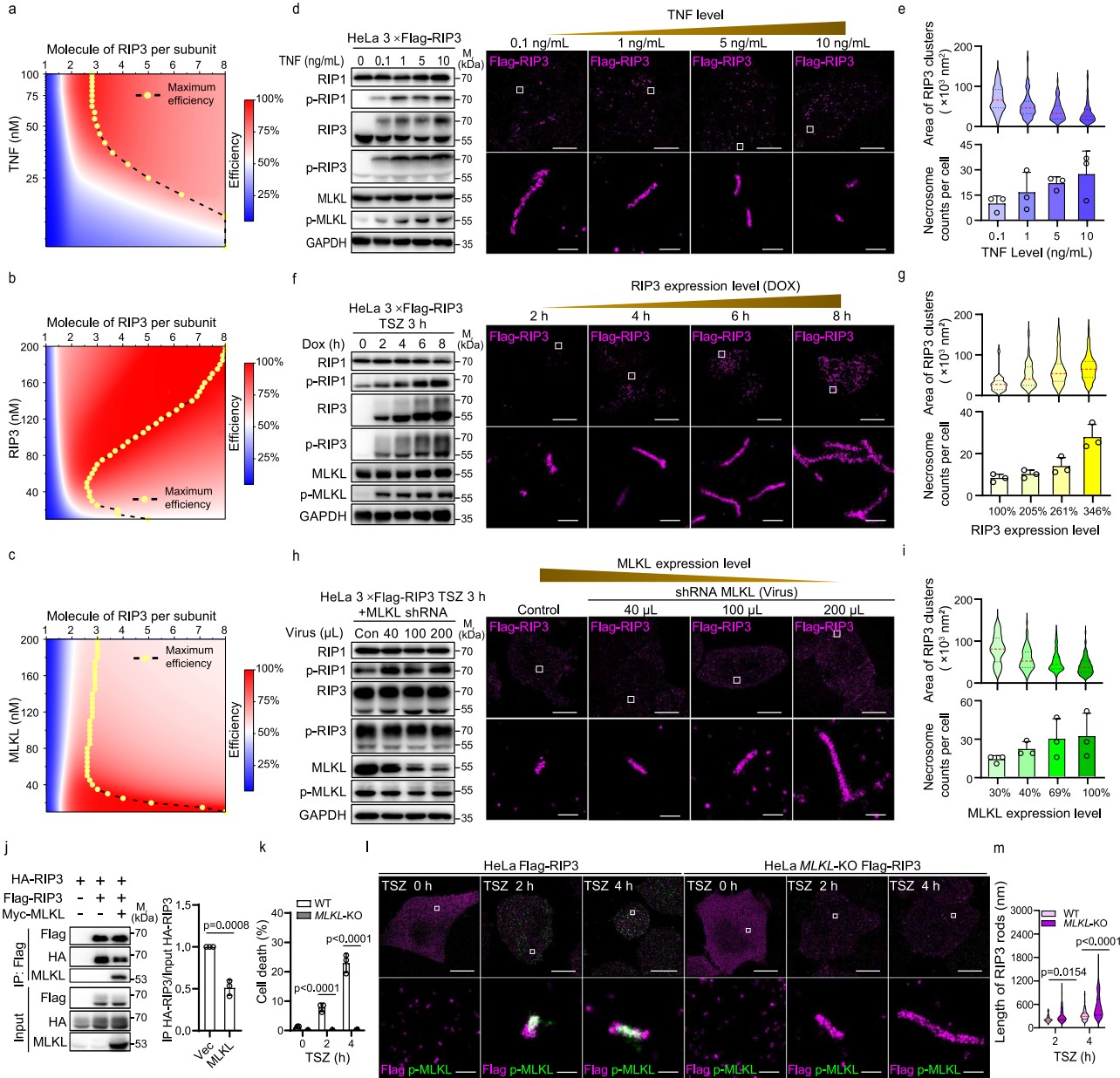

**Fig. 5 | Assembly strategies of RIP1-RIP3 necrosomes under various physiological conditions.** Theoretical predictions illustrating the regulation of TNF (**a**), RIP3 (**b**), and MLKL (**c**) on RIP1-RIP3 assembly. Here, p-MLKL/MLKL$_{tot}$ represents system efficiency with MLKL$_{tot}$ indicating MLKL expression level. Maximum efficiency corresponds the optimal RIP3 assembly degrees under given conditions. Experimental validation of TNF (**d**), RIP3 (**f**), and MLKL (**h**) effects on RIP3 assembly via immunoblotting and STORM. Changes in RIP3 and MLKL phosphorylation, and corresponding rod-shaped RIP3 structures (STORM), are shown under varying TNF stimulation (**d**), doxycycline-induced RIP3 expression (**f**), and shRNA-mediated MLKL knockdown (**h**).**e, g, i,** Statistical analysis of areas and counts of rod-shaped RIP3 structures in STORM images from cells under different TNF doses (**e**), RIP3 expression levels (**g**), and MLKL expression levels (**i**). Experiments in (**d-i**) used Flag-RIP3-expressing HeLa cells treated with TSZ under graded TNF concentrations (**d,e**; $n = 28,48,64,80$ structures), induced RIP3 expression (**f,g**; $n = 24,30,42,86$), or MLKL knockdown (**h, i**; $n = 40,64,88,94$). RIP3/MLKL levels were quantified by immunoblot densitometry. Necrosome counts were presented as mean ± SD from three cells. **j** Co-immunoprecipitation in 293 T cells transiently expressing HA-RIP3, Flag-RIP3, and Myc-MLKL. Lysates were immunoprecipitated with anti-Flag beads, and inputs/IPs were immunoblotted for Flag, HA, and MLKL. HA-RIP3 binding to Flag-RIP3 was quantified as IP/input ratio. Data are presented as mean ± SD from three independent experiments. **k−m** Flag-RIP3−expressing wild-type or MLKL-KO HeLa cells treated with TSZ over time. Cell death was measured by PI uptake and shown as mean ± SD from three biological replicates (**k**). Dual-color STORM of RIP3 and p-MLKL (**l**); insets show necrosome morphology. RIP3 fibril lengths were quantified (**m**; n = 67/111 structures at 2 h, 64/135 at 4 h for WT/KO). Data in (**d, f, h** and **l**) are representative of two independent experiments. *p* values were calculated using unpaired two tailed t test (**j**), or Two-Way ANOVA Tukey's multiple comparisons test (**k** and **m**). Scale bars: 10 µm (overviews in **d, f, h,** and **l**) and 200 nm (insets in **d, f, h,** and **l**). Source data are provided as a Source Data file.

RIP3 but also negatively regulates RIP1-RIP3 supramolecular assembly. To confirm MLKL-mediated feedback on RIP3 activation, we examined the interaction between Flag-RIP3 and HA-RIP3 in the presence or absence of overexpressed Myc-MLKL. MLKL expression reduced the amount of HA-RIP3 pulled down with Flag-RIP3, suggesting that MLKL weakens RIP3 homo-interaction and thereby inhibits RIP3 oligomerization (Fig. 5j). We further examined RIP3 fibril dynamics over time in MLKL-proficient and MLKL-deficient cells (Fig. 5k−m). In MLKL-deficient cells, RIP3 fibrils continued to elongate over time, but cell death remained low. In contrast, in MLKL-proficient

cells, RIP3 fibril growth plateaued as p-MLKL levels increased, followed by cell death. These data demonstrate that MLKL provides negative feedback, constraining RIP3 fibril elongation and ensuring signal termination.

Our data and prior structures support a unified structural rationale for the phase behavior in Fig. 5a–c (Supplementary Fig. 7). TNF elevates RIPK1 recruitment, favoring the alternated RIPK1:RIPK3 RHIM β-spine and effectively capping or diverting RIP3 from homo-elongation, thereby shortening RIP3 filaments[10]. Increasing RIP3 expression follows mass-action logic and promotes RHIM-mediated axial stacking, yielding longer homo-filaments[28]. By contrast, higher MLKL expression correlates with shorter RIP3 filaments without a concomitant rise in MLKL phosphorylation (Fig. 5k–m). This is consistent with structural evidence that MLKL binding induces a conformationally inactive state of RIP3, reducing the availability of stacking-competent RHIM interfaces and redistributing RIP3 into smaller assemblies[29,37]. In summary, RIP3 supramolecular assembly is dynamically regulated by multiple factors (i.e., TNF dosage, RIP1 abundance, RIP3 expression, and MLKL levels), ensuring precise spatial control over necrosome formation and necroptotic output.

## Regulatory strategies governing RIP3 stoichiometry are ubiquitous

Elucidating the general mechanisms governing the assembly of supramolecular signalosomes is of urgent importance. To assess whether RIP3 assembly strategies within necrosomes exhibit universality, we conducted a comprehensive computational analysis across all possible scenarios of TNF-induced necrosome formation, including stochastic variations in RIP3 assembly degree (Fig. 6a). We uniformly sampled $10^4$ parameter sets within reasonable biological intervals on a logarithmic scale in the $n_p$-dimensional parameter space[9,35,36]. For each parameter set, we calculated the respective MLKL phosphorylation efficiency corresponding to the stochastic RIP3 stoichiometry, generating a distribution plot encompassing all $10^4$ models across the phase plane defined by RIP3 stoichiometry and MLKL phosphorylation efficiency (Fig. 6a, right panel). Drawing from the proposition that necrosome assemblies are optimized for signal transduction, we identified the top 5% of models with the highest efficiencies and conducted statistical analyses on their corresponding average RIP3 ratios to RIP1. Subsequently, we explored the general regulatory mechanisms underlying RIP3 assembly by examining how factors such as TNF intensity and RIP3 or MLKL levels influence RIP3 stoichiometry.

The distributions of RIP3 ratio and corresponding efficiency for the top 5% of models under varying TNF stimulation intensities and RIP3/MLKL expression levels are illustrated in Fig. 6b. Statistical analysis reveals that at 10 nM TNF stimulation, RIP3 ratios are primarily distributed between 4.5 and 8, with an average value of approximately 6.2 (Fig. 6c, upper panel). Under 20 nM TNF stimulation, the ratio predominantly falls between 1.6 and 6, averaging around 4. As stimulation intensity increases, the average RIP3 ratio consistently decreases. When TNF exceeds 70 nM, the ratio is distributed between 1 to 4, with an average of approximately 3. In contrast, higher levels of RIP3 lead to a gradual increase in both the distribution range and average ratio (Fig. 6c, middle panel). At a RIP3 expression level of 10 nM, ratios range from 1 to 4, with an average of about 2.5. Conversely, at 200 nM RIP3, ratios primarily fall between 2 and 8, averaging around 5.5. The influence of MLKL on RIP3 ratio mirrors that of TNF (Fig. 6c, lower panel). At a MLKL expression level of 10 nM, RIP3 ratios range from 3 to 7.5, averaging approximately 6. However, when MLKL exceeds 50 nM, the ratios drop to between 1 and 4, with an average near 3. These statistical findings suggest that in necrosomes, TNF negatively regulates RIP3 homo-assembly, RIP3 positively regulates its own assembly, and MLKL exerts a negative effect, all through a non-specific parameter-dependent mechanism (Fig. 6d). This theoretical framework supports the universality of these regulatory mechanisms across different cell types.

Besides discussing RIP3 configuration, we examined the trends in MLKL/RIP3 phosphorylation efficiency under various conditions. With escalating TNF stimulation, MLKL phosphorylation efficiency progressively rises (Fig. 6e, upper panel). RIP3 phosphorylation efficiency remains consistently high without a discernible trend, as corroborated by the quantitative experimental data illustrated in Fig. 5d (Fig. 6f, upper panel). This indicates that under low TNF stimulation, cells harbor fewer necrosomes, necessitating the formation of larger RIP3 assemblies within each necrosome to sustain a heightened overall phosphorylation level (Fig. 5e). Conversely, during heightened stimulation, the abundance of necrosomes escalates, allowing RIP3 to maintain its phosphorylation level with shorter chains. The impact of RIP3 expression on MLKL/RIP3 phosphorylation efficiency parallels that of TNF stimulation (Fig. 6e, middle panel). Higher RIP3 levels enhance MLKL phosphorylation efficiency, while RIP3 phosphorylation efficiency remains high. This aligns with the experimental data shown in Fig. 5f (Fig. 6f, middle panel), indicating that under low RIP3 expression, smaller RIP3 assemblies suffice to maintain high RIP3 phosphorylation efficiency. Conversely, at higher RIP3 levels, cells require more necrosomes and larger RIP3 assemblies to sustain this efficiency. Despite the consistently high RIP3 phosphorylation efficiency, lower RIP3 expression levels result in a smaller total amount of phosphorylated RIP3 within cells, leading to relatively lower MLKL phosphorylation efficiency. Furthermore, as MLKL expression levels increase, MLKL phosphorylation efficiency gradually decreases, while RIP3 phosphorylation efficiency remains consistently high (Fig. 6e, down panel). This trend is also evident in the experimental data from Fig. 5h (Fig. 6f, down panel). At low MLKL levels, characterized by fewer necrosomes, larger RIP3 assemblies are required to maintain a heightened overall RIP3 phosphorylation level (Fig. 5i). However, at high MLKL levels, with more necrosomes present, smaller RIP3 assemblies are adequate. Despite the constant RIP3 phosphorylation level, the limited capacity of RIP3 to activate MLKL results in decreased MLKL phosphorylation efficiency as MLKL expression levels rise. In summary, cells adaptively modulate the assembly strategy of RIP3 within necrosomes, as well as the quantity of necrosomes, to maintain efficient RIP3 phosphorylation and signal transduction under varying conditions (Fig. 6g). When stimulation or core protein levels are low, resulting in fewer necrosomes, RIP3 forms larger assemblies to ensure high-efficiency phosphorylation. Conversely, under conditions of high stimulation or elevated protein levels, where more necrosomes are present, RIP3 only needs to form smaller assemblies to maintain high phosphorylation levels.

## Distinct stoichiometries of RIP3 and caspase-8 within necrosomes orchestrate a biphasic necroptotic response

Necroptotic process is blocked by caspase-8 within necrosomes through the cleavage of RIP1 and RIP3[31,32]. The ratio of caspase-8 to RIP1 is approximately 1:1 in the early stages (Fig. 2f). To quantitatively evaluate the functionality of RIP1-FADD-caspase-8 complex, we developed a protein interaction model of TNF signaling in RIP3-depleted cells (Fig. 7a and Supplementary Table 2). Simulation results indicate that at a 1:1 ratio between caspase-8 and RIP1, the recruitment of caspase-8 into RIP1-caspase-8 complex decreases linearly as RIP1 levels diminish (Fig. 7b, upper panel). This linear response aligns with our earlier experiments, where RIP1 knockdown progressively reduced caspase-8 recruitment upon treatment (Fig. 7b, lower panel)[9].

Beyond the linear/nonlinear responses elicited by higher-order assemblies, the regulatory behavior of these signalosomes is more complicated than previously thought. Caspase-8 activation exhibits a linear dependency on RIP1 levels, while RIP3 activation displays a nonlinear pattern (Figs. 3e and 7b). Within necrosomes, both RIP1-FADD-caspase-8 and RIP1-RIP3 complexes play critical roles. RIP1-

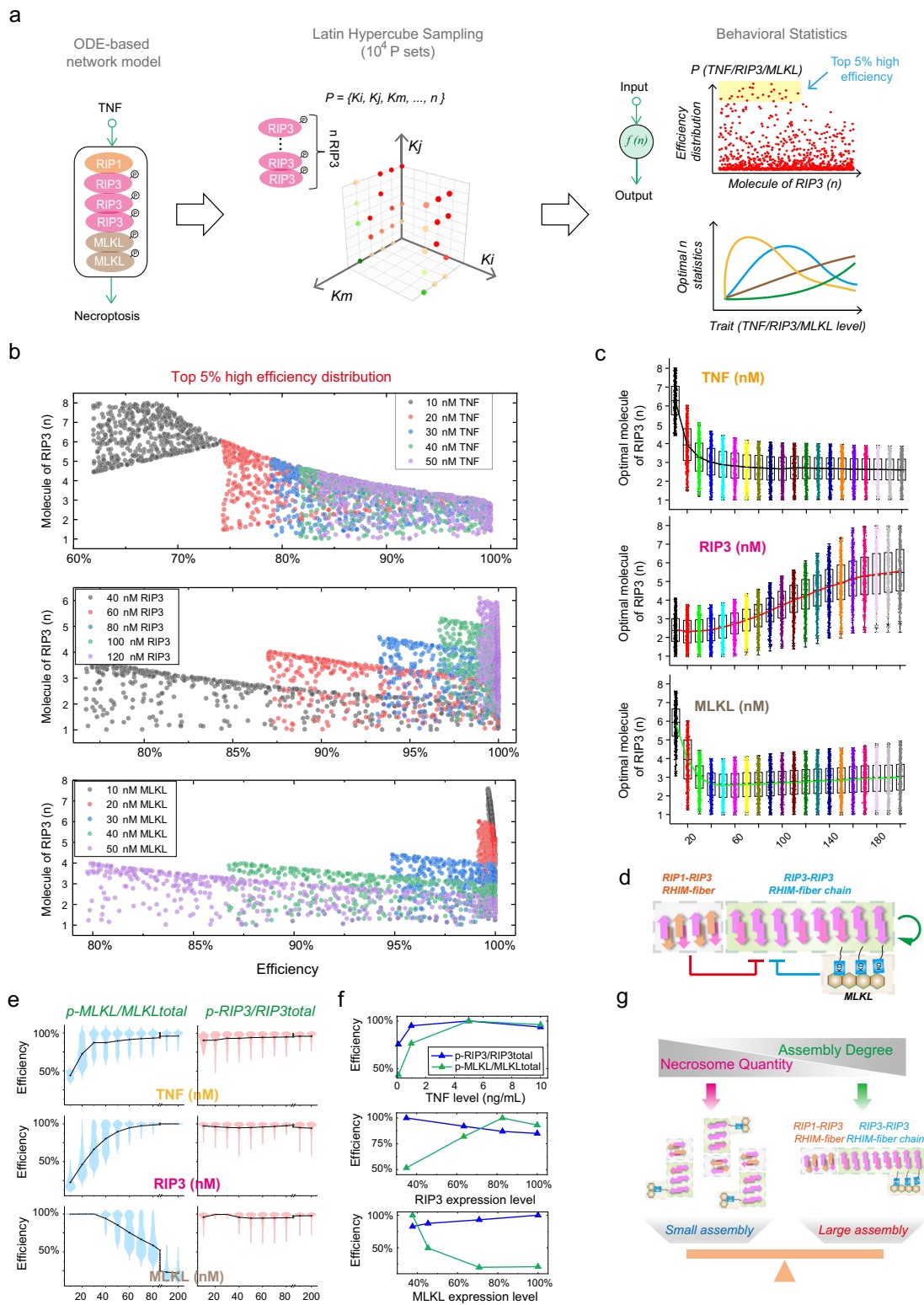

mediated caspase-8 cleaves RIP3 to block necroptosis[31,32], while RIP3 also suppresses caspase-8 activation[38]. This mutual inhibition between caspase-8 and RIP3 likely contributes to a biphasic response in downstream signaling (Fig. 7c). To examine this, incorporating this mutual inhibition into model revealed that RIP3 phosphorylation (p-RIP3) initially increases with decreasing RIP1 levels but then sharply declines with further reduction in RIP1 (Fig. 7d, upper panel). The

biphasic response of p-RIP3 to RIP1 is corroborated by the paradoxical experimental observations, showing that although RIP1 is essential for necroptosis, its elevation down-regulates necroptosis (Fig. 7d, lower panel)[9,39].

Our quantitative assembly analysis shows that caspase-8 occupancy on RIP1 is low (Fig. 2f), and this stoichiometric arrangement is insufficient to drive robust amplification of the apoptotic cascade in

**Fig. 6 | Exhaustive analysis of the general regulatory strategies of RIP3 assembly. a** Schematic workflow outlining the investigation of regulatory strategies using randomized signaling models. A TNF signaling network model was constructed (Supplementary Table 2) and subjected to LHS method to sample parameter variables within physiological ranges, including stochastic stoichiometries of RIP3 assembly. Kinetic analysis was performed on all sampled models, with the top 5% identified based on MLKL phosphorylation efficiencies. Statistical analysis of these models was conducted to explore trends in RIP3 assembly under varying TNF, RIP3, and MLKL conditions. **b** Distribution of RIP3 ratios to RIP1 and corresponding MLKL phosphorylation efficiencies in the top 5% efficient models under different conditions. **c** Statistical results corresponding to the distribution of the three scenarios shown in panel **b**. Box plots: centre line, median; box bounds,

25th and 75th percentiles (IQR); whiskers extend to the lowest and highest data points within 1.5×IQR from the lower and upper quartiles. To visualize the distribution, all simulation outcomes are overlaid as jittered dots. The overlaid solid line connects the per-condition mean of the 500 simulations to show the trend. For each condition, $n = 500$ independent stochastic simulations; concentrations as indicated (nM). **d** Regulation of TNF, RIP3, and MLKL on RIP3 ratios. **e, f** Theoretical predictions and experimental validations of the impacts of TNF, RIP3, and MLKL on MLKL/RIP3 phosphorylation efficiencies. **g** Schematic illustrating strategic balance between necrosome abundance and RIP3 stoichiometries in cells: high necrosome abundance favors smaller RIP3 assemblies, while low abundance necessitates larger RIP3 assemblies. Source data are provided as a Source Data file.

RIP3-deficient cells. Previous studies have demonstrated that loss of RIP1 predisposes cells to apoptosis, mainly because of impaired NF-κB activation and reduced expression of anti-apoptotic proteins such as c-FLIP[40,41]. To test whether c-FLIP functions as a gatekeeper of caspase-8 activation and aggregation, we treated HeLa cells with TNF plus cycloheximide (TC) to block protein synthesis and performed time-resolved comparisons between wild-type and RIP1-deficient cells at 0, 3, 6, and 9 h (Fig. 7e–g). TC treatment caused a time-dependent decline in cell viability, with a significantly stronger reduction in RIP1-deficient than in wild-type cells (Fig. 7e). Immunoblotting revealed a progressive loss of c-FLIP and faster accumulation of cleaved caspase-8 in RIP1-deficient cells. Moreover, densitometry analysis confirmed steeper slopes in RIP1-deficient cells, indicating accelerated c-FLIP depletion and earlier, stronger caspase-8 activation (Fig. 7f). Confocal imaging provided single-cell evidence, showing that caspase-8 signal changed from diffuse to punctate or oligomeric structures, with RIP1-deficient cells displaying denser and larger aggregates at later time points (Fig. 7g). Collectively, these data suggest that RIP1 supports c-FLIP expression, thereby restraining higher-order caspase-8 oligomerization and suppressing apoptosis in RIP3-deficient settings. In this context, the low stoichiometric occupancy of caspase-8 on RIP1 limits apoptotic amplification, while maintaining regulatory capacity over RIP3 activation. Therefore, RIP3 and caspase-8 form distinct supramolecular configurations within necrosomes, acting as opposing regulators that define the signaling bifurcation between apoptosis and necroptosis. Their stoichiometry and mutual inhibition constitute intrinsic checkpoints that govern cell fate decisions in response to TNF signaling.

## Discussion

While the concept of supramolecular/cooperative assemblies driving signal amplification has been widely reported[1,2], our study extends current conventional view by proposing and elucidating the intrinsic signal attenuation mechanism that governs supramolecular assembly termination. Within necrosomes, RIP3 adopts an optimal stoichiometry to balance signal amplification and attenuation, dynamically adjusting its assembly strategies in response to cellular conditions. Supported by energy-based molecular modeling and our experimental findings, we present a comprehensive framework for the assembly principles and spatiotemporal regulation of TNF-induced cell death signaling (Fig. 8).

Existing studies highlight that higher-order assemblies involving RIP3 and MLKL during necroptosis facilitate signal amplification and efficient transmission[15,42,43]. Here, we propose an alternative perspective: higher-order assemblies of upstream signals attenuate or even terminate downstream signaling. The impact of these assemblies—whether amplifying or attenuating—depends on their hierarchical positioning (Fig. 3c). While RIP3 amplifies signaling through oligomerization as a downstream effector of RIP1, it attenuates signaling as an upstream regulator of MLKL. Our previous mass spectrometry data showed that proteins like NEMO and IKK within TNFR1 complex are present at levels two to three orders of magnitude lower than other

major proteins[9], likely due to their downstream positioning in the ubiquitin chain. This signal-diminishing mechanism, driven by higher-order complexes, possibly allows proteins like NEMO and IKK to exist in smaller signalosome sizes, facilitating the efficient translocation of signals, such as NF-κB, through the nuclear pore into the nucleus. In summary, our findings establish a foundational framework for designing molecular interventions that target supramolecular assembly, enabling the creation of more flexible and precise biological systems.

While the stacking of proteins in signalosomes facilitates spontaneous signal amplification, intrinsic mechanisms are essential to constrain their configuration sizes. We emphasize that excessively higher-order assembly of RIP3 attenuates necroptotic signaling. This attenuation serves as an intrinsic size control mechanism, restricting further RIP3 assembly and preventing uncontrolled signaling. The size constraints mechanism imposed by signal attenuation can manifest in various forms, including dissociation, degradation, phosphorylation, ubiquitination, or feedback processes. In necrosomes, several constraints regulate the assembly size of RIP3 (Fig. 6d). TNF stimulation intensity negatively impacts RIP3 assembly size, mirroring observations in apoptosis, where caspase-8 forms longer chains under low CD95 stimulation and shorter chains under high stimulation[44]. This suggests that the molecular assembly responsive to stimulation may serve as a general regulatory mechanism across various signaling. While prevailing views posit that RIP1 constrains RIP3 assembly through recruiting caspase-8[45,46], our results imply that RIP1 continues to limit RIP3 assembly size even when caspase-8 activity is blocked. Notably, MLKL also restricts RIP3 assembly, indicating that downstream signals can inherently inhibit upstream signal assembly. Currently, no literature documents MLKL's inhibitory effect on RIP3, necessitating further exploration of its underlying mechanisms. One possibility is that abundant MLKL competes with RIP3 for spatial binding sites, preventing the formation of larger RIP3 assemblies. These insights have significant implications for understanding pathological conditions linked to RIP1, RIP3, and MLKL dysregulation. Targeting these molecules, particularly the unexpectedly identified MLKL, to regulate necrosome assembly provides a promising therapeutic approach for treating diseases associated with necroptosis.

RHIM-mediated amyloid formation between RIP1 and RIP3, as well as RIP3 homo-amyloid assembly, is crucial for building rod-like necrosomes and thereby amplifying necroptotic signaling[10,42]. Our previous analyses suggested that RIP3 complexes sized tetramer or larger are essential for necroptosis[13]. Herein, we refine this concept by showing that a three-to-one stoichiometric ratio of RIP3 to RIP1 is the functional configuration, in agreement with recent work indicating that the structural integrity of RIPK3 fibrils with three-β-strand architecture is critical for necroptosis[42]. This stoichiometry amplifies signals and imposes a threshold on RIP1-mediated RIP3 phosphorylation, which improves signal accuracy by filtering out sub-threshold biological noise, preventing erroneous necroptosis initiation. Higher-order assembly of caspase-8 is restrained by c-FLIP in RIP1-mediated complexes, yielding weak apoptotic signaling (Fig. 7g). In the absence of c-

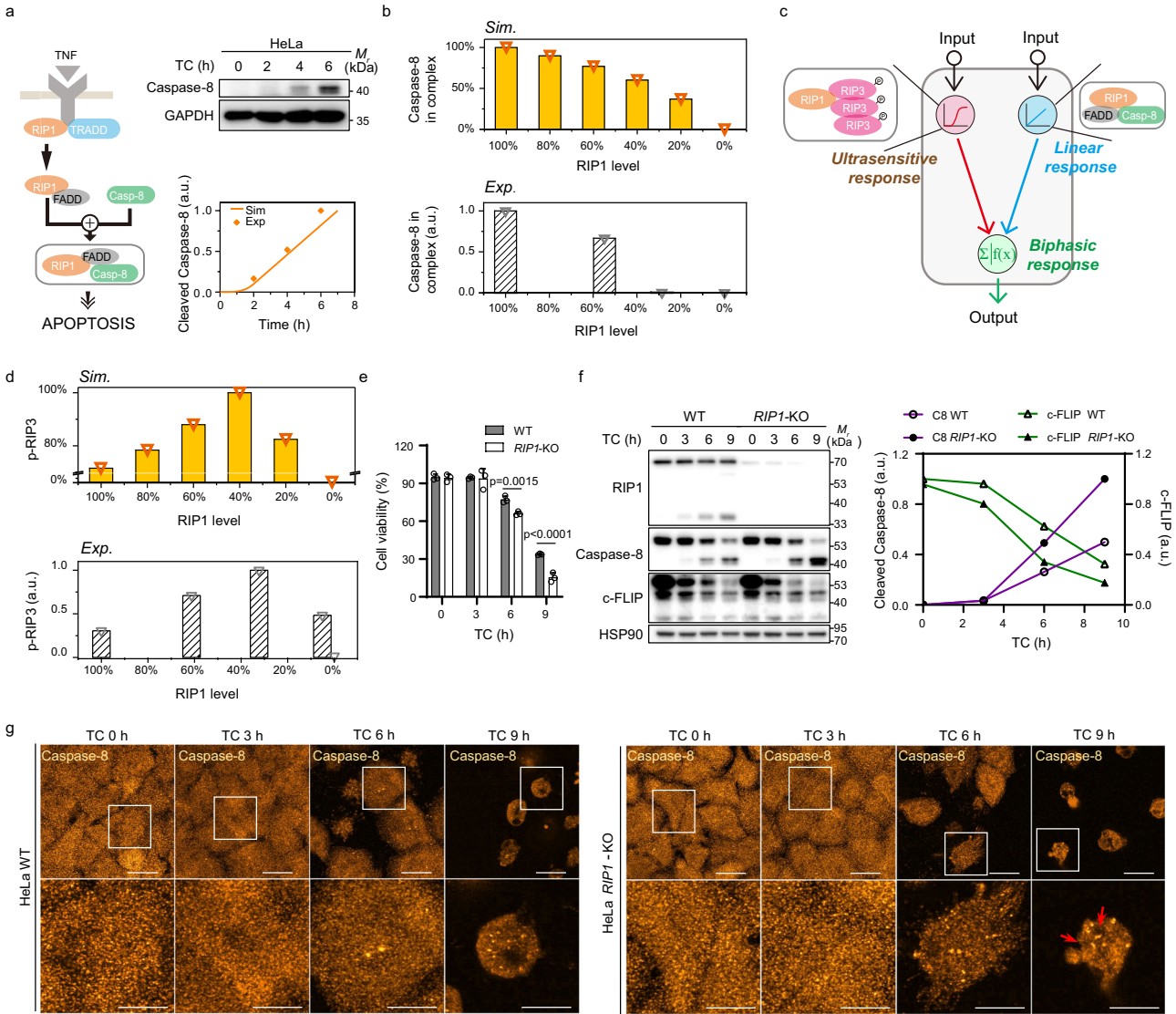

**Fig. 7 | Distinct assembly of RIP3 and caspase-8 within TNF signaling complexes. a** Schematic representation of the TNF-mediated assembly model for RIP1-FADD-caspase-8 complex in HeLa cells, validated by experimental caspase-8 activation data. Western blot analysis was performed for the indicated proteins, with the levels of cleaved caspase-8 quantified by relative band intensities. Data are representative of two independent experiments. **b** Model predictions illustrating the effect of reduced RIP1 expression on caspase-8 recruitment into RIP1 complex (upper panel), alongside corresponding experimental data using literature data[9] (lower panel). Specifically, Caspase-8 levels in complex were quantified by western blotting and RIP1 levels were controlled by shRNA-mediated knockdown. **c** Diagram depicting the biphasic response driven by threshold and linear response behaviors of RIP3 and caspase-8 within necrosomes. **d** Modeling prediction of a biphasic response induced by decreasing RIP1 levels (upper panel), with experimental

validation using literature data[9] (lower panel). Specifically, p-RIP3 levels were quantified by western blotting using a phospho-specific RIP3 antibody and RIP1 levels were controlled by shRNA-mediated knockdown. **e**–**g** WT or RIP1-deficient HeLa cells treated with TNF plus cycloheximide (TC) for indicated durations. Cell viability was assessed using a CCK-8 assay and shown as mean ± SD from three biological replicates per group (**e**). Western blot analysis was performed for the indicated proteins, with the levels of c-FLIP and cleaved caspase-8 quantified by relative band intensities. Blots are representative of four independent experiments (**f**). Confocal images showing caspase-8 distribution in TC-treated cells. Insets highlight enlarged caspase-8 structures. Images are representative of two independent experiments (**g**). *p* values were calculated using Two-Way ANOVA Tukey's multiple comparisons test. Scale bars, 30 μm (overviews) and 10 μm (insets). Source data are provided as a Source Data file.

FLIP, caspase-8 forms helical filaments, becomes strongly activated, and drives robust apoptosis. This is consistent with reports that caspase-8 is 7- to 9-fold more abundant than FADD[44,47,48]. Thus, c-FLIP functions as an effective regulatory gate, flexibly tuning apoptosis between weak and strong signaling and between linear versus threshold-like responses. More importantly, higher-order signalosome regulation is even more intricate[9,49]. Within necrosomes, linear caspase-8 activation meets nonlinear RIP3 phosphorylation, allowing RIP1 to exhibit a biphasic regulation on necroptosis. Both high and low RIP1 doses fail to induce efficient necroptosis, maintaining homeostasis across a broad range of signal intensities. This ensures accurate signal transduction, preventing excessive or inadequate responses.

Proper RIP1 levels also prevent unnecessary necroptosis and inflammation, safeguarding tissue and organ function.

Notwithstanding these insights, numerous unresolved questions remain. Currently, our model is intentionally streamlined, focusing on stoichiometry as a tractable entry point to decode necrosome assembly. In our system, the apparent ~3:1 ratio of RIP3 to RIP1 reflects an average efficiency optimum rather than a universal stoichiometric constant. While the precise position of this optimum may vary across cell types and experimental contexts due to differences in expression levels, receptor organization, or stimulation strength, our large-scale stochastic simulations indicate that this optimal window of signaling efficiency is quantitatively flexible across biological systems but

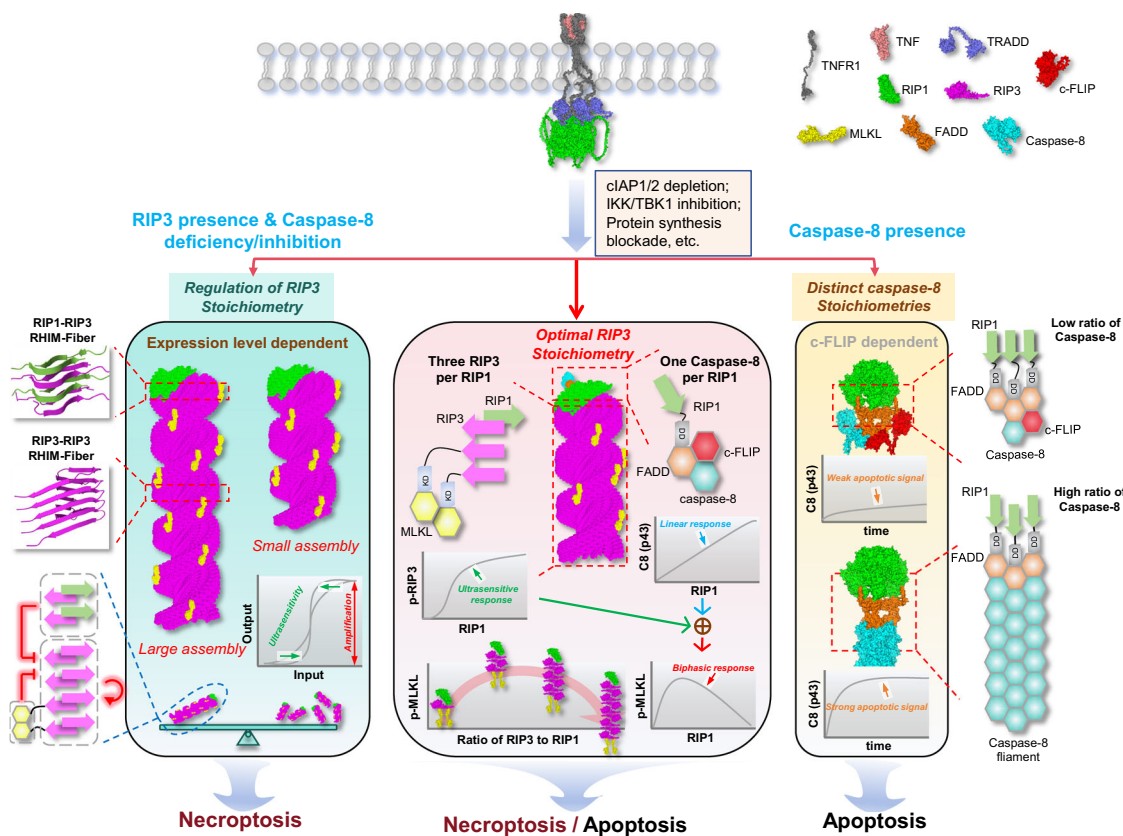

**Fig. 8 | Flexible and multi-strategic assembly of death complexes in TNF signaling.** When cells are stimulated with TNF under specific conditions (e.g., cIAP1/2 depletion or IKK/TBK1 inhibition), RIP1 dissociates from TNFR1 and assembles into cytosolic supramolecular complexes containing RIP3, MLKL, FADD, and caspase-8 (middle schematic). Within these complexes, RIP3 adopts a rod-shaped configuration of approximately three molecules per RIP1, functioning as a signaling amplifier that confers an ultrasensitive response to RIP1 levels. RIP1 and caspase-8 assemble at an approximate 1:1 ratio, generating a linear activation response, while mutual inhibition between RIP3 and caspase-8 produces biphasic regulation of necroptotic signaling by RIP1. A RIP3:RIP1 ratio of roughly 3:1 optimizes MLKL phosphorylation efficiency. In the absence or inhibition of caspase-8 (left schematic), RIP3 stoichiometry becomes dynamically regulated: larger assemblies correlate with stronger signal amplification and more pronounced threshold responses. RIP1 and MLKL act as negative regulators of RIP3 homo-assembly, whereas RIP3 promotes its own polymerization. Efficient signal transmission requires a balance between the size of individual rod-like RIP3 assemblies and the total number of necrosomes, as larger complexes form in smaller numbers. When caspase-8 is present (right schematic), RIP1-recruited caspase-8 remains non-filamentous in the presence of high c-FLIP levels, leading to weak apoptotic signaling. Upon c-FLIP depletion, caspase-8 polymerizes into helical filaments that drive robust apoptosis. Although the precise optimal RIP1:RIP3 ratio may vary across systems and contexts, our large-scale stochastic simulations (Fig. 6) recapitulate the deterministic model, indicating that these behaviors reflect conserved principles of supramolecular assembly.

mechanistically conserved. In addition, other regulatory layers including protein conformational changes, kinase activity, and post-translational modifications likely exert decisive control over complex assembly and signal propagation. Future studies should systematically incorporate these factors, ideally via targeted perturbations (e.g., Nec-1, RIP1 mutants, SMAC mimetics) combined with quantitative single-cell imaging and modeling. Besides, population-averaged biochemical approaches such as IP-MS cannot resolve individual necrosomes, but they provide essential constraints for tractable modeling. While absolute molecule counting by SMLM remains technically challenging, our new calibration strategy with MAP7 provides direct validation that antibody-based STORM can achieve quantitative accuracy under controlled conditions. As single-molecule imaging advances, it will elucidate the spatiotemporal principles of various signalosomes with absolute single-complex precision.

In summary, by integrating IP-MS, in situ quantitative STORM imaging, and computational modeling, we present a cross-scale framework that reveals key features of necrosome organization and signaling. Certainly, cell signaling is much more complex than depicted here; the necrosomes example presented merely scratches the surface of higher-order signaling. Uncovering the mechanisms and regulatory strategies of supramolecular signalosomes in more signaling is essential to establish their role as a universal paradigm in biological systems.

## Methods

### Cell culture

HeLa, HEK293T, and mouse fibroblast L929 cells were obtained from the ATCC (CCL2, CRL-3216, CCL1). *RIP1*-KO, *RIP3*-KO and *MLKL*-KO HeLa or L929 cells were generated by using CRISPR/Cas9 method as previously reported[13]. The disruption of target gene was determined by the immunoblotting of cell lysates with antibodies. All cell lines were maintained at 37 °C and 5% $CO_2$ in DMEM (Invitrogen) supplemented with 10% fetal bovine serum (Gibco), 2 mM L-glutamine, 100 U/mL penicillin, and 100 μg/mL streptomycin. Cells were routinely tested to be free of mycoplasma contamination.

### Plasmids, reagents, and antibodies

RIP1, RIP3, MLKL, and MAP7 with Flag or HA tag were cloned into the lentiviral vector pBOB (Addgene, 12337). The primer sequences flanking the RIP1 region were agagaattcggatcccaaccagacatgtcc (BamH I-RIP1) and cttccatggctcgaggttctggctgacgta (Xho I-RIP1), the RIP3 region were agag aattcggatcctcgtgcgtcaagtta (BamH I-RIP3) and cttccatggctcgagtttccc gctatgatt (Xho I-RIP3), the MLKL region were agagaattcggatccat

ggaaaatttgaag (BamH I-MLKL) and cttccatggctcgagcttagaaaaggtgga (Xho I-MLKL), the MAP7 region were gaattcggatccatggtgcgaagcgaaaca (BamH I-MAP7) and cttccatggctcgagagagagccctcaggtgg (Xho I-MAP7). Inducible expression of RIP3 was achieved by cloning the Flag-tagged RIP3 into the Tet-On expression vector (pLVX-TRE3G, Takara). Short hairpin RNA (shRNA) targeting *MLKL* was expressed from the lentiviral vector pLKO.1 (a gift from Dr. Qinxi Li at Xiamen University). All plasmids were verified by DNA sequencing. Recombinant human TNF-α (PHC3011) was from Thermo Fisher Scientific and used at 0.1–10 ng/mL. The following compounds were used: Smac mimetic SM-164 (APExBIO, A8815, 0.1 μM), pan-caspase inhibitor Z-VAD-FMK (Calbiochem, 627610, 20 μM), cycloheximide (MCE, HY-12320, 10 μg/mL) and doxycycline (Sigma, D9891, 1 μg/mL). The following antibodies were used throughout this report: anti-RIP1 (Cell Signaling, 3493, 1:150 for immunofluorescence (IF), 1:1,000 for western blotting (WB)), anti-phospho Ser166 RIP1 (Cell Signaling, 65746, 1:1,000 for WB), anti-RIP3 (Cell Signaling, 13526, 1:1,000 for WB), anti-phospho Ser227 RIP3 (Abcam, ab209384, 1:1,000 for WB), anti-MLKL (Abcam, ab184718, 1:200 for IF; 1:1,000 for WB), anti-phospho Ser358 MLKL (Abcam, ab187091, 1:1,000 for WB), anti-HA (Santa Cruz, sc-7392, 1:200 for IF, 1:1,000 for WB; Cell Signaling, 3724, 1:200 for IF, 1:1,000 for WB; ABclonal, AE008, 1:200 for IF), anti-Flag (Abmart, M20008L, 1:1000 for IF, 1:5,000 for WB; Biolegend, 637301, 1:200 for IF), anti-GAPDH (Proteintech, 60004-1-Ig, 1:5,000 for WB), goat anti-rabbit, anti-mouse and anti-rat secondary antibodies conjugated to Alexa Fluor 488, 568 or 647 (Thermo Fisher Scientific, A11034, A11004 or A21247, 1:1,000 for IF) and goat anti-rabbit, anti-mouse and anti-rat secondary antibodies conjugated to CF 488 A, CF568 or CF 647 (Biotium, 20015, 20800, 20801, 20808 or 20809, 1:500 for IF). Detailed information for reagents and antibodies used in this study is provided in Supplementary Data 1 and 2, respectively.

### Transfection and lentiviral infection

Cell transfection was performed using Turbofect reagent according to the manufacturer's instructions (Thermo Fisher Scientific, R0531). For lentivirus production, HEK293T cells were transfected with lentiviral vectors and virus packaging plasmids by the calcium phosphate precipitation method. The virus-containing medium was collected 36–48 h later and added to HeLa cells with 10 μg/mL polybrene as indicated. The infectious medium was replaced with fresh medium 12 h later, and the cells were kept in culture until analysis. To reduce the MLKL expression, HeLa cells were stably infected with the lentiviral vector pLKO.1 containing *MLKL*-targeting shRNA.

### Cell viability assay

Cell viability was assessed using a CCK-8 assay. Briefly, after treatments, cells were incubated with CCK-8 reagent (Apexbio, K1018) for 1 h at 37 °C, and the absorbance was measured at 450 nm. Viability was normalized to vehicle-treated controls. To specifically quantify necroptosis, cells were stained with Hoechst 33342 (1 μg/mL) and propidium iodide (PI, 5 μg/mL) for 20 min at 37 °C. Cells were immediately imaged using fluorescence microscopy. The total cell count was determined from Hoechst-positive nuclei, and necroptotic cells were defined as PI-positive. Image analysis was performed using Fiji/ImageJ software.

### Immunoprecipitation and immunoblotting

Cell pellets were obtained in ice-cold PBS and re-suspended in lysis buffer (20 mM Tris-HCl, pH 7.5, 150 mM NaCl, 1 mM Na$_2$EDTA, 1 mM EGTA, 1% Triton X-100, 2.5 mM sodium pyrophosphate, 1 mM β-glycerophosphate, 1 mM Na$_3$VO$_4$) plus protease inhibitor cocktail (MCE, HY-K0010). The re-suspended cell pellets were sonicated and centrifuged at 20,000 g for 30 min at 4 °C. The supernatant was immunoprecipitated with Flag-M2 affinity resin (Sigma, A2220) at 4 °C for 3 h or overnight. After the immunoprecipitation, the beads were washed four times in lysis buffer, and the immunoprecipitates were subsequently eluted with SDS sample buffer (50 mM Tris pH 6.8, 2% SDS and 10% glycerol). To analyze the oligomeric status of RIP1, RIP3, and MLKL, cells were directly treated with SDS sample buffer without β-mercaptoethanol and resolved by 4–12% gradient NuPAGE (Invitrogen, NP0336BOX). After electrophoresis, protein samples were transferred onto PVDF membranes. Membranes were blocked with 3% BSA for 1 h and incubated with primary antibodies overnight at 4 °C, followed by incubation with horseradish peroxidase-conjugated secondary antibodies. Proteins were visualized using the blot and gel imager (AI680, GE Healthcare). GAPDH was used as a loading control.

### Confocal microscopy

For immunostaining, cells were grown on #1.5 coverslips (NEST, 801008) coated with poly-l-lysine (Sangon, E607015-0006). Cells were fixed with freshly prepared 3.7% formaldehyde for 15 min at room temperature and then permeabilized with 0.25% Triton X-100 in PBS. After blocking with 3% BSA for 1 h, samples were stained with primary antibodies overnight at 4 °C, washed three times with PBS, and incubated with secondary antibodies for 1 h at room temperature. Slides were mounted with antifade reagent (Invitrogen, P36934). For multichannel imaging, extensive controls were performed to ensure that there was no non-specific staining or crosstalk between channels. These controls included (1) use of cells lacking one of the proteins of interest and/or (2) staining without one of the primary or secondary antibodies. All images were captured and processed with identical settings in the LSM 980 laser scanning confocal microscope (ZEN 2019, Zeiss) with a 63×/1.40 numerical aperture (NA) oil objective. Duplicate cultures were examined and similar results were obtained in at least two independent experiments.

### STORM imaging

STORM imaging was performed on an N-STORM microscope (Ti-E, Nikon Instruments). Briefly, the N-STORM system uses an Agilent MLC-400B laser launch with a red diode laser (647 nm, 300 mW; MPBC), a green solid-state laser (561 nm, 150 mW; Coherent), a blue solid-state laser (488 nm, 200 mW; Coherent), a violet diode laser (405 nm, 100 mW; Coherent) and a 100×NA 1.49 oil immersion objective. Emission fluorescence was separated with appropriate filters (FF02-520/28-25, FF01-586/20-25×3.5, and FF01-692/40-25; Semrock) and detected with a back-illuminated EMCCD camera (iXon DU897, Andor). Cells were pre-cultured on eight-well chamber coverslips (Thermo Fisher Scientific, 155409). Sample preparation for STORM imaging is similar to that for confocal microscopy. After labeling with appropriate fluorescence-conjugated antibodies, samples were immersed in GLOX imaging buffer containing 50 mM Tris (pH 8.0), 10 mM NaCl, 0.5 mg/mL glucose oxidase (Sigma, G2133), 40 μg/mL catalase (Sigma, C40), 10% (wt/vol) glucose, and 143 mM β-mercaptoethanol. For single-colour STORM imaging of CF 647, cells were exposed to a 647 nm laser at a power density of 2 kW cm$^{-2}$. For two-colour STORM imaging, the CF 647 channel was acquired first, followed by the CF 568 channel at a laser power density of 1-3 kW cm$^{-2}$. For three-colour STORM imaging, the CF 647 channel was acquired first, followed by the CF 568 channel, and finally the CF 488 A channel at a laser power density of 1–3 kW cm$^{-2}$. Quantitative STORM imaging was performed using a dye combination of AF 647, CF 568, and AF 488. The imaging procedure was consistent across dyes, with the exception that AF 488 was excited in a specialized DABCO imaging buffer containing 65 mM DABCO (BBI, A601472-0100), 30 mM DTT (BBI, B645939-001), 30 mM sodium sulfite (BBI, A100628-0500), prepared as reported previously[50,51]. For each channel, 10,000 to 20,000 frames were typically acquired at 70 fps. Final images were reconstructed using the SMAP platform following the provided guidelines.

### STORM image analysis

For STORM image analysis, we used the SMAP software for data processing. As described previously[52], the overall workflow includes steps

such as Import, Fitting, Post-processing, and Rendering. To analyze molecular clusters, we employed the POCA software, which performs clustering based on Voronoi tessellation[53]. Localization data exported from SMAP was imported into POCA to extract parameters such as the area of each cluster and the total number of localizations. Since POCA does not provide direct measurements of cluster length, we developed a custom Python script to calculate the maximum Feret diameter using the localization and clustering information. For multi-color STORM datasets, channel separation was performed using frame-sequence metadata, followed by alignment and colocalization analysis using custom scripts and Microsoft Excel. Importantly, quantification of molecular stoichiometry ratios was based on localization counts rather than raw signal overlap, ensuring that the analysis reflects objective molecular metrics and reduces susceptibility to rendering or visualization artifacts.

## Mass spectrometry analysis

Flag-RIP3- or Flag-MLKL-reconstituted L929 cells were treated with 10 ng/mL TNF for 0, 120, 210, 330, and 420 min for isolation of RIP3 or MLKL complexes. Ten 15-cm dish cells were collected for each time point experiment, and three biological replicates were carried out. After TNF treatment, cells were immediately washed twice with ice-cold PBS and harvested by scraping. The harvested cells were washed with PBS and lysed for 30 min on ice in HBS lysis buffer (12.5 mM HEPES, 150 mM NaCl, 1% Nonidet P-40, pH 7.5) with the protease inhibitor cocktail. Cell lysates were then spun down at 20,000 g for 30 min. The soluble fraction was collected and immunoprecipitated overnight with anti-Flag M2 antibody-conjugated agarose at 4 °C. Resins containing protein complexes were washed three times with HBS lysis buffer. Proteins were subsequently eluted twice with 0.2 mg/mL of 3× flag peptides in HBS lysis buffer for 30 min each time, and elution was pooled for a final volume of 300 μL. Proteins in the elution were precipitated with 20% trichloroacetic acid (TCA), and the pellet was washed two times with 1 mL cold acetone and dried in speedVac. TCA-precipitated proteins were re-suspended in 50 μL 1% SDC (sodium deoxycholate) in 10 mM TCEP (Tris(2-carboxyethyl)phosphine hydrochloride), 40 mM CAA (chloroacetamide), 100 mM Tris-HCl pH 8.5. 1% SDC was then diluted to 0.5% SDC with water, and trypsin (Sigma) was added at a ratio of 1:50 (trypsin: protein). The digestion was performed at 37 °C overnight. 1% TFA was added to stop the reactions, followed by centrifugation at 20,000 g for 10 min. The supernatants were subsequently transferred to C18 STAGEtips for desalting. Tryptic peptides were eluted with 70% acetonitrile/1% formic acid and dried in speedvac. Peptides were dissolved in 0.1% formic acid and analyzed by mass spectrometry in DDA and SWATH mode. MS analysis was performed on a TripleTOF 5600 (Sciex) mass spectrometry coupled to NanoLC Ultra 2D Plus (Eksigent) HPLC system. Peptides were first bound to a 5 mm × 500 μm trap column packed with Zorbax C18 5 μm 200 Å resin using 0.1% (v/v) formic acid/2% acetonitrile in H$_2$O at 10 μL/min for 5 min, and then separated from 2% to 35% buffer B (buffer A: 0.1% (v/v) formic acid, 5% DMSO in H$_2$O, buffer B: 0.1% (v/v) formic acid, 5% DMSO in acetonitrile) on a 35 cm × 75 μm in-house pulled emitter-integrated column packed with Magic C18 AQ 3 μm 200 Å resin. Predicted library generation and targeted analysis of DIA data were performed by DIA-NN (v1.8.1)[54]. "FASTA digest for library-free search" and "Deep learning-based spectra, RTs and IMs predication" were enabled. Output was filtered at 1% FDR to get significantly identified spectra and peptide lists.

## Protein structure modeling

The published structures of RIP1–RIP3 hetero-amyloid core (Protein Data Bank (PDB) 5V7Z), RIP3 homo-amyloid core (PDB 7DAC), MLKL–RIP3 complex (PDB 7MON), Death effector domains (DED) complex of FADD&CASP8 (PDB 8YD8), TNF (PDB 1TNF), RIP1 kinase domain (PDB 4NEU), RIP1 death domain (PDB 6AC5) and RIP3 kinase domain (PDB 4M66) were used in structural modelling and generating figures. The hetero-amyloid structure of RIP1–RIP3 in the published

paper[7] showed a slight twist with an angle of 5.4° ± 0.5° for RIP3 layers and 5.3° ± 0.3° for RIP1 layers, respectively. To generate the proposed RIP1–RIP3 model, the initial helical rise and twist angle of structure containing two layers of RIP3 and two layers of RIP1 (derived from PDB 5v7z) were set to 19.5 Å and 10.6°, respectively. The structures of death domains (DD) complex of FADD&RIP1, phosphorylated MLKL tetramer, and other full-length proteins and/or their complexes were generated by AlphaFold Server[55]. PyMOL (The PyMOL Molecular Graphics System, Version 3.0, Schrödinger, LLC.) was used to generate all the figures of structures.

## Modeling principle

We established a quantitative dynamic model of TNF signaling-induced cell death complex assembly based on the law of mass action. The modeling process begins with identifying the fundamental biochemical reactions involved in the system, classifying them by reaction type, followed by translating each reaction into a rate law, and finally formulating a system of ordinary differential equations (ODEs) that captures the temporal evolution of all molecular species.

**Reaction types and rate laws.** Each reaction in the signaling cascade falls into one of the following fundamental categories:

- Association (binding): e.g., $A + B \xrightarrow{k_{on}} C$ Rate: $r_{assoc} = k_{on}[A][B]$
- Dissociation (unbinding): e.g., $C \xrightarrow{k_{off}} A + B$ Rate: $r_{dissoc} = k_{off}[C]$
- Catalytic activation (e.g., phosphorylation, cleavage): e.g., $A \xrightarrow{k_{cat}} A^*$ Rate: $r_{cat} = k_{cat}[A]$
- Degradation or inactivation: e.g., $A^* \xrightarrow{k_{deg}} \varnothing$ Rate: $r_{deg} = k_{deg}[A^*]$
- Multimerization or oligomer assembly (e.g., higher-order RIP3 complex): e.g., $nA \xrightarrow{k_n} A_n$ Rate: $r_{oligo} = k_n[A]^n$

Each reaction is assigned a rate constant k and contributes to the change in concentration of the involved species.

## Construction of the ODE system

For each molecular species $X_i$, its concentration over time is described by:

$$\frac{d[X_i]}{dt} = \sum_{j=1}^{R} \nu_{ij} \cdot r_j(X, \theta) \tag{1}$$

Where:

- $[X_i]$: concentration of species i;
- $r_j$: rate of reaction j;
- $\nu_{ij}$: stoichiometric coefficient of species $X_i$ in reaction j;
- $\theta$: vector of kinetic parameters.

This leads to a system of ODEs such as:

$$\begin{cases} \frac{d[A]}{dt} = -k_{on}[A][B] + k_{off}[C] \\ \frac{d[B]}{dt} = -k_{on}[A][B] + k_{off}[C] \\ \frac{d[C]}{dt} = k_{on}[A][B] - k_{off}[C] - k_{cat}[C] \\ \dots \end{cases} \tag{2}$$

Each equation reflects the net effect of all reactions in which the species participates.

## Basic components assembly modeling

To capture the hierarchical and sequential assembly of signaling complexes, we developed a mass-action-based kinetic model describing two fundamental interaction modules: a two-component system (A-B) and a three-component extension (A-B-C). These modules abstract the core steps of scaffold recruitment, binding, and downstream activation.

1. Two-Component Assembly (A + B)

In the first stage, protein A binds to protein B, forming an intermediate complex $A_mB_n$, which then facilitates the activation of B. The assembly and activation steps are modeled as:

$$mA + nB \underset{k_{-1}}{\overset{k_1}{\rightleftharpoons}} A_mB_n \overset{\kappa_1}{\rightarrow} mA + nB^*$$

- $A_mB_n$: the assembled but inactive complex formed by m units of A and n units of B.
- $B^*$: the activated form of protein B.
- $k_1, k_{-1}$ : binding and unbinding rates between A and B.
- $\kappa_1$: rate constant for irreversible activation of B.

2. Three-Component Assembly (A + B + C)

In the second stage, activated B ($B^*$) recruits protein C and promotes its activation, forming a ternary complex:

$$nB^* + hC \underset{k_{-2}}{\overset{k_2}{\rightleftharpoons}} B_n^*C_h \overset{\kappa_2}{\longrightarrow} nB^* + hC^*$$

- $B_n^*C_h$: the intermediate complex consisting of n activated B units and h C units.
- $C^*$: the activated form of protein C.
- $k_2, k_{-2}$: binding and unbinding rates between $B^*$ and C.
- $\kappa_2$: irreversible activation rate of C.

3. Model Variables and Interpretation

The stoichiometric coefficients m,n,h reflect the molecular configurations or oligomeric states of A, B, and C within their respective complexes. These variables allow the model to capture non-1:1 stoichiometries, which are common in supramolecular signaling assemblies. The complete set of ordinary differential equations derived from these reactions describes the time evolution of each species and complex. For example:

$$\frac{d[A_mB_n]}{dt} = k_1[A]^m[B]^n - k_{-1}[A_mB_n] - \kappa_1[A_mB_n] \tag{3}$$

$$\frac{d[B_n^*C_h]}{dt} = k_2[B^*]^n[C]^h - k_{-2}[B_n^*C_h] - \kappa_2[B_n^*C_h] \tag{4}$$

$$\frac{d[B^*]}{dt} = \kappa_1[A_mB_n] - k_2[B^*]^n[C]^h + k_{-2}[B_n^*C_h] \tag{5}$$

This modeling framework captures the essential sequential logic of multi-protein assembly: scaffold → recruitment → activation → downstream assembly. A detailed list of equations and parameter values is provided in Supplementary Table 1.

## Determined necrosome assembly modeling

To systematically explore the emergent and intricate features of TNF-induced necrosome formation, we developed a mass-action-based kinetic model that incorporates the core molecular events of protein recruitment, binding, catalytic activation, and mutual inhibition. The model is formulated as a set of coupled ordinary differential equations (ODEs), following the principles described in the Modeling Principle section. It captures the time-dependent evolution of individual molecular species involved in necrosome assembly and downstream necroptosis signaling. The model includes the following key modules: 1). TNF-induced recruitment of RIP1 and TRADD to complex I; 2). Dissociation of RIP1 from complex I and its incorporation into the necrosome; 3). Assembly of necrosome via RIP1-RIP3 binding and phosphorylation; 4). Activation of MLKL by phosphorylated RIP3; 5).

Crosstalk and mutual inhibition between RIP3 and caspase-8 (C8); 6). TRADD-mediated activation of C8. This section focuses on modeling the core necrosome assembly cascade, which includes the stepwise formation of RIP1-RIP3-MLKL complexes and their activation events.

## Reactions and stoichiometry

Mass spectrometry-based quantitative analysis revealed the stoichiometric ratios of key components in the necrosome as RIP1:RIP3:MLKL = 1:3:2. The major reactions modeled are:

- Reaction R1: Binding of one active RIP1 molecule with three RIP3 molecules to form a ternary complex

$$RIP1_{active} + 3RIP3 \underset{k_{-1}}{\overset{k_1}{\rightleftharpoons}} RIP1_{active}\text{-}RIP3_3$$

- Reaction R2: Catalytic phosphorylation of RIP3 by RIP1 within the RIP1-RIP3 complex

$$RIP1_{active}\text{-}RIP3_3 \overset{\kappa_2}{\rightarrow} RIP1_{active} + 3RIP3_{pho}$$

- Reaction R3: Binding of three phosphorylated RIP3 molecules to two MLKL molecules

$$3RIP3_{pho} + 2MLKL \underset{k_{-3}}{\overset{k_3}{\rightleftharpoons}} RIP3_{pho_3}\text{-}MLKL_2$$

- Reaction R4: Phosphorylation and activation of MLKL

$$RIP3_{pho_3}\text{-}MLKL_2 \overset{\kappa_4}{\rightarrow} 3RIP3_{pho} + 2MLKL_{pho}$$

Where:
- $RIP1_{active}$: the activated form of RIP1 released from complex I
- $RIP3$, $RIP3_{pho}$: unphosphorylated and phosphorylated forms of RIP3
- $MLKL$, $MLKL_{pho}$: inactive and activated MLKL
- $k_1$, $k_{-1}$, $\kappa_2$, $k_3$, $k_{-3}$, $\kappa_4$: forward, reverse, and catalytic rate constants for each step

## Differential equations

The dynamics of each species is governed by the following ODEs:
- For the RIP1-RIP3 ternary complex:

$$\frac{d[RIP1_{active}\text{-}RIP3_3]}{dt} = k_1[RIP1_{active}][RIP3]^3 \\ - k_{-1}[RIP1_{active}\text{-}RIP3_3] - \kappa_2[RIP1_{active}\text{-}RIP3_3] \tag{6}$$

- For free active RIP1:

$$\frac{d[RIP1_{active}]}{dt} = -k_1[RIP1_{active}][RIP3]^3 + k_{-1}[RIP1_{active}\text{-}RIP3_3] \\ + \kappa_2[RIP1_{active}\text{-}RIP3_3] \tag{7}$$

- For phosphorylated RIP3:

$$\frac{d[RIP3_{pho}]}{dt} = 3\kappa_2[RIP1_{active}\text{-}RIP3_3] - 3k_3[RIP3_{pho}]^3[MLKL]^2 \\ + 3k_{-3}[RIP3_{pho_3}\text{-}MLKL_2] + 3\kappa_4[RIP3_{pho_3}\text{-}MLKL_2] \tag{8}$$

- For the RIP3-MLKL complex:

$$\frac{d[RIP3_{pho_3}\_MLKL_2]}{dt}$$
$$= k_3[RIP3_{pho}]^3[MLKL]^2 - k_{-3}[RIP3_{pho_3}\_MLKL_2] \qquad (9)$$
$$- \kappa_4[RIP3_{pho_3}\_MLKL_2]$$

- For activated MLKL:

$$\frac{d[MLKL_{pho}]}{dt} = 2\kappa_4[RIP3_{pho_3}\_MLKL_2] \qquad (10)$$

The equations presented here focus on modeling the core necrosome assembly process and the key interactions between RIP1, RIP3, and MLKL. However, due to the multi-step, highly interconnected nature of necrosome formation, the actual system is more complex and involves additional molecular interactions. These include further crosstalk between RIP3, caspase-8, TRADD, and other components of the necrosome signaling network. As such, the full model comprises 23 reaction equations and 39 ODEs in total, capturing a broader range of protein-protein interactions, feedback loops, and regulatory mechanisms that govern necrosome assembly and its downstream effects. The complete set of equations and parameter values is provided in Supplementary Table 2.

## Parameters values and initial amounts selection

**Kinetic parameters.** The selection of kinetic parameters for our model was based on experimental data, and these parameters were optimized using R-square fitting. The parameters include rate constants for various reactions such as binding/unbinding rates, catalytic activation rates, and stoichiometric coefficients. These values were chosen to be within biologically relevant ranges, with a focus on fitting the experimental data derived from time-course measurements of protein concentrations involved in necrosome assembly. The fitting procedure aimed to minimize the discrepancy between experimental and model-predicted values using the coefficient of determination (R-square), which quantifies the goodness of fit. The R-square value is calculated as:

$$R^2 = 1 - \frac{\sum_{i=1}^{N}\left([X_i^{exp}](t_i) - [X_i^{sim}](t_i;\theta)\right)^2}{\sum_{i=1}^{N}\left([X_i^{exp}](t_i) - \overline{[X_i^{exp}]}\right)^2} \qquad (11)$$

Where:
- $[X_i^{exp}](t_i)$ are the experimental concentration measurements at time $t_i$,
- $[X_i^{sim}](t_i;\theta)$ are the model-predicted concentrations using the parameter set $\theta$,
- $\overline{[X_i^{exp}]}$ is the mean value of the experimental concentrations,
- $N$ is the number of data points used in the fitting.

The parameters were iteratively adjusted to maximize $R^2$, ensuring that the model predictions closely matched the experimental data, thereby reflecting the dynamic behavior of the signaling network.

**Initial amounts of species.** The initial concentrations of the species involved in necrosome assembly, such as RIP1, RIP3, and MLKL, were selected based on previously published studies and experimental proteomics data. These initial amounts were used to set the starting conditions for the ODE simulations. For species where direct initial concentration data were unavailable, reasonable estimates were made based on the literature, ensuring they remained within physiological ranges. The initial conditions were set for the concentrations of RIP1,

RIP3, and MLKL at t = 0, which were assumed to be in their inactive forms prior to TNF stimulation. These values were then used as the baseline for fitting the model.

## Stochastic simulation of necrosome assembly dynamics

To account for intrinsic cell-to-cell variability, we performed large-scale stochastic simulations of TNF-induced necrosome assembly by systematically perturbing kinetic parameters and structural features within biologically plausible ranges.

**Parameter space and latin Hypercube sampling.** We defined a np-dimensional parameter space $\theta = \{\theta_1, \theta_2,..., \theta_{np}\}$, with each $\theta_i \in [\theta_{i,min}, \theta_{i,max}]$ sampled in log-scale according to prior physiological knowledge:

$$\theta_i \sim log\, U(log\theta_{i,min}, log\theta_{i,max})\, for\, i = 1, 2, \ldots, n_p \qquad (12)$$

The sampling ranges were defined as:

- Kinetic constants (k):           $\theta \in [0.1, 10]$
- Catalytic or cooperativity terms (j):   $\theta \in [10^{-3}, 10^2]$
- Degradation rates (d):           $\theta \in [10^{-2}, 1]$
- RIP3 filament size (n):           $n \in \{1, 2,..., 8\}$ (discrete uniform)

Using Latin Hypercube Sampling (LHS), we generated N = 10,000 unique samples:

$$\{\theta^{(1)}, \theta^{(2)}, \ldots, \theta^{(10000)}\}$$

Each sample $\theta^{\wedge}(i)$ corresponds to a full set of reaction rates and initial conditions for a TNF-induced signaling network model.

## Simulation of dynamics and efficiency computation

For each parameter set $\theta^{(i)}$, the system of ODEs describing TNF-driven necrosome assembly was numerically integrated until a steady state or fixed time point $t = T$ was reached. The core readout of the signaling response was the MLKL phosphorylation efficiency $\eta^{(i)}$, defined as:

$$\eta^{(i)} = \frac{[pMLKL]^{(i)}(T)}{[MLKL_{total}]^{(i)}} \qquad (13)$$

where pMLKL is the steady-state concentration of phosphorylated MLKL under parameter set $i$.

To explore how oligomeric configurations affect signaling, we introduced a variable $n_{RIP3}$ representing the stochastic RIP3 chain length. For each $\theta^{(i)}$, $\eta$ was evaluated at different $n_{RIP3}$ values:

$$\eta^{(i)}(n_{RIP3}) = f\left(\theta^{(i)}, n_{RIP3}\right) \qquad (14)$$

This enabled generation of a phase map over the plane $(n_{RIP3}, \eta)$, as illustrated in Fig. 6a.

**Selection of high-efficiency models and statistical analysis.** From the 10,000 simulated models, we extracted the top $\alpha = 5\%$ models that achieved the highest MLKL phosphorylation efficiency:

$$M_{top} = \left\{\theta^{(i)} : \eta^{(i)} \geq \eta_{95\%}\right\} \qquad (15)$$

where $\eta_{95\%}$ is the 95th percentile threshold. For each model in $M_{top}$, the corresponding optimal RIP3 chain length $n^*$ was identified:

$$n^* = \underset{n_{RIP3}}{argmax}\, \eta^{(i)}(n_{RIP3}) \qquad (16)$$

These values were then statistically analyzed to determine whether specific RIP3 configurations conferred robust and optimal

signaling. Additionally, to explore the influence of TNF stimulus and expression levels of RIP3 or MLKL on necrosome topology, we repeated the above simulations under perturbed conditions (e.g., *[TNF]*, *[RIP3]*, *[MLKL]*) and analyzed shifts in the distribution of $n^*$.

## Computational implementation

All simulations were implemented in Python using the SciPy.integrate.odeint solver for ODEs. Sampling was performed using pyDOE's Latin Hypercube Sampling routines. Each simulation ran independently in parallel. The entire code base is available at: https://github.com/XMU-Xu/RIP3_assembly.git.

## Statistics and reproducibility

For all datasets subjected to comparative analysis, the appropriate statistical method was selected based on the assessment of variance homogeneity as determined by GraphPad Prism. Depending on whether the standard deviation (SD) was equal, either parametric or nonparametric tests were applied accordingly within the software. All statistical tests were two-tailed. Where error bars are shown, they represent one standard deviation (±SD). $p < 0.05$ denotes statistical significance. All analyses were carried out using GraphPad Prism version 8.3.0(538) (released October 15, 2019).

## Reporting summary

Further information on research design is available in the Nature Portfolio Reporting Summary linked to this article.

## Data availability

All data supporting the findings of this study are available within the paper and its Supplementary Information. The proteomic raw data have been deposited to the ProteomeXchange Consortium via the PRIDE repository with the dataset identifier PXD067729. Source data are provided with this paper.

## Code availability

The mathematical modeling code is publicly available on GitHub (https://github.com/XMU-Xu/RIP3_assembly.git; DOI: https://doi.org/10.5281/zenodo.17388397). The custom Python script for calculating the maximum Feret diameter of POCA-defined clusters is available at https://github.com/xchenxmu/POCA_caculation_MAX_Feret_diameter.git.

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

## Acknowledgements

We thank Dr. Q. Liu for expert assistance in fluorescence microscopy and Nikon Precision for technical assistance with super-resolution imaging. We are grateful to Dr. Yandong Huang and Mr. Zhitao Cai (Jimei University) for their insightful suggestions and generous help with the protein dynamics simulations. This work was supported by the National Natural Science Foundation of China (31871386, 32070736, and 32370803 to X.C.; 12474198 to X.L.; 12090052, U24A2014, and STI2030-Major Projects 2021ZD0201900 to J.S.; 12404233 to F.X.; 32200598 to R.Z.), the Natural Science Foundation of Fujian Province of China (2023J06003 to X.C.; 2023J05002 to X.L.), and the Fundamental Research Funds for the Central Universities (20720210114 to X.C.; 20720230017 to X.L.).

## Author contributions

X.L. and X.C. conceptualized the study. Y.C., Y.Z., and Y.K. carried out the experimental work. R.Z. performed molecular modeling. C.L. (Chengjie Lan) contributed to the analysis of super-resolution images. X.L. and F.X. developed the models and performed simulations. C.Z. performed proteomics analysis. C.L. (Cheng Lin), Z.L., H.Q., Y.H., G.S., and Y.X. helped to analyze data. X.L. and X.C. designed experiments, interpreted data, and wrote the manuscript. X.L., J.S., and X.C. conceived and supervised the study.

## Competing interests

The authors declare no competing interests.
