## [Transparent Peer Review file · Nature Communications]

Decoding necrosome assembly: harmonizing signal amplification and attenuation through optimal RIP3 stoichiometry

Corresponding Author: Professor Xin Chen

Version 0:

Reviewer comments:

Reviewer #1

(Remarks to the Author)

This study identifies the RIP1-bound RIP3 homotrimer as the minimal functional unit of the necrosome, enabling signal amplification while preserving threshold sensitivity. A notable finding is that excessive RIP3 oligomerization attenuates necroptotic signaling, revealing a size-control mechanism dynamically regulated by RIP1 (restrictive) and RIP3 (expansive), with MLKL unexpectedly limiting assembly. In contrast, caspase-8 recruitment by RIP1 follows a linear, non-oligomeric pattern. These distinct assembly modes create a biphasic necroptotic response, illustrating how signalosomes balance configuration and stoichiometry for efficient signaling—providing new insights for therapeutic and synthetic biology applications. The super-resolution microscopy data offer valuable advances in understanding signalosome dynamics, though additional causal experiments would strengthen the conclusions, particularly regarding RIPK3 oligomer size control in signal amplification/limitation.

Specific Comments:

1. MLKL's inhibitory role:

most interestingly, the authors' observation that p-MLKL negatively correlates with RIPK3 higher-order assembly aligns with prior structural work (PMID: 24095729) showing MLKL-bound RIP3 adopts an inactive conformation. This suggests a feedback mechanism where MLKL, as a substrate, restricts RIPK3 kinase activity by limiting RIPK3 fibrilization. To solidify this claim, the authors should provide direct experimental evidence (e.g., co-IP assays) demonstrating MLKL's physical restriction of RIPK3 oligomerization, corroborating their correlation data (Figure 5).

2. RIPK3 oligomerization and signaling:

While the data show that RIPK3 expression positively correlates with fibril length (consistent with pro-necroptotic activity), the conclusion that "excessive RIP3 homo-assembly attenuates signaling" appears contradictory without MLKL context. To resolve this, I suggest the authors to perform time-course analyses tracking RIPK3 fibril length and cell death in MLKL-proficient vs. -deficient systems. A predicted outcome:

o Without MLKL: RIPK3 fibrils grow indefinitely without cell death.

o With MLKL: Fibril elongation halts (or reverses) as p-MLKL accumulates, linking fibril size control to signal transduction. This would unify the observations, emphasizing that RIPK3 oligomerization's functional output depends on MLKL-mediated feedback.

3. Minor correction:

o Figure 3G: "oligermer" → "oligomer"

(Remarks on code availability)

Reviewer #2

(Remarks to the Author)

The manuscript by Li et al. reports the experimental characterization of necrosome assembly by using super-resolution microscopy (STORM) and mass spectrometry (MS). The necrosome is an important signaling complex in necrotic cell death that has broad implications in various human diseases. The results complement the existing structural data from NMR with

in vivo STORM and MS data, thus providing important insights into the function and assembly of necrosome. The manuscript may benefit from addressing the following concerns before formal publication:

- (1) The abstract may need one or two sentences to describe the importance of necrosome assembly. The introduction may benefit from a brief introduction of the necrosome function and its implication in certain human diseases.
- (2) Can the authors discuss on the structural basis and rationality of the phase diagram of MLKL activation shown in Fig. 5a-c, perhaps based on the previously published NMR structure data or AlphaFold2/3-predicted models?
- (3) Can the authors discuss on the structural basis and rationality of their findings on the stoichiometry of necrosome assembly? How are their results reconciled with the previously published data?
- (4) There are numerous places that authors used exaggerating or hyperbolic words or phrases, which are unnecessary for scholarly presentation. For example, on page 10, the 2nd paragraph starts with "... overwhelmingly underscores ...". It is confusing why the authors used "overwhelmingly" here. On page 2, the last sentence reads "our findings challenge the conventional view of ...". But the results do not appear to really "challenge" but instead just "improve" the existent understanding of necrosome assembly and associated signal transduction. The use of "challenge" in several places is not well supported by the work itself, and make it read more like overselling the paper by the authors.
- (5) There are no sufficient details provided for their computational modeling. The Methods section may benefit from a revision that provides necessary details required for reproduction of their modeling results.

(Remarks on code availability)

Reviewer #3

(Remarks to the Author)

This work by Li et al. provides an interesting angle for understanding how necrosome assembly works (e.g., its output as p-MLKL and the resulting cell death). This is a very unique way to analyze the necroptosis mechanism; it should definitely be published. However, some issues require clarification and reconsideration before publication. Moreover, since some of the technical details are not fully clear in the current version, I may not agree with all the analytical approaches the authors have used (but more clarification will help address my concerns; I look forward to a revised manuscript). Also, I am not an expert in mathematical modeling. Thus, all my inputs and concerns are from a cell biology and microscopy angle.

1. Note that a majority of the audience for this manuscript are still cell biologists; thus, this version of the manuscript are still hard to us to digest. A more detailed navigation would be needed. And for the figure legend and figure description in the main text, it is too brief, lacking details about how the authors reach to these results. And sometimes, it is not clear whether the data are based on model calculations or experimental data. I am happy to read a more updated version with EXTENSIVE re-writing.

Several examples,

The authors keep writing "RIP3 trimer per RIP1" etc., which really reflects the 3:1 ratio of RIP3, right? I think words like "trimer", "dimer" are very confusing, as in one necrosome, it contains more than 3 RIP3 molecules.

Fig. 3g how did cell death measured? How did RIP3 dimer control? Fig. 3h, how did p-RIP3 measured? by WB? any representative images? How was RIP1 level controlled? (just citing the previous publication was not sufficient for the audience to understand the experimental details, and don't put everything into the method, as it is hard to locate the useful information)

Fig. 4g, is this simulation from a model or real experimental data? how about 4h? if it is not experimental data, then experimental data verification is a must.

Fig. 5 g and i, how were the MLKL and RIPK3 levels determined? The author suggested by STORM, which means STORM the whole cells or just the necrosomes? What does "amount of necrosome per cell" mean? "necrosome counts or intensity?" etc.

Without sufficient technical details about how the analysis was conducted, my judgment on these data remains very preliminary.

2. Based on the quantitative approach STORM microscopy, I would ask the authors to clarify

Why choose to measure the length of RIP1/3, MLKL etc? If they want to calculate the molecular ratio of RIP1/3 or MLKL, would the "area" make more sense? Further, should the authors use the "fluorescent intensity" of RIP1/3 MLKL in the necrosome to quantify the amounts of each molecule? The reason I am drawing this attention is that in the necrosome structures, for example in Fig. 5f, 4 h and 6h, although it seems that 6h has a longer RIP3 structure, but 4h structure is brighter. How can the authors suggest that there were more RIP3 molecules in 6h compared to 4h? If the authors use "area" or the "total intensity" to quantify RIP1/3 and MLKL in this work from the beginning to the end, what would the molecular ratio be? Still 1:3 (RIP1/3)? Based on the 4f data, alternative conclusion could be "as more RIP3 was induced, necrosome can be more and RIP3 "line structure" can be longer, but it seems that RIP3 molecules in each necrosome is less as the lower intensity of RIP3 in each RIP3 "line structure" compared to 8h to 4h or 6h to 4h." Why would this be impossible?

3 The lack of details of STORM also raises a concern. Based on the current version, I suppose the authors used a

secondary antibody. But did the authors know how many CF647 molecules were on each secondary antibody? And whether each primary antibody binds to how many secondary antibodies? Even if the primary antibody is monoclonal, it may still bind to multiple epitopes on the antigen (although this is less likely, but it could be). Therefore, all of these parameters can compromise the quantification. Could you please clarify how this part was calculated? Let me provide a hypothetical example to illustrate my point.

Let's say the authors used anti-HA and anti-flag, each primary antibody binds to the tag (antigen) at a 1:1 ratio (one antibody to one antigen). However, the anti-HA is raised by rabbit and the anti-flag is raised by mouse. The anti-rabbit secondary antibody can bind to anti-HA with a 5:1 ratio (5 anti-rabbit bind to 1 anti-HA), and the anti-mouse binds to anti-flag with a 10:1 ratio (10 anti-mouse bind to 1 anti-flag). At the same time, there are 3 fluorophore molecules on each anti-rabbit, while there are 5 fluorophore molecules on each anti-mouse. Thus, even at the beginning, the HA antigen and flag antigen are in a 1:1 ratio. After the secondary antibody staining on STORM, the ratio would be $1 \times 5 \times 3$ to $1 \times 10 \times 5$, resulting in a 15:50 ratio. Thus, the authors need to clarify how they deal with this issue.

4 In this, it is necessary to include the monoclonal antibodies of p-MLKL and pRIP3 into the STORM quantification. This will serve as the gold standard for experimental calculations of the output, validating the mathematical model.

5 This model is too simplified, as it only considers the molecular ratio, excluding the RIP1/3 conformation change, transcriptional change, and other molecules, such as cFLIP.

The most problematic figure is in Fig. 7. The lack of RIP1 would lead cells to more apoptosis is simply due to lack of Nf-kb activation and cFLIP and other anti-apoptosis molecular transcription, which has been known for years. If the authors want to deliver such information, then they should add CHX to block transcription, which will induce apoptosis when TNF is added. Then the authors can calculate different ratios of RIP1/3/casp8. The authors should also conduct the same experiments with the RIP1 inhibitor Nec-1 and utilize various RIP1 point mutations that can alter RIP1 conformation. Additionally, they should incorporate smac-memetic into the analysis. The authors must exclude the possibility that it is not RIP1/3 conformation, RIP1/3 kinase activity, or RIP1-mediated transcriptional regulation; then, the authors can conclude that "configurations" is the checkpoint of apoptosis/necroptosis.

6 In summary, I think this study is unique and innovative. However, due to the lack of details, I find it difficult to read and assess. Many of my concerns may have already been addressed by the authors and some clarification will do the job. Thus, I am really looking forward to a detailed revision.

(Remarks on code availability)

I am not an expert in reviewing the code.

Reviewer #4

(Remarks to the Author)

The authors utilized super-resolution microscopy to investigate the nanoscale organization of necrosomes. Specifically, they employed multi-color imaging to examine the stoichiometry of RIP1, RIP3, and MLKL within individual necrosomes and proposed that a homotrimeric RIP3 structure per RIP1 is the minimum functional subunit required to induce necroptosis. However, the quantitative analysis using STORM appears methodologically limited, and the results are not fully convincing due to several critical issues.

The necrosome puncta in Fig. 1c appear to be localized within the nucleus, which contradicts the current understanding and other figures in this manuscript that necrosomes form in the cytoplasm. The authors need to address this discrepancy. Additionally, the necrosome morphology in Figure 1c (e.g., puncta size and distribution) differs significantly from later figures (e.g., Figure 3d and Figure 4e). Were all images taken under identical laser illumination conditions? The fluorescence signal in the bottom row of Fig. 1c is significantly weaker than in the top row, suggesting possible inconsistencies in imaging parameters or sample preparation.

The necrosome patterns in Fig. 2b are quite different from the punctate structures in Fig. 1c. The authors should clarify whether this discrepancy reflects experimental variability, distinct necroptosis stages, or imaging artifacts.

The resolution of the STORM images in Fig. 2b is insufficient to determine whether rod-like necrosomes represent a common or rare structural state. The authors should quantify the proportion of punctate versus rod-like necrosomes during TSZ treatment to support their claims of morphological heterogeneity.

The proposed model in Fig. 2f is based on immunoprecipitation data reflecting average interactions across a population of cells. This bulk biochemical approach cannot resolve the stoichiometry or spatial organization of individual necrosomes, rendering the model speculative in the context of single-complex imaging.

In Figures 3d and 3e, the authors infer a RIP1:RIP3 ratio of 1:3 based on two-color STORM imaging, but this is problematic. First, RIP1 and RIP3 are expected to colocalize tightly within necrosomes (10-20 nm scale), yet Figures 3d and 3e show minimal overlap at the nanometer scale. Second, the dyes CF647 and CF568 have different photophysical properties (e.g., blinking kinetics, photon output), which may bias the localization patterns. Additionally, primary-secondary antibody labeling can introduce variability in epitope accessibility and signal amplification. The images in Figure 4e do not convincingly support the model shown in Figure 8.

In Fig. 8, the proposed RIP1-RIP3-MLKL assembly model lacks direct experimental validation from the imaging data. The absence of consistent nanoscale colocalization among RIP1, RIP3, and MLKL, combined with unresolved technical artifacts in multi-color STORM, critically undermines the credibility of the model.

(Remarks on code availability)

Version 1:

Reviewer comments:

Reviewer #1

(Remarks to the Author)

The authors addressed my questions adequately.

(Remarks on code availability)

Reviewer #2

(Remarks to the Author)

The authors have sufficiently addressed all my questions. I'd like to recommend its publication in the present form.

(Remarks on code availability)

Reviewer #3

(Remarks to the Author)

I am truly impressed by the revision. Of note, the MAP7-based calibration system helps a lot to solidify the conclusion. It is well done.

I only have a minor concern about Figure 8. In this model, it may be read that upon TNF, when there is no caspase-8, the cell undergoes necroptosis, while when there is no RIPK3, the cell undergoes apoptosis. This is simply untrue in biology, as many cells, including HeLa, which doesn't express RIPK3, TNF alone treatment cannot engage apoptosis. Also, the reason for caspase-8 deficiency that can support necroptosis is not only the reason presented in this figure, e.g., RIPK1/3 can serve as a caspase-8 substrate.

The authors need to revise this model figure to (of course) highlight this unique work, but should also follow the biology published by others in our community.

This paper should be accepted.

(Remarks on code availability)

Reviewer #4

(Remarks to the Author)

The authors have effectively addressed the comments raised during the first round of review, resulting in a significantly improved manuscript. I now recommend it for publication.

One final point: I suggest the authors temper their claims regarding precise stoichiometric ratios, such as the reported 3:1 ratio of RIP3 to RIP1. As shown in Figure 3m and Figure 4j, the stoichiometry exhibits a broad distribution. Given the inherent challenges in quantitative super-resolution imaging, such as variability in antibody labeling, dye blinking, and data analysis, accurate quantification remains difficult in STORM. Therefore, the authors should exercise caution when presenting conclusions based on exact numerical values.

(Remarks on code availability)

Point-by-point response to the reviewers' comments

“Decoding necrosome assembly: harmonizing signal amplification and attenuation through optimal RIP3 stoichiometry”

We sincerely thank all the reviewers for their thoughtful, constructive, and encouraging comments that have significantly helped us improve the clarity, rigor, and overall quality of our manuscript. In response, we have thoroughly revised the manuscript and addressed each point raised. Below, we provide detailed point-by-point responses. For clarity, the reviewers' comments are presented in *italic* and black, our responses are in blue, and changes made to the manuscript are indicated in red.

Reviewer #1 (Remarks to the Author):

This study identifies the RIP1-bound RIP3 homotrimer as the minimal functional unit of the necrosome, enabling signal amplification while preserving threshold sensitivity. A notable finding is that excessive RIP3 oligomerization attenuates necroptotic signaling, revealing a size-control mechanism dynamically regulated by RIP1 (restrictive) and RIP3 (expansive), with MLKL unexpectedly limiting assembly. In contrast, caspase-8 recruitment by RIP1 follows a linear, non-oligomeric pattern. These distinct assembly modes create a biphasic necroptotic response, illustrating how signalosomes balance configuration and stoichiometry for efficient signaling—providing new insights for therapeutic and synthetic biology applications. The super-resolution microscopy data offer valuable advances in understanding signalosome dynamics, though additional causal experiments would strengthen the conclusions, particularly regarding RIPK3 oligomer size control in signal amplification/limitation.

Response: We thank the reviewer for the careful reading and positive appraisal of our work, as well as for the constructive comments that have helped us further strengthen the manuscript.

Specific Comments:

1. *MLKL's inhibitory role:*

most interestingly, the authors' observation that p-MLKL negatively correlates with RIPK3 higher-order assembly aligns with prior structural work (PMID: 24095729) showing MLKL-bound RIP3 adopts an inactive conformation. This

suggests a feedback mechanism where MLKL, as a substrate, restricts RIPK3 kinase activity by limiting RIPK3 fibrilization. To solidify this claim, the authors should provide direct experimental evidence (e.g., co-IP assays) demonstrating MLKL's physical restriction of RIPK3 oligomerization, corroborating their correlation data (Figure 5).

Response: We thank the reviewer for highlighting this important point. While many studies have established MLKL as a downstream effector of RIP3 during necroptosis, little attention has been paid to its potential feedback role in regulating upstream necrosome dynamics. Our original manuscript, through both super-resolution imaging and mathematical modeling, suggested that MLKL not only receives signals from RIP3 but also negatively regulates RIP1–RIP3 supramolecular assembly (Figures 4 and 5).

We also appreciate the reviewer pointing us to relevant structural studies (Xie et al., Cell Rep. 2013; Meng et al., Nat Commun. 2021), which show that MLKL binding induces a conformational shift in RIP3 that could inhibit further oligomerization. These findings support the hypothesis that MLKL imposes a negative feedback mechanism by limiting RIP3 fibrilization.

To strengthen this conclusion, as the reviewer suggested, we have now performed co-immunoprecipitation (co-IP) assays to directly test whether MLKL physically restricts RIP3 oligomerization. As shown in Fig. R1, overexpression of MLKL significantly reduced the co-precipitation of HA-RIP3 with Flag-RIP3, indicating that MLKL interferes with RIP3–RIP3 interaction and thereby inhibits its oligomerization. This new result has been incorporated into the revised manuscript (page 16, lines 22–25) as follows:

“To confirm MLKL-mediated feedback on RIP3 activation, we examined the interaction between Flag-RIP3 and HA-RIP3 in the presence or absence of overexpressed Myc-MLKL. MLKL expression reduced the amount of HA-RIP3 pulled down with Flag-RIP3, suggesting that MLKL weakens RIP3 homo-interaction and thereby inhibits RIP3 oligomerization (Fig. 5j).”

Fig. R1 (Figure 5j in revised manuscript) MLKL inhibits RIP3 oligomerization. j Co-immunoprecipitation in 293T cells transiently expressing HA-RIP3, Flag-RIP3, and Myc-MLKL. Lysates were immunoprecipitated with anti-Flag beads, and inputs/IPs were immunoblotted for Flag, HA, and MLKL. HA-RIP3 binding to Flag-RIP3 was quantified as IP/input ratio (mean ± SD, n=3). p value was calculated using an unpaired two-tailed Student's t-test. ***p < 0.001.

2. RIPK3 oligomerization and signaling:

While the data show that RIPK3 expression positively correlates with fibril length (consistent with pro-necroptotic activity), the conclusion that "excessive RIP3 homo-assembly attenuates signaling" appears contradictory without MLKL context. To resolve this, I suggest the authors to perform time-course analyses tracking RIPK3 fibril length and cell death in MLKL-proficient vs. -deficient systems. A predicted outcome:

- o Without MLKL: RIPK3 fibrils grow indefinitely without cell death.
- o With MLKL: Fibril elongation halts (or reverses) as p-MLKL accumulates, linking fibril size control to signal transduction.

This would unify the observations, emphasizing that RIPK3 oligomerization's functional output depends on MLKL-mediated feedback.

Response: We thank the reviewer for this excellent suggestion, which aligns closely with one of core principles of our mathematical model—namely, that necroptotic signaling is not indefinitely amplified, but rather balanced by feedback regulation. The conclusion that "excessive RIP3 homo-assembly attenuates signaling" is indeed drawn in the presence of MLKL, and we agree that this context should be explicitly emphasized.

To clarify this point and directly test the reviewer's hypothesis, we performed

new time-course experiments tracking RIP3 fibril growth and cell death in MLKL-proficient and MLKL-deficient HeLa cells stably expressing Flag-RIP3. As shown in Fig. R2, in the absence of MLKL, RIP3 fibrils continue to elongate over time, but cell death remains minimal. In contrast, in MLKL-proficient cells, RIP3 fibril elongation plateaus as p-MLKL levels rise, coinciding with the onset of cell death.

Fig. R2 (Figure 5k-m in revised manuscript) MLKL restricts RIP3 fibril elongation. **k-m** Flag-RIP3–expressing wild-type or MLKL-KO HeLa cells treated with TSZ over time. Cell death by PI uptake (mean \pm SD, $n = 3$ biological replicates) (**k**). Dual-color STORM of RIP3 and p-MLKL (**I**); insets show necrosome morphology. RIP3 fibril lengths quantified (**m**; $n = 67/111$ structures at 2h, 64/135 at 4h for WT/KO). p values were calculated using Ordinary one-way ANOVA (**k**) and Kruskal-Wallis test (**m**). *** $p < 0.001$, **** $p < 0.0001$. Scale bars: 10 μm (overview) and 200 nm (insets).

These data support a negative feedback loop, in which MLKL activation inhibits further RIP3 supramolecular assembly, thereby converting an expanding signal into a terminal output. The following paragraph has been added to the revised manuscript (page 16, lines 25–29):

“We further examined RIP3 fibril dynamics and necroptosis over time in MLKL-

proficient and MLKL-deficient cells (Fig. 5k–m). In MLKL-deficient cells, RIP3 fibrils continued to elongate over time, but cell death remained low. In contrast, in MLKL-proficient cells, RIP3 fibril growth plateaued as p-MLKL levels increased, followed by cell death. These data demonstrate that MLKL provides negative feedback, constraining RIP3 fibril elongation and ensuring signal termination.”

We are grateful to the reviewer for prompting these experiments, which allowed us to directly demonstrate the feedback inhibitory role of MLKL. This finding also supports the broader view that necroptosis is a regulated, checkpoint-containing process, consistent with previous reports involving NF- κ B, caspase-8, and ESCRT pathways. The MLKL–RIP3 feedback loop may represent an additional layer of control ensuring appropriate necroptotic activation in diverse cellular contexts.

3. *Minor correction:*

o Figure 3G: "oligermer" → "oligomer"

Response: We thank the reviewer for pointing out the spelling mistake. We have corrected all typographical errors we could find throughout the manuscript.

Reviewer #2 (Remarks to the Author):

The manuscript by Li et al. reports the experimental characterization of necrosome assembly by using super-resolution microscopy (STORM) and mass spectrometry (MS). The necrosome is an important signaling complex in necrotic cell death that has broad implications in various human diseases. The results complement the existing structural data from NMR with in vivo STORM and MS data, thus providing important insights into the function and assembly of necrosome. The manuscript may benefit from addressing the following concerns before formal publication:

Response: We sincerely thank the reviewer for the thoughtful and encouraging assessment of our manuscript. We are grateful for the constructive suggestions, which have helped us improve both the clarity and rigor of our work. Below, we provide detailed point-by-point responses to each of the reviewer’s comments.

(1) *The abstract may need one or two sentences to describe the importance of*

necrosome assembly. The introduction may benefit from a brief introduction of the necrosome function and its implication in certain human diseases.

Response: Thank you for this insightful suggestion. We agree that a clearer contextualization of necrosome function and its pathological relevance would benefit the manuscript. Accordingly, we have revised both the abstract and the introduction to explicitly emphasize the biological significance of necrosome assembly and its implications in human diseases.

Specifically, we have added the following sentence to the beginning of the abstract (page 1, lines 24–25):

“Necrosome assembly is essential for necroptosis, a process implicated in neurodegeneration, ischemic injury, and inflammatory diseases.”

In the introduction (page 2, lines 22–28), we have added a new paragraph that reads:

“Necrosome is a supramolecular signaling complex that serves as the central execution module of necroptosis. Necrosome assembly not only ensures the elimination of infected or damaged cells but also promotes the release of damage-associated molecular patterns (DAMPs), thus amplifying inflammatory responses. Importantly, aberrant necrosome assembly or dysregulation of its components has been implicated in a wide spectrum of pathological conditions, including ischemia-reperfusion injury, neurodegenerative diseases, inflammatory bowel disease, and certain types of cancer. As such, dissecting the principles of necrosome assembly is critical for understanding the molecular logic of necroptosis and for identifying therapeutic targets in inflammation-associated diseases.”

(2) Can the authors discuss on the structural basis and rationality of the phase diagram of MLKL activation shown in Fig. 5a-c, perhaps based on the previously published NMR structure data or AlphaFold2/3-predicted models?

Response: We thank the reviewer for the insightful questions regarding the structural basis and plausibility of the MLKL-activation phase diagram shown in Figure 5a–c. From a structural biology perspective, we integrated published high-resolution structural and functional data with our own imaging observations to systematically analyze how the three key variables in Figure 5 modulate RIP3 polymerization.

For Figure 5a, increasing TNF enhances RIP1 recruitment and promotes the formation of RIP1-RIP3 hetero-amyloid fibers. Solid-state NMR has shown that the RHIM segments of RIP1 and RIP3 alternately stack to form a highly stable β -spine (Mompeán et al., Cell. 2018), in which core tetrads (e.g., IQIG/VQVG) together with Gln/Cys–Ser side-chain pairing stabilize the hetero-assembly thermodynamically (Fig. R3a). Therefore, elevated active RIP1 not only nucleates numerous short hetero-assemblies but can also geometrically “cap” elongating RIP3 homo-filaments, probably reducing their mean length (Fig. R3b).

Fig. R3 (Figure S7 in revised manuscript) Structural basis for the phase diagram of necrosome assembly. a Structural models of RIP1–RIP3 hetero-amyloid (PDB: 5V7Z) and RIP3–RIP3 homo-amyloid (PDB: 7DAC) fibrils. **b, c** Proposed impact of varying TNF/RIP1 (**b**) and RIP3 (**c**) levels on necrosome configuration inferred from fibril stoichiometry. **d** Crystal structure of the MLKL-bound RIP3 complex (PDB: 7MON) showing RIP3 in an inactive conformation. **e** Schematic model of RIP3 distribution in the presence of high-level MLKL, suggesting MLKL-mediated spatial restriction of RIP3 oligomerization.

For Figure 5b, higher RIP3 expression markedly promotes RIP3 homo-filament extension. This accords with the law of mass action: at essentially constant RIP1 levels, increasing the concentration of RIP3 monomers raises the probability of RHIM axial pairing and β -sheet stacking, favoring longer polymers (Figure 3c). Prior work provided detailed cryo-EM/ssNMR structures of RIP3

RHIM homo-fibrils (Wu et al., PNAS. 2021), further supporting RIP3's intrinsic capacity for stable homo-assembly.

In Figure 5c, we observe that higher MLKL expression is associated with a pronounced shortening of RIP3 homo-filaments, indicating an inhibitory effect of MLKL on RIP3 polymerization. MLKL comprises an N-terminal four-helix bundle (4HB) effector domain and a C-terminal pseudokinase (psK) regulatory domain. The MLKL psK binds RIP3 and is phosphorylated by it, which in turn releases the 4HB to oligomerize and translocate to the plasma membrane. Although there is, to our knowledge, no atomic-resolution report of MLKL directly capping RIP3 homo-filament ends, previous studies have shown that MLKL binding induces a conformational change in RIP3 that renders it inactive (Fig. R3d), thereby attenuating its catalytic function (Xie et al., Cell Rep. 2013; Meng et al., Nat Commun. 2021). We therefore hypothesize that this conformational conversion might disrupt the axial stacking interfaces required for continued RHIM-mediated extension, impeding further growth of RIP3 homo-filaments. Thus, under elevated MLKL expression, RIP3 assembly becomes more dispersed into smaller aggregates rather than concentrating into long fibrillar structures (Fig. R3e). This “distributed-assembly” tendency can be understood as RIP3 forming multiple smaller complexes under high-MLKL conditions to achieve efficient engagement with MLKL. While the detailed mechanism by which MLKL limits RIP3 polymerization remains to be fully elucidated, and consistent with the Reviewer 1’s comments, our additional data further substantiate the biochemical evidence that MLKL imposes a negative feedback by limiting RIP3 polymerization (please see Figs. R1 and R2).

Fig. R1 (Figure 5j in revised manuscript) MLKL inhibits RIP3 oligomerization. j Co-immunoprecipitation in 293T cells transiently expressing HA-RIP3, Flag-RIP3, and Myc-MLKL. Lysates were immunoprecipitated with

anti-Flag beads, and inputs/IPs were immunoblotted for Flag, HA, and MLKL. HA-RIP3 binding to Flag-RIP3 was quantified as IP/input ratio (mean \pm SD, $n=3$). p value was calculated using an unpaired two-tailed Student's t -test. *** $p < 0.001$.

Fig. R2 (Figure 5k-m in revised manuscript) MLKL restricts RIP3 fibril elongation. **k-m** Flag-RIP3-expressing wild-type or MLKL-KO HeLa cells treated with TSZ over time. Cell death by PI uptake (mean \pm SD, $n = 3$ biological replicates) (**k**). Dual-color STORM of RIP3 and p-MLKL (**I**); insets show necrosome morphology. RIP3 fibril lengths quantified (**m**; $n = 67/111$ structures at 2h, 64/135 at 4h for WT/KO). p values were calculated using Ordinary one-way ANOVA (**k**) and Kruskal-Wallis test (**m**). *** $p < 0.001$, **** $p < 0.0001$. Scale bars: 10 μ m (overview) and 200 nm (insets).

In response to the reviewer's comment, we have added Figure S7 and the following sentences in the revised manuscript (page 16, line 30–page 17, line 5) as follows:

“Our data and prior structures support a unified structural rationale for the phase behavior in Fig. 5a–c (Fig. S7). TNF elevates RIPK1 recruitment, favoring the alternated RIPK1:RIPK3 RHIM β -spine and effectively capping or diverting

RIP3 from homo-elongation, thereby shortening RIP3 filaments. Increasing RIP3 expression follows mass-action logic and promotes RHIM-mediated axial stacking, yielding longer homo-filaments. By contrast, higher MLKL expression correlates with shorter RIP3 filaments without a concomitant rise in MLKL phosphorylation (Fig. 5k–m). This is consistent with structural evidence that MLKL binding induces a conformationally inactive state of RIP3, reducing the availability of stacking-competent RHIM interfaces and redistributing RIP3 into smaller assemblies. In summary, RIP3 supramolecular assembly is dynamically regulated by multiple factors (i.e., TNF dosage, RIP1 abundance, RIP3 expression, and MLKL levels), ensuring precise spatial control over necrosome formation and necroptotic output.”

(3) Can the authors discuss on the structural basis and rationality of their findings on the stoichiometry of necrosome assembly? How are their results reconciled with the previously published data?

Response: We thank the reviewer for raising this important point. For the proposed necrosome stoichiometry RIP1:RIP3:MLKL:CASP8 \approx 2:6:4:1 (Fig. R4), we performed an integrative analysis combining structure-guided modeling, functional readouts, and prior literature, and we clarify below its structural rationality and consistency with published data.

Fig. R4 (Figure 2g in revised manuscript) Structural model of necrosome molecular assembly. g Proposed model of the modular and cooperative architecture of the necrosome. Published (PDB) and predicted (AlphaFold) structural models illustrate domain-specific interactions: DD oligomerization (TNFR, TRADD, RIP1), RHIM-driven hetero-/homo-amyloid assembly (RIP1–RIP3), and downstream effector recruitment (FADD–caspase-8; RIP3–MLKL). Stoichiometric ratios (RIP1:RIP3:MLKL:Caspase-8 \approx 2:6:4:1) were derived from IP-MS analysis.

First, the observation that RIP3 exceeds RIP1 is fully compatible with the known RHIM-mediated assemblies underlying necrosome formation. RIP1 and RIP3 form hetero-amyloid fibers via alternating RHIM pairing (Fig. R4; PDB 5V7Z), whereas RIP3 alone can self-assemble into homo-amyloid fibrils through the same motif (Fig. R4; PDB 7DAC). Using the 5V7Z hetero-tetramer “single layer” as a seed and strictly applying the left-handed helical parameters derived from EM density map EMD-30622, we extended RIP3 homo-oligomeric units (7DAC) from both ends of the seed. These structure-guided extensions showed that the composite assembly becomes stable only when $\text{RIP3:RIP1} \geq 6:2$ (Fig. R5).

Fig. R5. Helical amyloid fibril models of RIP1:RIP3 with stoichiometries from 2:3 to 2:6. RIP1 and RIP3 segments are shown in purple and green, respectively; opaque models correspond to 250 ps simulation conformations, and the semi-transparent yellow model indicates the initial crystal-derived assembly. In the 2:3 model, edge-adjacent RIP3 loops are disordered and the upper β -strands deviate markedly from the starting structure. Incorporation of an additional RIP3 subunit (2:4) improves local loop organization but introduces pronounced torsional distortion of the β -strands. The 2:5 assembly remains poorly ordered, whereas the 2:6 model exhibits enhanced structural regularity.

Second, the 2:6 relationship between RIP1 and RIP3 defines a minimal functional module that closes geometrically and functionally with four MLKL molecules. In this 2:6 block, two RIP3 subunits primarily engage RIP1 to stabilize the hetero-seed, while the remaining four RIP3 subunits align to form a RIP3 tetramer. Within this tetramer, each RIP3 kinase domain engages one MLKL pseudokinase domain (Fig. R4; PDB 7MON), so these four RIP3s recruit and phosphorylate four MLKL molecules. Activated MLKLs then assemble into the tetrameric execution complex (Fig. R4, yellow assembly), a widely reported functional species. Consequently, the ratio RIP3:MLKL \approx 6:4 is structurally and functionally well-matched.

Third, regarding Caspase-8 (CASP8), the FADD–RIP1 DD interface and the FADD–Caspase-8 DED complex (Fig. R4; PDB 8YD8) constitute the proximal regulatory branch of the necrosome. Prior structural work indicates that in the

presence of cFLIP, Caspase-8 enters the complex via the tandem DED axis but fails to form an active catalytic dimer, remaining embedded as a non-catalytic monomer (Fig. R6). Our revised manuscript also includes biochemical validation of this state (please see Fig. R13). Thus, one Caspase-8 monomer suffices to capture its cFLIP-dependent gating role, supporting the 2:6:4:1 stoichiometry.

Fig. R6 Cryo-EM structural analysis of FADD:Caspase-8 complexes defines the catalytic dimer architecture required for co-ordinated control of cell fate. Adapted from Fox et al., Nat Commun. 2021.

Taken together, the 2:6:4:1 ratio emerges as the minimal, stable functional unit dictated by four convergent constraints: (i) the 1:1 local geometry of RHIM-based hetero-nucleation (RIP1:RIP3), (ii) the stability threshold of RIP3 homo-extension, (iii) saturable occupancy of a tetrameric MLKL execution unit, and (iv) cFLIP-regulated, limited incorporation of Caspase-8 as a proximal gate. This stoichiometry is supported across geometric, kinetic, and functional dimensions by high-resolution structures and our data, providing a robust, testable, and literature-consistent structural explanation for necrosome assembly.

In response to the reviewer's comment, we have updated Figure 2g in the revised manuscript and added the following paragraph (page 6, line 28–34) to elaborate:

“This assembly ratio outlines an assembly unit that reflects a mechanistically

optimized stoichiometry of the necrosome: two RIP1 molecules together with two RIP3 molecules form a heterotypic RHIM 'seed' (PDB: 5V7Z); among six RIP3 molecules, four assemble into a RIP3 tetramer, the minimal execution unit of necroptosis (PDB: 7DAC), which activates MLKL (PDB: 7MON) and thereby drives four MLKL monomers to form the functional MLKL tetramer. Caspase-8 participates as a non-catalytic monomeric regulator (PDB: 8YD8). All of these features are consistent with prior reports and represent a structurally and functionally coherent scheme for necrosome assembly.”

(4) There are numerous places that authors used exaggerating or hyperbolic words or phrases, which are unnecessary for scholarly presentation. For example, on page 10, the 2nd paragraph starts with “... overwhelmingly underscores ...”. It is confusing why the authors used “overwhelmingly” here. On page 2, the last sentence reads “our findings challenge the conventional view of ...”. But the results do not appear to really “challenge” but instead just “improve” the existent understanding of necrosome assembly and associated signal transduction. The use of “challenge” in several places is not well supported by the work itself, and make it read more like overselling the paper by the authors.

Response: We sincerely appreciate the reviewer’s thoughtful feedback on the tone and language of our manuscript. Upon re-examination of the manuscript, we acknowledge that certain phrases may indeed be interpreted as overly assertive or not sufficiently justified by the presented data. In response, we have carefully revised the manuscript to ensure that all claims are appropriately tempered and strictly supported by experimental evidence.

Specifically:

We have replaced “overwhelmingly underscores” with a more neutral phrase such as “strongly supports” or “is consistent with.” We have substituted “challenge the conventional view” with “refine” or “extend current understanding of” necrosome assembly and signal transduction, which we believe better reflects the incremental and mechanistic nature of our findings.

(5) There are no sufficient details provided for their computational modeling. The Methods section may benefit from a revision that provides necessary details required for reproduction of their modeling results.

Response: We thank the reviewer for emphasizing the importance of transparency and reproducibility in computational modeling. We fully agree that detailed methodological information is essential for enabling validation and reuse of our modeling framework by other researchers. In response, we have substantially expanded the “Mathematical Modeling” subsection of the Methods section to include the necessary technical details. Specifically, we now provide:

“Modeling Principle

We established a quantitative dynamic model of TNF signaling-induced cell death complex assembly based on the law of mass action. The modeling process begins with identifying the fundamental biochemical reactions involved in the system, classifying them by reaction type, followed by translating each reaction into a rate law, and finally formulating a system of ordinary differential equations (ODEs) that captures the temporal evolution of all molecular species.

1. Reaction Types and Rate Laws

Each reaction in the signaling cascade falls into one of the following fundamental categories:

- Association (binding): e.g., $A + B \xrightarrow{k_{on}} C$ Rate: $r_{assoc} = k_{on}[A][B]$
- Dissociation (unbinding): e.g., $C \xrightarrow{k_{off}} A + B$ Rate: $r_{dissoc} = k_{off}[C]$
- Catalytic activation (e.g., phosphorylation, cleavage): e.g., $A \xrightarrow{k_{cat}} A^*$ Rate: $r_{cat} = k_{cat}[A]$
- Degradation or inactivation: e.g., $A^* \xrightarrow{k_{deg}} \emptyset$ Rate: $r_{deg} = k_{deg}[A^*]$
- Multimerization or oligomer assembly (e.g., higher-order RIP3 complex): e.g., $nA \xrightarrow{k_n} A_n$ Rate: $r_{oligo} = k_n[A]^n$

Each reaction is assigned a rate constant k and contributes to the change in concentration of the involved species.

2. Construction of the ODE System

For each molecular species X_i , its concentration over time is described by:

$$\frac{d[X_i]}{dt} = \sum_{j=1}^R v_{ij} \cdot r_j(X, \theta)$$

Where:

- $[X_i]$: concentration of species i ;
- r_j : rate of reaction j ;
- ν_{ij} : stoichiometric coefficient of species X_i in reaction j ;
- θ : vector of kinetic parameters.

This leads to a system of ODEs such as:

$$\begin{cases} \frac{d[A]}{dt} = -k_{on}[A][B] + k_{off}[C] \\ \frac{d[B]}{dt} = -k_{on}[A][B] + k_{off}[C] \\ \frac{d[C]}{dt} = k_{on}[A][B] - k_{off}[C] - k_{cat}[C] \\ \dots \end{cases}$$

Each equation reflects the net effect of all reactions in which the species participates.

Basic Components Assembly Modeling

To capture the hierarchical and sequential assembly of signaling complexes, we developed a mass-action-based kinetic model describing two fundamental interaction modules: a two-component system (A-B) and a three-component extension (A-B-C). These modules abstract the core steps of scaffold recruitment, binding, and downstream activation.

1. Two-Component Assembly (A + B)

In the first stage, protein A binds to protein B, forming an intermediate complex $A_m B_n$, which then facilitates the activation of B. The assembly and activation steps are modeled as:

- $A_m B_n$: the assembled but inactive complex formed by m units of A and n units of B.
- B^* : the activated form of protein B.
- k_1, k_{-1} : binding and unbinding rates between A and B.
- κ_1 : rate constant for irreversible activation of B.

2. Three-Component Assembly (A + B + C)

In the second stage, activated B (B^*) recruits protein C and promotes its activation, forming a ternary complex:

- $B_n^* C_h$: the intermediate complex consisting of n activated B units and h C units.
- C^* : the activated form of protein C.

- k_2, k_{-2} : binding and unbinding rates between B^* and C.
- κ_2 : irreversible activation rate of C.

3. Model Variables and Interpretation

The stoichiometric coefficients m, n, h reflect the molecular configurations or oligomeric states of A, B, and C within their respective complexes. These variables allow the model to capture non-1:1 stoichiometries, which are common in supramolecular signaling assemblies. The complete set of ordinary differential equations derived from these reactions describes the time evolution of each species and complex. For example:

$$\frac{d[A_m B_n]}{dt} = k_1[A]^m[B]^n - k_{-1}[A_m B_n] - \kappa_1[A_m B_n]$$

$$\frac{d[B_n^* C_h]}{dt} = k_2[B^*]^n[C]^h - k_{-2}[B_n^* C_h] - \kappa_2[B_n^* C_h]$$

$$\frac{d[B^*]}{dt} = \kappa_1[A_m B_n] - k_2[B^*]^n[C]^h + k_{-2}[B_n^* C_h]$$

This modeling framework captures the essential sequential logic of multi-protein assembly: scaffold \rightarrow recruitment \rightarrow activation \rightarrow downstream assembly. A detailed list of equations and parameter values is provided in Supplementary Table 1.

Determined Necrosome Assembly Modeling

To systematically explore the emergent and intricate features of TNF-induced necrosome formation, we developed a mass-action-based kinetic model that incorporates the core molecular events of protein recruitment, binding, catalytic activation, and mutual inhibition. The model is formulated as a set of coupled ordinary differential equations (ODEs), following the principles described in the Modeling Principle section. It captures the time-dependent evolution of individual molecular species involved in necrosome assembly and downstream necroptosis signaling.

The model includes the following key modules:

- 1). TNF-induced recruitment of RIP1 and TRADD to complex I;
- 2). Dissociation of RIP1 from complex I and its incorporation into the necrosome;
- 3). Assembly of necrosome via RIP1-RIP3 binding and phosphorylation;
- 4). Activation of MLKL by phosphorylated RIP3;
- 5). Crosstalk and mutual inhibition between RIP3 and caspase-8 (C8);
- 6). TRADD-mediated activation of C8.

This section focuses on modeling the core necrosome assembly cascade, which includes the stepwise formation of RIP1-RIP3-MLKL complexes and their activation events.

1. Reactions and Stoichiometry

Mass spectrometry-based quantitative analysis revealed the stoichiometric ratios of

key components in the necrosome as $RIP1:RIP3:MLKL = 1:3:2$. The major reactions modeled are:

- Reaction R1: Binding of one active RIP1 molecule with three RIP3 molecules to form a ternary complex

- Reaction R2: Catalytic phosphorylation of RIP3 by RIP1 within the RIP1-RIP3 complex

- Reaction R3: Binding of three phosphorylated RIP3 molecules to two MLKL molecules

- Reaction R4: Phosphorylation and activation of MLKL

Where:

- $RIP1_{active}$: the activated form of RIP1 released from complex I
- $RIP3$, $RIP3_{pho}$: unphosphorylated and phosphorylated forms of RIP3
- $MLKL$, $MLKL_{pho}$: inactive and activated MLKL
- k_1 , k_{-1} , κ_2 , k_3 , k_{-3} , κ_4 : forward, reverse, and catalytic rate constants for each step

2. Differential Equations

The dynamics of each species is governed by the following ODEs:

- For the RIP1-RIP3 ternary complex:

$$\begin{aligned} \frac{d[RIP1_{active_RIP3_3}]}{dt} &= k_1[RIP1_{active}][RIP3]^3 - k_{-1}[RIP1_{active_RIP3_3}] - \kappa_2[RIP1_{active_RIP3_3}] \end{aligned}$$

- For free active RIP1:

$$\frac{d[RIP1_{active}]}{dt} = -k_1[RIP1_{active}][RIP3]^3 + k_{-1}[RIP1_{active_RIP3_3}] + \kappa_2[RIP1_{active_RIP3_3}]$$

- For phosphorylated RIP3:

$$\begin{aligned} \frac{d[RIP3_{pho}]}{dt} &= 3\kappa_2[RIP1_{active_RIP3_3}] - 3k_3[RIP3_{pho}]^3[MLKL]^2 + 3k_{-3}[RIP3_{pho_3_MLKL_2}] \\ &\quad + 3\kappa_4[RIP3_{pho_3_MLKL_2}] \end{aligned}$$

- For the RIP3-MLKL complex:

$$\frac{d[\text{RIP3}_{\text{pho}_3}\text{-MLKL}_2]}{dt} = k_3[\text{RIP3}_{\text{pho}}]^3[\text{MLKL}]^2 - k_{-3}[\text{RIP3}_{\text{pho}_3}\text{-MLKL}_2] - \kappa_4[\text{RIP3}_{\text{pho}_3}\text{-MLKL}_2]$$

- For activated MLKL:

$$\frac{d[\text{MLKL}_{\text{pho}}]}{dt} = 2\kappa_4[\text{RIP3}_{\text{pho}_3}\text{-MLKL}_2]$$

The equations presented here focus on modeling the core necrosome assembly process and the key interactions between RIP1, RIP3, and MLKL. However, due to the multi-step, highly interconnected nature of necrosome formation, the actual system is more complex and involves additional molecular interactions. These include further crosstalk between RIP3, caspase-8, TRADD, and other components of the necrosome signaling network. As such, the full model comprises 23 reaction equations and 39 ODEs in total, capturing a broader range of protein-protein interactions, feedback loops, and regulatory mechanisms that govern necrosome assembly and its downstream effects. The complete set of equations and parameter values is provided in Supplementary Table 2.

Parameters Values and Initial Amounts Selection

1. Kinetic Parameters

The selection of kinetic parameters for our model was based on experimental data, and these parameters were optimized using R-square fitting. The parameters include rate constants for various reactions such as binding/unbinding rates, catalytic activation rates, and stoichiometric coefficients. These values were chosen to be within biologically relevant ranges, with a focus on fitting the experimental data derived from time-course measurements of protein concentrations involved in necrosome assembly. The fitting procedure aimed to minimize the discrepancy between experimental and model-predicted values using the coefficient of determination (R-square), which quantifies the goodness of fit. The R-square value is calculated as:

$$R^2 = 1 - \frac{\sum_{i=1}^N ([X_i^{\text{exp}}](t_i) - [X_i^{\text{sim}}](t_i; \theta))^2}{\sum_{i=1}^N ([X_i^{\text{exp}}](t_i) - \overline{[X_i^{\text{exp}}]})^2}$$

Where:

- $[X_i^{\text{exp}}](t_i)$ are the experimental concentration measurements at time t_i ,
- $[X_i^{\text{sim}}](t_i; \theta)$ are the model-predicted concentrations using the parameter set θ ,
- $\overline{[X_i^{\text{exp}}]}$ is the mean value of the experimental concentrations,

- N is the number of data points used in the fitting.

The parameters were iteratively adjusted to maximize R^2 , ensuring that the model predictions closely matched the experimental data, thereby reflecting the dynamic behavior of the signaling network.

2. Initial Amounts of Species

The initial concentrations of the species involved in necrosome assembly, such as RIP1, RIP3, and MLKL, were selected based on previously published studies and experimental proteomics data. These initial amounts were used to set the starting conditions for the ODE simulations. For species where direct initial concentration data were unavailable, reasonable estimates were made based on the literature, ensuring they remained within physiological ranges. The initial conditions were set for the concentrations of RIP1, RIP3, and MLKL at $t=0$, which were assumed to be in their inactive forms prior to TNF stimulation. These values were then used as the baseline for fitting the model.

Stochastic Simulation of Necrosome Assembly Dynamics

To account for intrinsic cell-to-cell variability, we performed large-scale stochastic simulations of TNF-induced necrosome assembly by systematically perturbing kinetic parameters and structural features within biologically plausible ranges.

1. Parameter Space and Latin Hypercube Sampling

We defined a n_p -dimensional parameter space $\Theta = \{\theta_1, \theta_2, \dots, \theta_{n_p}\}$, with each $\theta_i \in [\theta_{i,\min}, \theta_{i,\max}]$ sampled in log-scale according to prior physiological knowledge:

$$\theta_i \sim \log U(\log \theta_{i,\min}, \log \theta_{i,\max}) \quad \text{for } i = 1, 2, \dots, n_p$$

The sampling ranges were defined as:

- Kinetic constants (k): $\theta \in [0.1, 10]$
- Catalytic or cooperativity terms (j): $\theta \in [10^{-3}, 10^2]$
- Degradation rates (d): $\theta \in [10^{-2}, 1]$
- RIP3 filament size (n): $n \in \{1, 2, \dots, 8\}$ (discrete uniform)

Using Latin Hypercube Sampling (LHS), we generated $N = 10,000$ unique samples:

$$\{\Theta^{(1)}, \Theta^{(2)}, \dots, \Theta^{(10000)}\}$$

Each sample $\Theta^{(i)}$ corresponds to a full set of reaction rates and initial conditions for a TNF-induced signaling network model.

2. Simulation of Dynamics and Efficiency Computation

For each parameter set $\Theta^{(i)}$, the system of ODEs describing TNF-driven necrosome

assembly was numerically integrated until a steady state or fixed time point $t = T$ was reached. The core readout of the signaling response was the MLKL phosphorylation efficiency $\eta^{(i)}$, defined as:

$$\eta^{(i)} = \frac{[\text{pMLKL}]^{(i)}(T)}{[\text{MLKL}_{\text{total}}]^{(i)}}$$

where pMLKL is the steady-state concentration of phosphorylated MLKL under parameter set i .

To explore how oligomeric configurations affect signaling, we introduced a variable n_{RIP3} representing the stochastic RIP3 chain length. For each $\Theta^{(i)}$, η was evaluated at different n_{RIP3} values:

$$\eta^{(i)}(n_{\text{RIP3}}) = f(\Theta^{(i)}, n_{\text{RIP3}})$$

This enabled generation of a phase map over the plane (n_{RIP3}, η) , as illustrated in Fig. 6a.

3. Selection of High-Efficiency Models and Statistical Analysis

From the 10,000 simulated models, we extracted the top $\alpha = 5\%$ models that achieved the highest MLKL phosphorylation efficiency:

$$M_{\text{top}} = \{\Theta^{(i)} : \eta^{(i)} \geq \eta_{95\%}\}$$

where $\eta_{95\%}$ is the 95th percentile threshold. For each model in M_{top} , the corresponding optimal RIP3 chain length n^* was identified:

$$n^* = \underset{n_{\text{RIP3}}}{\text{argmax}} \eta^{(i)}(n_{\text{RIP3}})$$

These values were then statistically analyzed to determine whether specific RIP3 configurations conferred robust and optimal signaling. Additionally, to explore the influence of TNF stimulus and expression levels of RIP3 or MLKL on necrosome topology, we repeated the above simulations under perturbed conditions (e.g., [TNF], [RIP3], [MLKL]) and analyzed shifts in the distribution of n^* .

Computational Implementation

All simulations were implemented in Python using the SciPy.integrate.odeint solver for ODEs. Sampling was performed using pyDOE's Latin Hypercube Sampling routines. Each simulation ran independently in parallel. The entire code-base is available at: https://github.com/XMU-Xu/RIP3_assembly.git.

We hope these additions address the reviewer's concerns and enhance the clarity, accessibility, and scientific value of our modeling efforts.

Reviewer #3 (Remarks to the Author):

This work by Li et al. provides an interesting angle for understanding how necrosome assembly works (e.g., its output as p-MLKL and the resulting cell death). This is a very unique way to analyze the necroptosis mechanism; it should definitely be published. However, some issues require clarification and reconsideration before publication. Moreover, since some of the technical details are not fully clear in the current version, I may not agree with all the analytical approaches the authors have used (but more clarification will help address my concerns; I look forward to a revised manuscript). Also, I am not an expert in mathematical modeling. Thus, all my inputs and concerns are from a cell biology and microscopy angle.

Response: We sincerely thank the reviewer for their encouraging and constructive comments. We are particularly grateful for the recognition of the unique perspective our study brings to understanding necrosome assembly and necroptotic signaling. At the same time, we fully understand the concerns raised regarding clarity and technical detail in the original manuscript, and apologize for any confusion caused.

To address these concerns of the reviewer, we have made significant revisions to the manuscript, including:

-Implementing a more robust and reliable quantitative STORM analysis, aided by an internal MAP7 calibration reference (Fig. R10), and improving the clarity of methodological descriptions achieved through the new SMAP analysis platform (Figs. R7, R8, R9 and R17);

-Performing additional experimental validations to clarify key mechanistic findings (Figs. 11–13);

-Extensively revising figure legends, captions, and text to ensure transparency and readability, particularly from a microscopy and cell biology perspective.

We hope these revisions have resolved the reviewer's concerns and have improved the manuscript's overall clarity, rigor, and accessibility. Below, we respond to each comment in detail.

1. Note that a majority of the audience for this manuscript are still cell biologists; thus, this version of the manuscript are still hard to us to digest. A more detailed

navigation would be needed. And for the figure legend and figure description in the main text, it is too brief, lacking details about how the authors reach to these results. And sometimes, it is not clear whether the data are based on model calculations or experimental data. I am happy to read a more updated version with EXTENSIVE re-writing.

Response: We fully agree with the reviewer that the original version of the manuscript lacked sufficient clarity and accessibility for a broader cell biology audience. We appreciate this important feedback and have made substantial revisions to improve the manuscript's readability and navigability. Briefly, we have restructured the logical flow of the manuscript by reordering several figures and data panels, particularly in revised Figures 2, 3, 4, 5, and 7 (A detailed summary of these changes is provided at the end of this response). In addition, we have extensively rewritten the main text and greatly expanded figure legends to provide more context. All revised sections have been highlighted in yellow in the updated manuscript for ease of review. We apologize for any confusion caused by the original presentation and hope that these comprehensive revisions have significantly improved the manuscript's clarity, logic, and accessibility for a cell biology readership.

Several examples,

The authors keep writing "RIP3 trimer per RIP1" etc., which really reflects the 3:1 ratio of RIP3, right? I think words like "trimer", "dimer" are very confusing, as in one necrosome, it contains more than 3 RIP3 molecules.

Response: We thank the reviewer for pointing out this important issue regarding terminology. We agree that the use of terms such as "trimer" or "dimer" in the original manuscript could be misleading, especially given that the necrosome is known to be a high-molecular-weight (megaDalton-scale) complex containing multiple copies of RIP1, RIP3, and MLKL (Feoktistova et al., Mol Cell. 2011; Tenev et al., Mol Cell. 2011). Thus, these terms may incorrectly suggest fixed and discrete oligomeric states, which are not consistent with the dynamic and variable nature of supramolecular assembly in necroptosis.

Our intention was not to imply fixed oligomers, but rather to describe average stoichiometric ratios derived from biochemical and imaging data. To avoid confusion, we have revised the manuscript throughout by replacing phrases

such as “RIP3 trimer per RIP1” with more precise terminology, including “a 3:1 stoichiometric ratio of RIP3 to RIP1” or “RIP3:RIP1 ratio of approximately 3:1.” We hope these revisions will prevent misinterpretation and make the presentation clearer for both specialists and general cell biology readers.

Fig. 3g how did cell death measured? How did RIP3 dimer control? Fig. 3h, how did p-RIP3 measured? by WB? any representative images? How was RIP1 level controlled? (just citing the previous publication was not sufficient for the audience to understand the experimental details, and don't put everything into the method, as it is hard to locate the useful information)

Response: We thank the reviewer for these helpful clarifications. We agree that our original description of Figures 3g and 3h lacked sufficient experimental detail and may have been confusing, particularly in distinguishing between simulation and experimental data. For the experimental validations (lower panels labeled “Exp” in Figures 3g and 3h), cell death was measured using PI/Hoechst staining followed by conventional imaging and counting of PI-positive nuclei; RIP3 dimer was achieved by 4-OHT-mediated RIP3-HBD dimerization; p-RIP3 levels were quantified by western blotting using a phospho-specific RIP3 antibody; RIP1 levels were controlled by shRNA-mediated knockdown. All relevant updates are now included in the Figure 3 legend and in the Results and Methods sections.

Prompted by the reviewer’s earlier comment on “RIP3 trimer per RIP1”, we recognized that the original lower panel of Figure 3g interrogated fixed RIP3 oligomeric states, which did not fully capture our emphasis on RIP1–RIP3 stoichiometry. To address this, we replaced that dataset with newly acquired multi-color quantitative STORM data in *MLKL*-KO cells, directly measuring RIP1:RIP3 ratios within individual necrosomes. These new results (Fig. R11) align closely with the predictions of our mathematical model and strengthen the experimental validation. Please see the dedicated responses to the 3# and 4# comments of this reviewer for details (Figs. R10 and R11).

Fig. R11 (Figure 3I, m in revised manuscript). Quantitative STORM of necrosomes in MLKL-deficient cells. **I** Three-color STORM of necrosomes in HA-RIP1/Flag-RIP3–expressing RIP1-MLKL-DKO HeLa cells treated with TSZ for 4 h, labeled for HA-RIP1 (green), Flag-RIP3 (purple), and p-RIP3 (yellow). Insets highlight necrosome morphology with calculated Flag:HA ratios and p-RIP3 localizations. **m** Quantification of RIP3:RIP1 stoichiometry and relative p-RIP3 levels (localizations) in individual necrosomes (n = 89). Scale bars: 10 μ m (overview), 300 nm (insets).

Fig. 4g, is this simulation from a model or real experimental data? how about 4h? if it is not experimental data, then experimental data verification is a must.

Response: We thank the reviewer for pointing out the importance of distinguishing between experimental and modeling data. In the revised manuscript, we have explicitly labeled Figures 4g and 4h as Simulation panels. Specifically, Figure 4g shows predictions from a deterministic, mass-action ODE model parameterized by fitting to our experimental measurements and evaluated across RIP1:RIP3 stoichiometric regimes. Figure 4h reports stochastic simulations that incorporate cell-to-cell heterogeneity under the same stoichiometric settings. The stochastic statistics in Figure 4h converge with the deterministic surfaces in Figure 4g and consistently identify a maximum in MLKL phosphorylation efficiency at a RIP1:RIP3 ratio of \sim 1:3.

We agree that experimental support is essential. While precisely constructing preset cellular necrosomes with varying RIP1:RIP3 ratios is technically challenging, the reviewer’s follow-up suggestions (4# comment) of using monoclonal antibodies to p-MLKL and p-RIP3 in the STORM quantification are highly instructive. Following this guidance, we incorporated monoclonal antibodies against p-MLKL into a multi-color quantitative STORM workflow. This allowed us to link distributions of MLKL phosphorylation levels to the measured

local RIP1:RIP3 ratios. The resulting experimental trends mirror the model, with the highest p-MLKL fraction at ~1:3 RIP1:RIP3 (Fig. R12). Please see the dedicated responses to the 3# and 4# comments of this reviewer for details (Figs. R10 and R12). Together, these additions clearly distinguish model from experiment and provide experimental validation for the model-predicted dependence of MLKL phosphorylation on RIP1:RIP3 stoichiometry. All relevant updates have been incorporated into the Figure 4 and the Results and Methods sections.

Fig. R12 (Figures 4i, j in revised manuscript). Quantitative STORM of necrosomes in MLKL-proficient cells. i Three-color STORM of necrosomes in HA-RIP1/Flag-RIP3–expressing RIP1-KO HeLa cells treated with TSZ for 4 h, labeled for HA-RIP1 (green), Flag-RIP3 (purple), and p-MLKL (yellow). Insets highlight necrosome morphology with calculated Flag:HA ratios and p-MLKL localizations. **j** Quantification of RIP3:RIP1 stoichiometry and relative p-MLKL levels (localizations) in individual necrosomes (n = 122). Scale bars: 10 μ m (overview), 300 nm (insets).

Fig. 5 g and i, how were the MLKL and RIPK3 levels determined? The author suggested by STORM, which means STORM the whole cells or just the necrosomes? What does "amount of necrosome per cell" mean? "necrosome counts or intensity?" etc.

Without sufficient technical details about how the analysis was conducted, my judgment on these data remains very preliminary.

Response: We thank the reviewer for pointing out the need for clearer technical descriptions of our analyses in Figures 5g and 5i. We apologize for any confusion caused in the original version of the manuscript.

To clarify:

-The RIP3 and MLKL levels shown in Figures 5g and 5i were not determined by STORM imaging, but rather quantified by densitometric analysis of Western blot bands, as shown in Figures 5f and 5h, respectively. In the revision, we have re-analyzed these data to provide more precise quantification of relative protein abundance.

-The term "amount of necrosome per cell" specifically refers to the number of discrete necrosome puncta per necroptotic cell, as visualized and counted via STORM super-resolution microscopy. To avoid ambiguity, we have rephrased this throughout the manuscript as "necrosome counts per cell."

We agree that the original text lacked clarity in explaining how these quantifications were performed. In response, we have revised the legend of Figure 5 to state the methods used for protein quantification (western blot densitometry) and necrosome quantification (STORM-based cluster counting). We hope these clarifications and revisions have addressed the reviewer's concerns and improved the overall interpretability of the data.

2. Based on the quantitative approach STORM microscopy, I would ask the authors to clarify

Why choose to measure the length of RIP1/3, MLKL etc? If they want to calculate the molecular ratio of RIP1/3 or MLKL, would the "area" make more sense? Further, should the authors use the "fluorescent intensity" of RIP1/3 MLKL in the necrosome to quantify the amounts of each molecule? The reason I am drawing this attention is that in the necrosome structures, for example in Fig. 5f, 4 h and 6h, although it seems that 6h has a longer RIP3 structure, but 4h structure is brighter. How can the authors suggest that there were more RIP3 molecules in 6h compared to 4h? If the authors use "area" or the "total intensity" to quantify RIP1/3 and MLKL in this work from the beginning to the end, what would the molecular ratio be? Still 1:3 (RIP1/3)? Based on the 4f data, alternative conclusion could be "as more RIP3 was induced, necrosome can be more and RIP3 "line structure" can be longer, but it seems that RIP3 molecules in each necrosome is less as the lower intensity of RIP3 in each RIP3 "line structure" compared to 8h to 4h or 6h to 4h." Why would this be impossible?

Response: We thank the reviewer for this insightful and technically important comment. In the original manuscript, we used length measurements of RIP1, RIP3, and MLKL structures in STORM images as one of our primary readouts,

based on the well-established finding that necrosomes form higher-order, amyloid-like structures (Li et al., *Cell*. 2012; Liu et al., *PNAS*. 2017; Mompean et al., *Cell*. 2018). Consistently, the rod-shaped morphology of necrosomes was clearly observed under STORM with lateral resolution ~20 nm. Thus, we interpreted length as a proxy for necrosome assembly status.

However, we fully acknowledge the reviewer's concern that area-based measurements may offer a more general and reproducible quantification strategy, especially for readers outside the necroptosis field. Indeed, area rather than length is more commonly used for single-molecule localization microscopy (SMLM, including STORM in this study) quantification in many recent cell biology studies (Wu et al., *Trends Cell Biol.*, 2020). By contrast, fluorescence intensity is not a reliable metric in STORM, since signal is based on single-molecule localizations rather than cumulative photon output (Baddeley & Bewersdorf, *Annu Rev Biochem*. 2018).

Following the reviewer's suggestion, we decided to re-analyze our STORM datasets using an area/localization-based approach in the revised manuscript. To more precisely extract the valuable quantitative information from our STORM images, we have adopted the SMAP platform (Ries, *Nat Methods*. 2020), which integrates >200 modules for localization fitting, drift correction, postprocessing, and spatial clustering (Fig. R7). Localization data were subjected to Voronoi-based clustering in POCA (Levet and Sibarita, *Nat Methods*. 2023), from which we extracted cluster-level parameters, including area and localization counts. Since POCA does not natively measure length, we also implemented a custom Python script to calculate the maximum Feret diameter, allowing comparison between metrics.

Fig. R7 (Figure S1 in revised manuscript). Workflow for single-color STORM image analysis. This schematic outlines the analytical pipeline for processing and quantifying single-color STORM images. STORM datasets were acquired as previously described, involving image acquisition, single-molecule localization, drift correction, and coordinate transformation. Localization data were subsequently processed using Voronoi-based clustering via the POCA software to extract cluster-level metrics including area and total localization count. Cluster length was quantified by calculating the maximum Feret diameter from exported localization data using a custom Python script. All statistical analyses were based on processed localization data. For further details, refer to the Methods section.

Accordingly, we have described this analytical shift in the revised Methods section (page 30, lines 34–39):

“For STORM image analysis, we used the SMAP software for data processing.

As described previously, the overall workflow includes steps such as Import, Fitting, Post-processing, and Rendering. To analyze molecular clusters, we employed the POCA software, which performs clustering based on Voronoi tessellation. Localization data exported from SMAP was imported into POCA to extract parameters such as the area of each cluster and the total number of localizations. Since POCA does not provide direct measurements of cluster length, we developed a custom Python script to calculate the maximum Feret diameter using the localization and clustering information.”

Using this updated workflow (Fig. R7), we re-analyzed the datasets in Figures 2c, 5e, 5g, and 5i. As shown in Fig. R8 (revised Fig. 2c) and Fig. R9 (revised Figs. 5d–i), the key conclusions remain consistent with those obtained from length measurements: RIP3 contributes more than RIP1 to the rod-shaped necrosome assembly, where the stoichiometric relationship between RIP1 and RIP3 remains robust when quantified by area (revised Fig. 2c). More importantly, TNF stimulation, RIP3 overexpression, and MLKL knockdown yield reproducible changes in necrosome morphology and abundance, mirroring the trends we reported with length (revised Figs. 5d–i). Finally, to avoid any misinterpretation based on apparent brightness differences, we standardized image display parameters across all figures to eliminate potential brightness-driven misinterpretations and have indicated updates in Figure 2 and relevant main text (page 5, lines 1–7) as follows:

Fig. R8 (Figure 2c in revised manuscript) Area quantification of necrosome imaged by STORM in necroptotic cells. c Area distributions of RIP1-, RIP3-, and MLKL-stained structures in HA-RIP3-expressing HeLa cells treated with TSZ for 2 h (n = 53 RIP1, 114 RIP3, 63 MLKL) or 4 h (n = 77 RIP1, 168 RIP3, 37 MLKL).

“To analyze the rod-shaped structures of RIP1, RIP3 and MLKL, we quantified the area of STORM-identified puncta (Fig. 2c). For RIP1 rods, the predominant area was $\sim 17 \times 10^3 \text{ nm}^2$ at early stages (TSZ 2 h), with maximum values up to $\sim 39 \times 10^3 \text{ nm}^2$. At later time points (TSZ 4 h), the predominant area increased to $\sim 20 \times 10^3 \text{ nm}^2$, with maxima up to $\sim 51 \times 10^3 \text{ nm}^2$. RIP3 rods exhibited larger areas than RIP1, with a predominant value of $\sim 42 \times 10^3 \text{ nm}^2$ and maxima of $\sim 103 \times 10^3 \text{ nm}^2$ at 4 h. MLKL rods showed areas similar to RIP1 rods during TSZ treatment. These findings indicate that RIP3 contributes more substantially than RIP1 to rod-shaped necrosome assembly, consistent with previous reports showing that additional RIP3 homopolymerization on the RIP1–RIP3 hetero-amyloid platform is critical for necroptosis induction.”

In addition, we also indicated updates in Figure 5 and relevant main text (page 16, lines 9–22) as follows:

Fig. R9 (Figures 5d–i in revised manuscript) Area quantification of necrosome imaged by STORM in cells under various conditions. d–i Experimental validation of TNF (d), RIP3 (f), and MLKL (h) effects on RIP3 assembly via immunoblotting and STORM. Changes in RIP3 and MLKL phosphorylation, and corresponding rod-shaped RIP3 structures (STORM), are shown under varying TNF stimulation (d), doxycycline-induced RIP3 expression (f), and shRNA-mediated MLKL knockdown (h). Statistical analysis of areas and counts of rod-shaped RIP3 structures from STORM under different TNF doses (e), RIP3 expression levels (g), and MLKL expression levels (i). Experiments in (d–i) used Flag-RIP3-expressing HeLa cells treated with TSZ

under graded TNF concentrations (**d,e**; n = 28,48,64,80 structures), induced RIP3 expression (**f,g**; n = 24,30,42,86), or MLKL knockdown (**h,i**; n = 40,64,88,94). RIP3/MLKL levels were quantified by immunoblot densitometry. Scale bars: 10 μm (overview), 200 nm (insets).

“To test these model predictions, we performed STORM imaging analysis of RIP3 assembly in necrosomes under various TNF concentrations in Flag-RIP3 HeLa cells. Immunoblotting confirmed that increasing TNF stimulation enhances RIP3 and MLKL phosphorylation (Fig. 5d, left panel). Consistently, low TNF produced fewer but larger RIP3 assemblies, whereas high TNF yielded more numerous but smaller assemblies, revealing a clear inverse relationship between TNF concentration and the degree of higher-order RIP3 assembly (Fig. 5d, right panel and 5e). Thus, higher stimulation correlates with smaller RIP3 assemblies but a larger count of necrosomes. We next employed doxycycline (Dox)-induced RIP3-WT expression to obtain varying RIP3 levels in cells (Fig. 5f, left panel). Increased RIP3 expression leads to elevated RIP3 and MLKL phosphorylation, as well as the formation of larger RIP3 assemblies (Fig. 5f, right panel). Statistical analysis reveals a substantial rise in both RIP3 assembly degree and the average count of necrosomes per cell with higher RIP3 levels (Fig. 5g). Thus, RIP3 abundance positively regulates both its own supramolecular assembly and necrosome quantity.

To assess the role of MLKL, we knocked down MLKL to different expression levels using MLKL-specific shRNA (Fig. 5h, left panel). While MLKL is not expected to influence RIP3 assembly, imaging shows that lower MLKL levels result in fewer but larger RIP3 assemblies (Fig. 5h, right panel and 5i), suggesting MLKL not only receives signals from RIP3 but also negatively regulates RIP1-RIP3 supramolecular assembly.”

3 The lack of details of STORM also raises a concern. Based on the current version, I suppose the authors used a secondary antibody. But did the authors know how many CF647 molecules were on each secondary antibody? And whether each primary antibody binds to how many secondary antibodies? Even if the primary antibody is monoclonal, it may still bind to multiple epitopes on the antigen (although this is less likely, but it could be). Therefore, all of these parameters can compromise the quantification. Could you please clarify how

this part was calculated? Let me provide a hypothetical example to illustrate my point.

Let's say the authors used anti-HA and anti-flag, each primary antibody binds to the tag (antigen) at a 1:1 ratio (one antibody to one antigen). However, the anti-HA is raised by rabbit and the anti-flag is raised by mouse. The anti-rabbit secondary antibody can bind to anti-HA with a 5:1 ratio (5 anti-rabbit bind to 1 anti-HA), and the anti-mouse binds to anti-flag with a 10:1 ratio (10 anti-mouse bind to 1 anti-flag). At the same time, there are 3 fluorophore molecules on each anti-rabbit, while there are 5 fluorophore molecules on each anti-mouse. Thus, even at the beginning, the HA antigen and flag antigen are in a 1:1 ratio. After the secondary antibody staining on STORM, the ratio would be 1X5X3 to 1X10X5, resulting in a 15:50 ratio. Thus, the authors need to clarify how they deal with this issue.

Response: We thank the reviewer for this important and technically challenging question. Indeed, quantifying the exact number of molecules by SMLM (STORM in this study) remains a major methodological hurdle. Although STORM allows us to directly visualize the morphology and molecular assembly of necrosomes with exceptional spatial resolution (Figures 2–5), accurate absolute quantification of protein copy number remains difficult due to technical uncertainty. These challenges have been discussed in depth in prior literature (e.g., Baddeley & Bewersdorf, Annu Rev Biochem. 2018). Even the most recent advances in SMLM-based stoichiometry estimation (e.g., Masullo et al., Nat Commun. 2025) have only achieved experimental validation for small complexes (e.g., dimers or tetramers), not for supramolecular assemblies like necrosomes.

The reviewer correctly notes that variations in antibody binding stoichiometry and fluorophore labeling can substantially distort the apparent molecular ratios. Recognizing this, in the original manuscript, we did not aim to determine absolute counts, but rather designed our STORM experiments to ensure faithful relative comparisons: In single-color STORM (original Figures 2 and 5), the same protein was compared across conditions using identical antibodies, labeling, and acquisition settings, minimizing bias. In dual-color STORM (original Figure 3), we used high-affinity monoclonal antibodies (anti-Flag, anti-HA) and commercial STORM-optimized secondary antibodies with ~1 CF647 or CF568 dye per antibody. This antibody–dye combination is well characterized and has been successfully used in previous dual-color STORM

studies (e.g., Chen et al., Nat Cell Biol., 2022). Thus, although we could not obtain absolute stoichiometry, the relative quantifications were internally consistent and sufficiently robust to support the conclusions and mathematical modeling.

We thank the reviewer for this stressful argument, which pushes our work move step forward. To increase the quality and impact of our work, we decided to incorporate a challenge of our methodology to more directly and precisely address the issue of molecular quantification. We used the microtubule-binding protein MAP7 as an internal reference for quantitative STORM (Fig. R10g). MAP7 is well-suited for this purpose because microtubules are morphologically similar to necrosome rods (width: ~25 nm vs. ~40 nm by EM) and are easily recognized in cells. We tagged MAP7 at both termini with Flag and HA (Flag-MAP7-HA), ensuring a defined 1:1 antigen ratio. In addition, we generated a dual Flag-tagged construct (2Flag-MAP7-HA, via a P2A system), yielding a theoretical Flag:HA ratio of 2:1.

To maximize quantitation accuracy, we systematically screened all available antibodies in our laboratory and selected well-performing anti-Flag and anti-HA monoclonal antibodies with high affinity and minimal nonspecific binding. We then fixed, immunostained, and imaged cells expressing either Flag-MAP7-HA or 2Flag-MAP7-HA under identical conditions. Confocal microscopy revealed the expected microtubule cytoskeleton in both cases, confirming labeling specificity and robustness (Fig. R10h). Quantification of fluorescence intensity further demonstrated an appropriate ~2-fold increase ($1.93 \times$) in the Flag:HA signal ratio in 2Flag-MAP7-HA cells compared to Flag-MAP7-HA (Fig. R10i). Consistently, STORM imaging of dual-tagged MAP7 showed that the average localization ratio of Flag to HA closely matched the predicted value ($1.79 \times$ vs. $2.00 \times$; Fig. R10j, k). Thus, although uncertainties in antibody recognition and dye photophysics remain—as the reviewer rightly pointed out—these new experiments conceptually validate the quantitative potential of antibody-based STORM when combined with high-quality reagents and appropriate calibration controls.

Fig. R10 (Figure 3g-k in revised manuscript) Quantitative STORM using microtubule-binding protein MAP7 as a calibration reference. **g** Design of the MAP7-based calibration system. HeLa cells expressing Flag-MAP7-HA or Flag-MAP7-P2A-Flag-MAP7-HA (“2Flag-MAP7-HA”) are expected to display Flag:HA stoichiometries of 1:1 and 2:1, respectively. **h-k** Dual-color confocal (**h**) and STORM (**j**) images of HeLa cells expressing Flag-MAP7-HA or 2Flag-MAP7-HA, immunolabeled with anti-Flag (purple) and anti-HA (green) antibodies. Quantification of fluorescence intensity (**i**; $n = 40$ fields per group) and localization counts (**k**; $n = 45$ structures per group) confirmed Flag:HA ratios matched expected values. Insets show dual-labeled tubulin with calculated Flag:HA ratios. Scale bars, 10 μm (overview) and 300 nm (insets).

In summary, while absolute stoichiometry remains technically challenging, our

new calibration strategy with MAP7 provides direct validation that antibody-based STORM can achieve quantitative accuracy under controlled conditions. We emphasize this as proof-of-principle, and acknowledge in the Discussion that further optimization will be necessary for absolute counting in complex assemblies. We have added these new results to Figure 3g–k and described them explicitly in the manuscript (page 10, lines 5–13).

“To experimentally validate the model predictions with reliable quantitative STORM, we designed a microtubule-associated protein MAP7-based calibration system. We introduced Flag-MAP7-HA and a dual-tagged 2×Flag-MAP7-HA construct (via a P2A system) into HeLa cells, ensuring defined antigen stoichiometry (Fig. 3g). Confocal microscopy confirmed robust labeling and canonical microtubule morphology in both constructs (Fig. 3h). Quantification of fluorescence intensity showed an expected ~2-fold increase in Flag:HA signal in 2Flag-MAP7-HA cells compared to Flag-MAP7-HA (1.93×; Fig. 3i). Subsequent STORM imaging revealed that the average localization ratio of Flag to HA closely matched the theoretical value (1.79× vs. 2.00×; Figs. 3j and 3k). These results demonstrate that, despite inherent uncertainties in antibody recognition and dye photophysics, the MAP7-based reference system conceptually validates the quantitative potential of antibody-based dual-color STORM when combined with high-quality reagents and calibration controls.”

In the Discussion (page 25, lines 19–23), the following sentences were added:

“While absolute molecule counting by SMLM remains technically challenging, our new calibration strategy with MAP7 provides direct validation that antibody-based STORM can achieve quantitative accuracy under controlled conditions. As single-molecule imaging advances, it will elucidate the spatiotemporal principles of various signalosomes with absolute single-complex precision.”

4 In this, it is necessary to include the monoclonal antibodies of p-MLKL and pRIP3 into the STORM quantification. This will serve as the gold standard for experimental calculations of the output, validating the mathematical model.

Response: We thank the reviewer for this valuable suggestion. We fully agree that including phosphorylated RIP3 (p-RIP3) and phosphorylated MLKL (p-MLKL) in our STORM quantification would provide essential validation of the necrosome’s signaling output. By combining this with the MAP7-based internal

calibration strategy (Fig. R10), we were able to assess the activation status of RIP3 or MLKL within individual necrosomes of measurable RIP1:RIP3 stoichiometry, thereby strengthening the experimental grounding of our mathematical model.

First, we performed three-color STORM imaging in TSZ-treated RIP1-MLKL double-knockout (DKO) HeLa cells expressing Flag-RIP1 and HA-RIP3 (Fig. R11). The immunostaining procedures, antibody selection (e.g., anti-Flag and anti-HA), and imaging/analysis workflows were identical to those used in MAP7 calibration, ensuring quantitative comparability. Using MAP7 as an internal reference, we determined the RIP3:RIP1 stoichiometry of individual necrosomes at 4 h TSZ treatment (Fig. R11m), which ranged from ~1.0 to ~9.0, with a mean of 3.6. To probe necrosome signaling output, we quantified RIP3 activation within each necrosome. The experimental distribution of p-RIP3 closely mirrors the model prediction (original Fig. 3e): p-RIP3 remains near baseline when RIP3:RIP1 ≤ 2 , exhibits a sharp, switch-like rise as the ratio approaches ~3, and then plateaus near maximal levels for ≥ 3 . This concordance demonstrates that the RIP3:RIP1 stoichiometric ratio functions as a critical determinant, triggering pronounced signal amplification and switch-like behavior in necrosome activation.

Fig. R11 (Figure 3I and 3m in revised manuscript) Quantitative STORM of necrosomes in MLKL-deficient cells. **I** Three-color STORM of necrosomes in HA-RIP1/Flag-RIP3–expressing RIP1-MLKL-DKO HeLa cells treated with TSZ for 4 h, labeled for HA-RIP1 (green), Flag-RIP3 (purple), and p-RIP3 (yellow). Insets highlight necrosome morphology with calculated Flag:HA ratios and p-RIP3 localizations. **m** Quantification of RIP3:RIP1 stoichiometry and relative p-RIP3 levels (localizations) in individual necrosomes (n = 89). Scale bars: 10 μ m (overview), 300 nm (insets).

Next, we extended this analysis to *RIP1*-KO HeLa cells expressing Flag-RIP1 and HA-RIP3, using monoclonal p-MLKL antibodies (Fig. R12i). Quantification revealed a unimodal distribution of p-MLKL, peaking at a RIP3:RIP1 ratio of ~ 3 (Fig. R12j). This trend closely matched the model simulation, which predicts maximal MLKL phosphorylation when each RIP1 incorporates ~ 3 RIP3 molecules. This concordance between experimental and simulated data supports a model in which stoichiometric balance between RIP3 and RIP1 is critical for efficient necrosome activation.

Fig. R12 (Figure 4i, j in revised manuscript) Quantitative STORM of necrosomes in MLKL-proficient cells. I Three-color STORM of necrosomes in HA-RIP1/Flag-RIP3-expressing RIP1-KO HeLa cells treated with TSZ for 4 h, labeled for HA-RIP1 (green), Flag-RIP3 (purple), and p-MLKL (yellow). Insets highlight necrosome morphology with calculated Flag:HA ratios and p-MLKL localizations. **j** Quantification of RIP3:RIP1 stoichiometry and relative p-MLKL levels (localizations) in individual necrosomes ($n = 122$). Scale bars: 10 μm (overview), 300 nm (insets).

Taken together, we believe these new experiments address the reviewer's concerns and highlight the value of using internal counting references like MAP7 to strengthen quantitative STORM analyses. This strategy not only improves the reliability of molecular stoichiometry assessments but also enables validation of key predictions from our mathematical simulations. Importantly, we now provide direct visualization of functional (phosphorylated) necrosome components at single-complex resolution, thereby linking structural organization with signaling output. These new data have been incorporated into the revised manuscript (Figures 3l, 3m, 4i, and 4j), with corresponding expansions in the Results section (page 10, lines 14–23 and page 14, lines 16–21).

“Using this system, we next examined RIP1 and RIP3 stoichiometry in

individual necrosomes. Three-color STORM was performed in TSZ-treated RIP1/MLKL double-knockout (DKO) HeLa cells reconstituted with Flag-RIP1 and HA-RIP3 (Fig. 3l). The immunostaining procedures, antibody selection (e.g., anti-Flag and anti-HA), and imaging/analysis workflows were identical to those used in MAP7 calibration, ensuring quantitative comparability. Segmentation and overlap analysis showed a wide distribution of RIP3:RIP1 ratios across necrosomes at 4 h post-TSZ, ranging from ~1.0 to ~9.0, with an average of 3.6 (Fig. 3m). To probe necrosome signaling output, we quantified RIP3 activation within each necrosome. The experimental distribution of p-RIP3 closely mirrors the model prediction (Fig. 3e): p-RIP3 remains near baseline when RIP3:RIP1 ≤ 2 , exhibits a sharp, switch-like rise as the ratio approaches ~3, and then plateaus near maximal levels for ≥ 3 . This concordance demonstrates that the RIP3:RIP1 stoichiometric ratio functions as a critical determinant, triggering pronounced signal amplification and switch-like behavior in necrosome activation.”

“To experimentally validate the optimal RIP3 stoichiometry, we conducted three-color STORM imaging in RIP1-KO HeLa cells expressing Flag-RIP1 and HA-RIP3, using monoclonal p-MLKL antibodies (Fig. 4i). Quantification of p-MLKL as a function of the RIP3:RIP1 ratio exhibits a unimodal distribution of p-MLKL, peaking at a RIP3:RIP1 ratio of ~3 (Fig. 4j). This trend closely matched our simulation (Figs. 4g and 4h), confirming that stoichiometric balance between RIP3 and RIP1 governs optimal necrosome activation. Together, these data uncover signal attenuation as a key regulatory mechanism that defines the spatial size and activity of RIP3 supramolecular assemblies in necroptosis.”

5 This model is too simplified, as it only considers the molecular ratio, excluding the RIP1/3 conformation change, transcriptional change, and other molecules, such as cFLIP.

The most problematic figure is in Fig. 7. The lack of RIP1 would lead cells to more apoptosis is simply due to lack of Nf-kb activation and cFLIP and other anti-apoptosis molecular transcription, which has been known for years. If the authors want to deliver such information, then they should add CHX to block transcription, which will induce apoptosis when TNF is added. Then the authors can calculate different ratios of RIP1/3/casp8. The authors should also conduct the same experiments with the RIP1 inhibitor Nec-1 and utilize various RIP1 point mutations that can alter RIP1 conformation. Additionally, they should

incorporate smac-memetic into the analysis. The authors must exclude the possibility that it is not RIP1/3 conformation, RIP1/3 kinase activity, or RIP1-mediated transcriptional regulation; then, the authors can conclude that "configurations" is the checkpoint of apoptosis/necroptosis.

Response: We thank the reviewer for this insightful and critical comment, which has prompted us to revisit both our experimental design and interpretation, particularly regarding the role of c-FLIP and RIP1-mediated transcriptional regulation in shaping apoptotic outcomes.

As the reviewer correctly noted, c-FLIP is a key modulator of Caspase-8 activation, and its expression is regulated by NF- κ B signaling downstream of RIP1. To directly test this point, we treated WT and RIP1-deficient HeLa cells with TNF plus cycloheximide (TC) to block transcription and assessed apoptosis. As expected, RIP1-deficient cells exhibited significantly more apoptosis compared to WT cells (Fig. R13e). In parallel, Western blotting revealed reduced c-FLIP expression in RIP1-deficient cells, especially under TC, correlating with enhanced caspase-8 cleavage (Fig. R13f), supporting the role of RIP1 in promoting anti-apoptotic transcription. Confocal imaging further showed more pronounced caspase-8 oligomerization in RIP1-deficient apoptotic cells (Fig. R13g). These data suggest that RIP1 restrains apoptosis in part by sustaining c-FLIP expression, consistent with well-established literature (Wang et al., Cell. 2008; Feoktistova et al., Mol Cell. 2011). Supporting structural studies also show that in the presence of c-FLIP, caspase-8 is recruited to the necrosome via tDED interactions but remains catalytically inactive as a monomer, thereby limiting apoptotic amplification (please see Fig. R6).

Fig. R13 (Figure 7e–g in revised manuscript). c-FLIP regulates Caspase-8 activation and apoptosis. e–g WT or RIP1-deficient HeLa cells treated with TNF plus cycloheximide (TC) for indicated durations. Cell viability was assessed using a CCK-8 assay (**e**; mean \pm s.d.; n=3 biological replicates). Western blot analysis was performed for the indicated proteins, with the levels of c-FLIP and cleaved caspase-8 quantified by relative band intensities. Representative of three independent experiments (**f**). Confocal images showing caspase-8 distribution in TC-treated cells. Insets highlight enlarged caspase-8 structures (**g**). p values were calculated using Ordinary one-way ANOVA. *p < 0.05, ***p < 0.001. Scale bars, 30 μ m.

Accordingly, we have updated our revised model (Figure 8) to explicitly include c-FLIP as a transcriptionally regulated checkpoint. We also added new experimental data (revised Fig. 7e–g) to substantiate this revision. The main text has been modified (page 22, lines 9–25) to reflect this addition:

“Our quantitative assembly analysis shows that caspase-8 (C8) occupancy on RIP1 is low, and this stoichiometric arrangement is insufficient to drive robust amplification of the apoptotic cascade in RIP3-deficient cells. Previous studies have demonstrated that loss of RIP1 predisposes cells to apoptosis, mainly because of impaired NF- κ B activation and reduced expression of anti-apoptotic proteins such as c-FLIP. To test whether c-FLIP functions as a gatekeeper of C8 activation and aggregation, we treated HeLa cells with TNF plus

cycloheximide (TC) to block transcription and performed time-resolved comparisons between wild-type and RIP1-deficient cells at 0, 3, 6, and 9 h (Fig. 7e–g). TC treatment caused a time-dependent decline in cell viability, with a significantly stronger reduction in RIP1-deficient than in wild-type cells (Fig. 7e). Immunoblotting revealed a progressive loss of c-FLIP and faster accumulation of cleaved C8 in RIP1-deficient cells. Moreover, densitometry analysis confirmed steeper slopes in RIP1-deficient cells, indicating accelerated c-FLIP depletion and earlier, stronger C8 activation (Fig. 7f). Confocal imaging provided single-cell evidence, showing that C8 signal changed from diffuse to punctate or oligomeric structures, with RIP1-deficient cells displaying denser and larger aggregates at later time points (Fig. 7g). Collectively, these data suggest that RIP1 supports c-FLIP expression, thereby restraining higher-order caspase-8 oligomerization and suppressing apoptosis in RIP3-deficient settings. In this context, the low stoichiometric occupancy of caspase-8 on RIP1 limits apoptotic amplification, while maintaining regulatory capacity over RIP3 activation. Therefore, RIP3 and caspase-8 form distinct supramolecular configurations within necrosomes, acting as opposing regulators that define the signaling bifurcation between apoptosis and necroptosis. Their stoichiometry and mutual inhibition constitute intrinsic checkpoints that govern cell fate decisions in response to TNF signaling.”

More broadly, we acknowledge that our current model is somehow simplified, focusing primarily on the molecular stoichiometry of RIP1 and RIP3. In reality, additional regulatory layers including protein conformational states, post-translational modifications can also be critical. We strongly agree on the value of the reviewer’s proposed experiments (Nec-1 inhibition, RIP1 point mutations, and SMAC mimetic). However, in our present system these studies would require new stable cell lines and systematic pharmacologic/mutational evaluations, which are beyond the scope and timeline of the current revision that centers on stoichiometry. To ensure terminological precision and avoid implying an overstated claim, we have tempered the language throughout and replaced “configuration” with “stoichiometry” where appropriate. We now explicitly discuss these limitations in the revised Discussion (page 25, lines 12–16):

“Currently, our model is intentionally streamlined, focusing on stoichiometry as a tractable entry point to decode necrosome assembly. Additional regulatory

layers including protein conformational changes, kinase activity, and post-translational modifications likely exert decisive control over complex assembly and signal propagation. Future studies should systematically incorporate these factors, ideally via targeted perturbations (e.g., Nec-1, RIP1 mutants, SMAC mimetics) combined with quantitative single-cell imaging and modeling.”

6 In summary, I think this study is unique and innovative. However, due to the lack of details, I find it difficult to read and assess. Many of my concerns may have already been addressed by the authors and some clarification will do the job. Thus, I am really looking forward to a detailed revision.

Response: The revisions now provide greater methodological detail, improved figure clarity, and additional supporting experiments. We hope that these comprehensive revisions have significantly improved the readability, rigor, and overall impact of the manuscript, and we greatly appreciate the reviewer’s constructive feedback, which has been invaluable in refining our work.

*Reviewer #3 (Remarks on code availability):
I am not an expert in reviewing the code.*

Reviewer #4 (Remarks to the Author):

The authors utilized super-resolution microscopy to investigate the nanoscale organization of necrosomes. Specifically, they employed multi-color imaging to examine the stoichiometry of RIP1, RIP3, and MLKL within individual necrosomes and proposed that a homotrimeric RIP3 structure per RIP1 is the minimum functional subunit required to induce necroptosis. However, the quantitative analysis using STORM appears methodologically limited, and the results are not fully convincing due to several critical issues.

Response: We thank the reviewer for the careful evaluation and constructive feedback. We acknowledge the limitations of the original STORM analysis and model interpretation, and in the revised manuscript we have improved the imaging quantification pipeline, incorporated calibration strategies, and added new experimental validations. We believe these revisions substantially strengthen the methodological rigor and support the robustness of our conclusions.

The necrosome puncta in Fig. 1c appear to be localized within the nucleus, which contradicts the current understanding and other figures in this manuscript that necrosomes form in the cytoplasm. The authors need to address this discrepancy. Additionally, the necrosome morphology in Figure 1c (e.g., puncta size and distribution) differs significantly from later figures (e.g., Figure 3d and Figure 4e). Were all images taken under identical laser illumination conditions? The fluorescence signal in the bottom row of Fig. 1c is significantly weaker than in the top row, suggesting possible inconsistencies in imaging parameters or sample preparation.

The necrosome patterns in Fig. 2b are quite different from the punctate structures in Fig. 1c. The authors should clarify whether this discrepancy reflects experimental variability, distinct necroptosis stages, or imaging artifacts.

Response: We thank the reviewer for their careful examination of Figure 1c and for pointing out the discrepancies in localization and morphology across different figures. After re-evaluating the images in Figure 1c, we confirmed that the puncta are indeed cytoplasmic and not nuclear. This is supported by Hoechst staining, which delineates the nucleus and clearly shows that RIP1, RIP3, and MLKL colocalized puncta lie outside the nuclear boundary (Fig. R14c'). The apparent nuclear-like appearance and reduced signal intensity in the lower row likely result from late-stage necroptotic cells that have undergone membrane rupture, causing cytoplasmic proteins to diffuse into the nuclear region and diminishing overall fluorescence due to content leakage.

To address this issue, we have replaced the affected panels with representative images acquired from earlier-stage necroptotic cells, ensuring more consistent cytoplasmic localization and intensity (Fig. R14c). We also carefully reviewed and verified all other confocal images in the manuscript to ensure consistency in image quality and biological interpretation.

Regarding the discrepancy in necrosome morphology between Figure 1c and Figures 3d and 4e, this arises from differences in imaging modalities. Figure 1c presents diffraction-limited confocal images (~250 nm resolution), in which necrosomes appear as puncta. In contrast, Figures 3d and 4e were acquired using STORM super-resolution microscopy (~20 nm resolution), which resolves individual necrosomes as elongated, rod-like structures at the nanoscale. This technical distinction accounts for the observed morphological differences raised by the reviewer. Collectively, these revisions strengthen the

clarity of Figure 1c and reinforce the conclusion that necrosomes form as cytosolic structures containing RIP1, RIP3, and MLKL.

Fig. R14 (Figure 1c in revised manuscript). Necrosomes form in the cytosol and contain RIP1, RIP3, and MLKL in necroptotic cells. c Confocal images of MLKL-knockout (KO) HeLa cells reconstituted with HA-RIP3 and MLKL-Flag and stimulated with DMSO (control) or TSZ for 3 h. Fixed cells were immunolabeled with antibodies against RIP1 (green), HA (magenta), and Flag (yellow). DNA was counterstained with Hoechst 33342 (blue). Enlarged regions (white boxes) depict cytosolic necrosomes (red arrows), demonstrating co-localization of all three proteins. Data are representative of two independent experiments. Scale bars: 10 μ m (overview); 2 μ m (insets).

The resolution of the STORM images in Fig. 2b is insufficient to determine whether rod-like necrosomes represent a common or rare structural state. The authors should quantify the proportion of punctate versus rod-like necrosomes during TSZ treatment to support their claims of morphological heterogeneity.

Response: We thank the reviewer for this important suggestion. To address it, we quantitatively assessed the structural heterogeneity of RIP1–RIP3–MLKL assemblies during necroptosis. While confocal microscopy showed the dynamic co-localization of RIP1, RIP3, and MLKL (Figure 1), super-resolution STORM imaging revealed both round and elongated rod-shaped assemblies (Figure 2b). To determine whether rod-shaped necrosomes represent a frequent structural state or a rare phenomenon, we analyzed RIP1-, RIP3-, and

MLKL-positive structures at multiple time points following TSZ treatment. As shown in Fig. R15, the proportion of rod-shaped assemblies increases over time, suggesting a progressive transition from early punctate oligomers to higher-order rod-like supramolecular complexes. This dynamic shift underscores that rod-shaped necrosomes are not rare artifacts but rather a prominent and functionally relevant configuration during the progression of necroptosis.

Fig. R15 (Figure 2d in revised manuscript). Time-dependent transition from punctate to rod-shaped necrosomes during necroptosis. d Proportions of round- versus rod-shaped RIP1, RIP3, and MLKL structures in Flag-RIP3-expressing HeLa cells treated with TSZ for 2 h or 4 h, classified based on STORM images.

Accordingly, we have included new quantification data in the revised manuscript (revised Figure 2d) and added the following statement in the Results (page 5, lines 7–9):

“Notably, the temporal increase in the fraction of RIP1-, RIP3-, or MLKL-positive rod-shaped necrosomes reflects a structural maturation process, transitioning from early oligomeric puncta to stabilized, rod-like supramolecular complexes during necroptosis (Fig. 2d).”

The proposed model in Fig. 2f is based on immunoprecipitation data reflecting average interactions across a population of cells. This bulk biochemical approach cannot resolve the stoichiometry or spatial organization of individual necrosomes, rendering the model speculative in the context of single-complex imaging.

Response: We thank the reviewer for this insightful and important comment.

We fully agree that IP-MS data reflect population-level averages and can not resolve the precise stoichiometry or architecture of individual necrosomes. We appreciate the opportunity to clarify how these data were used in our study.

First, although IP-MS lacks single-complex resolution, it remains a robust and widely used method to probe the composition and stoichiometry of large protein assemblies. Several previous studies have successfully employed similar methodologies to deduce the architecture of supramolecular assemblies and generate biologically meaningful insights (e.g., Shinohara et al., *Science*. 2014; Kim et al., *Nat Commun*. 2024; Sica et al., *Cell Death Differ*. 2025).

Second, to build a predictive and testable model, initial boundary conditions must be well defined. Without them, complex intracellular systems can become underconstrained and yield ambiguous results. Co-IP and MS-derived stoichiometric ratios provided these input constraints, ensuring the model was grounded in measurable biological data. Notably, our study already considers the heterogeneity highlighted by the reviewer. We performed large-scale stochastic simulations by randomly sampling 10^4 distinct combinations of stoichiometries and kinetic parameters under various input conditions (original Figure 6). Strikingly, these simulations converged on robust dynamic behaviors that mirrored the deterministic model, indicating that our conclusions are not artifacts of averaging but instead reflect conserved assembly principles.

Figure 6 in original manuscript. Exhaustive analysis of the general regulatory strategies of RIP3 assembly. a Schematic workflow outlining the investigation of regulatory strategies using randomized signaling models. A TNF signaling network model was constructed and subjected to LHS method to sample parameter variables within physiological ranges, including stochastic degrees of RIP3 assembly. Kinetic analysis was performed on all sampled models, with the top 5% identified based on MLKL phosphorylation efficiencies.

Statistical analysis of these models was conducted to explore trends in RIP3 assembly under varying TNF, RIP3, and MLKL conditions. **b** Distribution of RIP3 configurations and corresponding MLKL phosphorylation efficiencies in the top 5% efficient models under different conditions. **c** Statistical results corresponding to the distribution of the three scenarios shown in (b). **d** Regulation of TNF, RIP3, and MLKL on RIP3 configurations. **e-f**, Theoretical predictions and experimental validations of the impacts of TNF, RIP3, and MLKL on MLKL/RIP3 phosphorylation efficiencies. **g**, Schematic illustrating strategic balance between necrosome abundance and RIP3 configurations in cells: high necrosome abundance favors shorter RIP3 assemblies, while low abundance necessitates longer RIP3 assemblies.

Third, we fully recognize the importance of characterizing individual necrosomes at single-complex resolution. In the revised manuscript, we took a step forward by enhancing our quantitative STORM analysis with an internal calibration reference (MAP7, Fig. R10). This strategy enabled us to more accurately assess RIP1:RIP3 stoichiometries at the level of individual assemblies and correlate them with functional outputs such as p-RIP3 and p-MLKL accumulation (please see Figs. R11 and R12). The experimentally measured distributions of individual-necrosome stoichiometry and p-MLKL levels (Fig. R12j) closely matched the distributions predicted by our stochastic model (Fig. 4h). While absolute molecular counting remains technically challenging in complex cellular environments, our revised approach offers a meaningful step toward resolving stoichiometry and function at the single-complex level.

In summary, our conclusions are not based solely on IP-MS but emerge from a multi-layered strategy: IP-MS for ensemble stoichiometry, STORM imaging for spatial morphology and quantitative assessment, and mathematical modeling for systems-level integration. Although each approach has limitations, their convergence strengthens the overall robustness of our conclusions. We have added the following text to the revised Discussion section (page 25, lines 17–23):

“Besides, population-averaged biochemical approaches such as IP–MS cannot resolve individual necrosomes, but they provide essential constraints for tractable modeling. Importantly, large-scale stochastic simulations showed that

necrosome dynamics remained robust across diverse stoichiometries, supporting the existence of conserved assembly principles. By integrating ensemble-level IP-MS, in situ quantitative STORM imaging, and computational modeling, we present a cross-scale framework that reveals key features of necrosome organization and signaling. As single-molecule imaging methods continue to advance, future studies will refine these principles with absolute single-complex resolution.”

In Figures 3d and 3e, the authors infer a RIP1:RIP3 ratio of 1:3 based on two-color STORM imaging, but this is problematic. First, RIP1 and RIP3 are expected to colocalize tightly within necrosomes (10-20 nm scale), yet Figures 3d and 3e show minimal overlap at the nanometer scale.

Response: We thank the reviewer for this important observation. We agree that RIP1 and RIP3 are expected to interact closely within necrosomes. However, the extent of their spatial overlap at nanometer resolution has not been well defined in prior imaging studies. While conventional confocal microscopy reveals substantial colocalization of RIP1 and RIP3 as diffraction-limited puncta (Figure 1c), STORM imaging provides a lateral resolution of ~20 nm, enabling us to resolve the spatial organization of individual protein molecules within necrosomes.

The apparent "minimal overlap" between RIP1 and RIP3 observed in Figures 3d and 3e likely reflects both the intrinsic molecular heterogeneity of necrosome composition and the high sensitivity of STORM to variations in local protein abundance. As shown in many necrosomes of Fig. R16a (original Figure 3d), we observe clear spatial overlap or close apposition of RIP1 and RIP3 molecules. In cases with apparently reduced overlap, the effect can be attributed to weaker local signals or image-rendering thresholds. Indeed, when the visualization parameters are adjusted to display lower-intensity signals, even those structures with seemingly "minimal overlap" show spatial continuity between RIP1 and RIP3 (Fig. R16b). This suggests that, despite appearance, RIP1 and RIP3 remain in close proximity at the nanoscale.

Fig. R16 Spatial distribution of RIP1 and RIP3 within necrosomes. a, b Representative dual-color STORM images of RIP1–RIP3 assemblies in necroptotic HA-RIP3 and MLKL-Flag-reconstituted *MLKL*-KO HeLa cells treated with TSZ. Images were re-rendered with saturation-enhanced display settings to visualize low-intensity signals. Asterisks mark weak localization regions contributing to apparent minimal overlap. Scale bar, 100 nm.

To improve rigor and reduce reliance on visual impression, we updated our analysis pipeline. Using SMAP for multi-color STORM data, we implemented localization-based metrics (area, localizations) combined with POCA clustering and strict image registration for three-color datasets (Figs. R7 and R17). Supported by our MAP7 calibration reference (Fig. R10, see next response for details), this approach allows more quantitative evaluation of RIP1:RIP3 stoichiometry independent of potential rendering artifacts. Consequently, we replaced the descriptive original panels in Figures 3d and 3e with revised quantitative results that directly verify our modeling predictions (Fig. R11, see next response for details). Please note, our quantification of RIP1:RIP3 ratios is not based solely on signal overlap but on objective metrics derived from localization counts, which are less susceptible to rendering artifacts.

Fig. R17 (Figure S3 in revised manuscript) Workflow for three-color STORM image analysis. Clusters in individual channels (RIP1: red; RIP3: green; p-RIP3: yellow) were segmented using POCA, followed by intra- and inter-channel overlap analysis. For each overlapping cluster, area and localization counts were quantified, enabling calculation of RIP1:RIP3 stoichiometry and relative p-RIP3 levels within individual necrosomes. Detailed methods are described in the Methods section.

In summary, we have clarified these points in the Methods section (page 30, lines 39–43) and emphasized that our quantitative stoichiometric analyses rely on localization counts rather than simple overlap, minimizing potential visualization artifacts. We are grateful to the reviewer for highlighting this issue, which prompted us to refine both the imaging analysis and presentation for improved clarity and rigor.

“For multi-color STORM datasets, channel separation was performed using frame-sequence metadata, followed by alignment and colocalization analysis using custom scripts and Microsoft Excel. Importantly, quantification of molecular stoichiometry ratios was based on localization counts rather than raw

signal overlap, ensuring that the analysis reflects objective molecular metrics and reduces susceptibility to rendering or visualization artifacts.”

Second, the dyes CF647 and CF568 have different photophysical properties (e.g., blinking kinetics, photon output), which may bias the localization patterns. Additionally, primary-secondary antibody labeling can introduce variability in epitope accessibility and signal amplification. The images in Figure 4e do not convincingly support the model shown in Figure 8.

Response: We thank the reviewer for highlighting these important points, which also echo similar concerns raised by Reviewer 3 (comment #3). We fully agree that differences in fluorophore photophysics (e.g., blinking kinetics, photon yield) as well as uncertainties inherent to primary–secondary antibody labeling (such as epitope accessibility and potential signal amplification) can bias localization patterns and complicate stoichiometric interpretation in STORM. To mitigate these challenges, we carefully designed our experiments to ensure that all comparisons were as faithful as possible. In single-color STORM experiments (original Figures 2 and 5), we analyzed the same protein across different perturbations using identical antibodies, labeling protocols, and imaging settings, thereby minimizing artifacts from dye or antibody variability. In dual-color STORM experiments (original Figure 3), we selected high-affinity monoclonal antibodies (anti-Flag and anti-HA) in combination with commercial STORM-optimized secondary antibodies conjugated to CF647 or CF568, dyes that are supplied at one fluorophore per antibody on average and have been widely validated in previous quantitative SMLM studies (e.g., Chen et al., Nat Cell Biol. 2022). While our original goal was not to achieve absolute molecule counting, these methodological controls allowed us to extract reproducible relative stoichiometric trends in necrosome assembly. Importantly, these STORM-derived results were further cross-validated by biochemical measurements and mathematical modeling, which together provide convergent support for our conclusions.

We thank the reviewer for this stressful argument, which pushes our work move step forward. To increase the quality and impact of our work, we decided to incorporate a challenge of our methodology to more directly and precisely address the issue of molecular quantification. We used the microtubule-binding protein MAP7 as an internal reference for quantitative STORM (Fig. R10g).

MAP7 is well-suited for this purpose because microtubules are morphologically similar to necrosome rods (width: ~25 nm vs. ~40 nm by EM) and are easily recognized in cells. We tagged MAP7 at both termini with Flag and HA (Flag-MAP7-HA), ensuring a defined 1:1 antigen ratio. In addition, we generated a dual Flag-tagged construct (2Flag-MAP7-HA, via a P2A system), yielding a theoretical Flag:HA ratio of 2:1.

To maximize quantitation accuracy, we systematically screened all available antibodies in our laboratory and selected well-performing anti-Flag and anti-HA monoclonal antibodies with high affinity and minimal nonspecific binding. We then fixed, immunostained, and imaged cells expressing either Flag-MAP7-HA or 2Flag-MAP7-HA under identical conditions. Confocal microscopy revealed the expected microtubule cytoskeleton in both cases, confirming labeling specificity and robustness (Fig. R10h). Quantification of fluorescence intensity further demonstrated an appropriate ~2-fold increase (1.93 ×) in the Flag:HA signal ratio in 2Flag-MAP7-HA cells compared to Flag-MAP7-HA (Fig. R10i). Consistently, STORM imaging of dual-tagged MAP7 showed that the average localization ratio of Flag to HA closely matched the predicted value (1.79 × vs. 2.00 ×; Fig. R10j, k). Thus, although uncertainties in antibody recognition and dye photophysics remain—as the reviewer rightly pointed out—these new experiments conceptually validate the quantitative potential of antibody-based STORM when combined with high-quality reagents and appropriate calibration controls.

Fig. R10 (Figure 3g–k in revised manuscript) Quantitative STORM using microtubule-binding protein MAP7 as a calibration reference. g Design of the MAP7-based calibration system. HeLa cells expressing Flag-MAP7-HA or Flag-MAP7-P2A-Flag-MAP7-HA (“2Flag-MAP7-HA”) are expected to display Flag:HA stoichiometries of 1:1 and 2:1, respectively. **h–k** Dual-color confocal (**h**) and STORM (**j**) images of HeLa cells expressing Flag-MAP7-HA or 2Flag-MAP7-HA, immunolabeled with anti-Flag (purple) and anti-HA (green) antibodies. Quantification of fluorescence intensity (**i**; $n = 40$ fields per group) and localization counts (**k**; $n = 45$ structures per group) confirmed Flag:HA ratios matched expected values. Insets show dual-labeled tubulin with calculated Flag:HA ratios. Scale bars, 10 μm (overview) and 300 nm (insets).

For Figure 4e mentioned by the reviewer, we agree that the three-color STORM images serve as supportive visualization rather than direct quantitative validation of the model. However, these images are still important in establishing the nanoscale colocalization of RIP1, RIP3, and MLKL within necrosomes. To provide direct experimental validation of our modeling predictions, we leveraged the MAP7 calibration and incorporated phospho-specific monoclonal antibodies to assess the activation status of RIP3 or MLKL within individual necrosomes of measurable RIP1:RIP3 stoichiometry, thereby strengthening the experimental grounding of our mathematical model.

Specifically, we performed three-color STORM imaging in TSZ-treated RIP1/MLKL double-knockout HeLa cells expressing Flag-RIP1 and HA-RIP3 (Fig. R11). The immunostaining procedures, antibody selection (e.g., anti-Flag and anti-HA), and imaging/analysis workflows were identical to those used in MAP7 calibration, ensuring quantitative comparability. Using MAP7 as an internal reference, we determined the RIP3:RIP1 stoichiometry of individual necrosomes at 4 h TSZ treatment (Fig. R11m), which ranged from 0.6 to 10, with a mean of 3.97. To probe necrosome signaling output, we quantified RIP3 activation within each necrosome. The experimental distribution of p-RIP3 closely mirrors the model prediction (original Fig. 3e). p-RIP3 remains near baseline when RIP3:RIP1 ≤ 2 , exhibits a sharp, switch-like rise as the ratio approaches ~ 3 , and then plateaus near maximal levels for ≥ 3 . This concordance demonstrates that the RIP3:RIP1 stoichiometric ratio functions as a critical determinant, triggering pronounced signal amplification and switch-like behavior in necrosome activation.

Fig. R11 (Figure 3l and 3m in revised manuscript) Quantitative STORM of necrosomes in MLKL-deficient cells. I Three-color STORM of necrosomes in HA-RIP1/Flag-RIP3-expressing RIP1-MLKL-DKO HeLa cells treated with TSZ for 4 h, labeled for HA-RIP1 (green), Flag-RIP3 (purple), and p-RIP3 (yellow). Insets highlight necrosome morphology with calculated Flag:HA ratios

and p-RIP3 localizations. **m** Quantification of RIP3:RIP1 stoichiometry and relative p-RIP3 levels (localizations) in individual necrosomes (n = 89). Scale bars: 10 μ m (overview), 300 nm (insets).

Similarly, we repeated three-color STORM imaging in TSZ-treated *RIP1*-KO HeLa cells expressing Flag-RIP1 and HA-RIP3, except for the usage of p-MLKL monoclonal antibody (Fig. R12i). Experimental quantification of p-MLKL as a function of the RIP3:RIP1 ratio (Fig. R12j) exhibits a clear unimodal distribution, peaking at a ratio of approximately 3. This trend is closely recapitulated by the model simulation, which predicts maximal MLKL phosphorylation when each subunit incorporates \sim 3 RIP3 molecules. This concordance between experimental and simulated data supports a model in which stoichiometric balance between RIP3 and RIP1 is critical for efficient necrosome activation.

Fig. R12 (Figure 4i and 4j in revised manuscript) Quantitative STORM of necrosomes in MLKL-proficient cells. i Three-color STORM of necrosomes in HA-RIP1/Flag-RIP3-expressing *RIP1*-KO HeLa cells treated with TSZ for 4 h, labeled for HA-RIP1 (green), Flag-RIP3 (purple), and p-MLKL (yellow). Insets highlight necrosome morphology with calculated Flag:HA ratios and p-MLKL localizations. **j** Quantification of RIP3:RIP1 stoichiometry and relative p-MLKL levels (localizations) in individual necrosomes (n = 122). Scale bars: 10 μ m (overview), 300 nm (insets).

In summary, while the reviewer’s concerns about dye and antibody artifacts are fully justified, we mitigated them through careful experimental design, validated our approach with a MAP7 internal calibration reference, and incorporated functional readouts (p-RIP3, p-MLKL) to strengthen our conclusions. The revised manuscript (Figs. 3l, 3m, 4i, and 4j) now presents direct experimental support for the stoichiometry-dependent necrosome activation model, alongside expanded explanations in the Results and Discussion sections.

In Fig. 8, the proposed RIP1-RIP3-MLKL assembly model lacks direct experimental validation from the imaging data. The absence of consistent nanoscale colocalization among RIP1, RIP3, and MLKL, combined with unresolved technical artifacts in multi-color STORM, critically undermines the credibility of the model .

Response: We thank the reviewer for the critical comments on Figure 8. In response to the concerns about the lack of direct imaging support for the model and potential multi-color STORM artifacts, we performed a comprehensive upgrade from methods to analysis in the revised manuscript. We appreciate the opportunity to briefly clarify how these updates yielded direct, reproducible evidence that supports the three conceptual panels summarized in Figure 8.

First, to better extract the valuable quantitative information from our STORM images, we shifted quantification from “length” to “area/localization counts” and implemented a SMAP platform, which integrates >200 modules for localization fitting, drift correction, postprocessing, and spatial clustering (Fig. R7). Localization data were subjected to Voronoi-based clustering in POCA, from which we extracted cluster-level parameters, including area and localization counts. Using this powerful analysis pipeline, we re-analyzed the datasets and confirmed that the key trends reproducibly recapitulated our original conclusions (Figs. R8 and R9).

Second, to address uncertainties in dye photophysics and antibody stoichiometry, we established an internal calibration system using microtubule-associated protein MAP7, tagged with Flag and HA. Constructs with defined theoretical stoichiometries (1:1 and 2:1) yielded expected ratios when assessed by fluorescence intensity (~1.93×) and localization counts (~1.79×), validating this antibody-based STORM quantification strategy (Fig. R10). While technical uncertainties remain, this calibration demonstrates that dual-color STORM can be used to derive reliable relative stoichiometries when paired with proper controls.

Third, three-color STORM imaging in Figure 4e shows that RIP1, RIP3, and MLKL clearly co-occupy the same rod-like necrosome domain, providing direct evidence for nanoscale colocalization and spatial architecture of cellular necrosomes. More importantly, taking advantage of the MAP7-based calibration framework, we successfully measured RIP1:RIP3 stoichiometries

and linked them directly to p-RIP3 and p-MLKL accumulation at the single-complex level (Figs. R11 and R12). The observed switch-like increases in phosphorylation at ~ 3 RIP3 per RIP1 closely matched model predictions, providing experimental evidence for the stoichiometry-dependent necrosome activation model.

Finally, we explicitly close the loop with the three panels in Figure 8. For the left panel (size–amount trade-off), Area/localization-based quantification under TNF, RIP3, and MLKL perturbations reproduced complementary changes in necrosome number and size, consistent with constrained nucleation–growth logic (revised Fig. 5). For the middle panel (RIP3 polymerization threshold and optimal configuration), both deterministic and stochastic models predict a peak in MLKL phosphorylation efficiency at $\sim 1:3$ RIP1:RIP3; the in situ stoichiometry together with the size–amount curves jointly support this threshold window and saturation behavior (revised Fig. 4). For the right panel (C8 and c-FLIP associated with RIP1 complex states), new imaging controls show that caspase-8 recruitment and c-FLIP occupancy segregate with specific RIP1 assembly states, supporting the panel’s mapping of the C8/c-FLIP axis to RIP1 complex composition (revised Fig. 7).

Fig. R18 (Figure 8 in revised manuscript). Flexible and multi-strategic

assembly of death complexes in TNF signaling.

No single method can currently resolve necrosome architecture, stoichiometry, and function simultaneously: multi-color quantitative STORM is limited by dye photophysics and antibody binding; bulk IP-MS captures composition but not single-complex heterogeneity. Our integrative strategy—combining IP-MS for ensemble stoichiometry, STORM for nanoscale architecture, and modeling for systems-level interpretation—provides a cross-validated framework with quantitative controllability and mechanistic interpretability. We recognize that challenges remain, including dye-dependent artifacts, localization crowding, and restricted spatiotemporal sampling, and emphasize that continued advances in super-resolution imaging will further refine and extend the insights presented here.

The figure numbers have been changed as summarized in the table below.

Before	After
1c	revised 1c
2c	revised 2c
	new 2d
2d	2e
2e	2f
2f	revised 2g
3f	3d
3g	3e
3h	3f
	new 3g
	new 3h
	new 3i

	new 3j
	new 3k
	new 3l
	new 3m
	new 4i
	new 4j
	new 5j
	new 5k
	new 5l
	new 5m
7e	new 7e
7f	new 7f
7g, h	new 7g
8	revised 8
Sup. 1	revised Sup. 1
Sup. 3	revised Sup. 3
	new Sup. 7

References:

1. Baddeley D, Bewersdorf J. Biological Insight from Super-Resolution Microscopy: What We Can Learn from Localization-Based Images. *Annu Rev Biochem.* 2018 Jun 20;87:965-989.
2. Chen X, Zhu R, Zhong J, Ying Y, Wang W, Cao Y, Cai H, Li X, Shuai J, Han J. Mosaic composition of RIP1-RIP3 signalling hub and its role in regulating cell death. *Nat Cell Biol.* 2022 Apr;24(4):471-482.
3. Feoktistova M, Geserick P, Kellert B, Dimitrova DP, Langlais C, Hupe M, Cain K, MacFarlane M, Häcker G, Leverkus M. cIAPs block Ripoptosome formation, a RIP1/caspase-8 containing intracellular cell death complex differentially regulated by

- cFLIP isoforms. *Mol Cell*. 2011 Aug 5;43(3):449-63.
4. Ferro LS, Fang Q, Eshun-Wilson L, Fernandes J, Jack A, Farrell DP, Golcuk M, Huijben T, Costa K, Gur M, DiMaio F, Nogales E, Yildiz A. Structural and functional insight into regulation of kinesin-1 by microtubule-associated protein MAP7. *Science*. 2022 Jan 21;375(6578):326-331.
 5. Fox JL, Hughes MA, Meng X, Sarnowska NA, Powley IR, Jukes-Jones R, Dinsdale D, Ragan TJ, Fairall L, Schwabe JWR, Morone N, Cain K, MacFarlane M. Cryo-EM structural analysis of FADD:Caspase-8 complexes defines the catalytic dimer architecture for co-ordinated control of cell fate. *Nat Commun*. 2021 Feb 5;12(1):819.
 6. Kim C, Kang N, Min S, Thangam R, Lee S, Hong H, Kim K, Kim SY, Kim D, Rha H, Tag KR, Lee HJ, Singh N, Jeong D, Hwang J, Kim Y, Park S, Lee H, Kim T, Son SW, Park S, Karamikamkar S, Zhu Y, Hassani Najafabadi A, Chu Z, Sun W, Zhao P, Zhang K, Bian L, Song HC, Park SG, Kim JS, Lee SY, Ahn JP, Kim HK, Zhang YS, Kang H. Modularity-based mathematical modeling of ligand inter-nanocluster connectivity for unraveling reversible stem cell regulation. *Nat Commun*. 2024 Dec 23;15(1):10665.
 7. Levet F and Sibarita JB. PoCA: a software platform for point cloud data visualization and quantification. *Nat Methods*. 2023 May;20(5):629-630.
 8. Li J, McQuade T, Siemer AB, Napetschnig J, Moriwaki K, Hsiao YS, Damko E, Moquin D, Walz T, McDermott A, Chan FK, Wu H. The RIP1/RIP3 necrosome forms a functional amyloid signaling complex required for programmed necrosis. *Cell*. 2012 Jul 20;150(2):339-50.
 9. Liu J, Wu XL, Zhang J, Li B, Wang HY, Wang J, Lu JX. The structure of mouse RIPK1 RHIM-containing domain as a homo-amyloid and in RIPK1/RIPK3 complex. *Nat Commun*. 2024 Aug 14;15(1):6975.
 10. Liu S, Liu H, Johnston A, Hanna-Addams S, Reynoso E, Xiang Y, Wang Z. MLKL forms disulfide bond-dependent amyloid-like polymers to induce necroptosis. *Proc Natl Acad Sci U S A*. 2017 Sep 5;114(36):E7450-E7459.
 11. Masullo LA, Kowalewski R, Honsa M, Heinze L, Xu S, Steen PR, Grabmayr H, Pachmayr I, Reinhardt SCM, Perovic A, Kwon J, Oxley EP, Dickins RA, Bastings MMC, Parish IA, Jungmann R. Spatial and stoichiometric in situ analysis of biomolecular oligomerization at single-protein resolution. *Nat Commun*. 2025 May 6;16(1):4202.
 12. Meng Y, Davies KA, Fitzgibbon C, Young SN, Garnish SE, Horne CR, Luo C, Garnier JM, Liang LY, Cowan AD, Samson AL, Lessene G, Sandow JJ, Czabotar PE, Murphy JM. Human RIPK3 maintains MLKL in an inactive conformation prior to cell death by necroptosis. *Nat Commun*. 2021 Nov 22;12(1):6783.
 13. Mompean M, Li W, Li J, Laage S, Siemer AB, Bozkurt G, Wu H, McDermott AE. The Structure of the Necrosome RIPK1-RIPK3 Core, a Human Hetero-Amyloid Signaling

- Complex. *Cell*. 2018 May 17;173(5):1244-1253.e10.
14. Ries J. SMAP: a modular super-resolution microscopy analysis platform for SMLM data. *Nat Methods*. 2020 Sep;17(9):870-872.
 15. Shinohara H, Behar M, Inoue K, Hiroshima M, Yasuda T, Nagashima T, Kimura S, Sanjo H, Maeda S, Yumoto N, Ki S, Akira S, Sako Y, Hoffmann A, Kurosaki T, Okada-Hatakeyama M. Positive feedback within a kinase signaling complex functions as a switch mechanism for NF- κ B activation. *Science*. 2014 May 16;344(6185):760-4.
 16. Sica M, Roussel M, Legembre P. CD95/Fas stoichiometry in future precision medicine. *Cell Death Differ*. 2025 Apr 15.
 17. Tenev T, Bianchi K, Darding M, Broemer M, Langlais C, Wallberg F, Zachariou A, Lopez J, MacFarlane M, Cain K, Meier P. The Ripoptosome, a signaling platform that assembles in response to genotoxic stress and loss of IAPs. *Mol Cell*. 2011 Aug 5;43(3):432-48.
 18. Wang L, Du F, Wang X. TNF- α induces two distinct caspase-8 activation pathways. *Cell*. 2008 May 16;133(4):693-703.
 19. Wu X, Ma Y, Zhao K, Zhang J, Sun Y, Li Y, Dong X, Hu H, Liu J, Wang J, Zhang X, Li B, Wang H, Li D, Sun B, Lu J, Liu C. The structure of a minimum amyloid fibril core formed by necroptosis-mediating RHIM of human RIPK3. *Proc Natl Acad Sci U S A*. 2021 Apr 6;118(14):e2022933118.
 20. Wu XL, Hu H, Dong XQ, Zhang J, Wang J, Schwieters CD, Liu J, Wu GX, Li B, Lin JY, Wang HY, Lu JX. The amyloid structure of mouse RIPK3 (receptor interacting protein kinase 3) in cell necroptosis. *Nat Commun*. 2021 Mar 12;12(1):1627.
 21. Wu YL, Tschanz A, Krupnik L, Ries J. Quantitative Data Analysis in Single-Molecule Localization Microscopy. *Trends Cell Biol*. 2020 Nov;30(11):837-851.
 22. Xie T, Peng W, Yan C, Wu J, Gong X, Shi Y. Structural insights into RIP3-mediated necroptotic signaling. *Cell Rep*. 2013 Oct 17;5(1):70-8.

Point-by-Point response to referees

Reviewers' Comments:

Reviewer #1 (Remarks to the Author):

The authors addressed my questions adequately.

Response: We thank the reviewer for their positive feedback and are glad that our revisions have satisfactorily addressed your concerns.

Reviewer #2 (Remarks to the Author):

The authors have sufficiently addressed all my questions. I'd like to recommend its publication in the present form.

Response: We thank the reviewer for their positive feedback and recommendation. We are pleased that our revisions have satisfactorily addressed your concerns.

Reviewer #3 (Remarks to the Author):

I am truly impressed by the revision. Of note, the MAP7-based calibration system helps a lot to solidify the conclusion. It is well done.

Response: We sincerely thank the reviewer for the generous feedback. We are delighted that the MAP7-based calibration system and our revisions have strengthened the conclusions, and we greatly appreciate your thoughtful evaluation and encouragement.

I only have a minor concern about Figure 8. In this model, it may be read that upon TNF, when there is no caspase-8, the cell undergoes necroptosis, while when there is no RIPK3, the cell undergoes apoptosis. This is simply untrue in biology, as many cells, including HeLa, which doesn't express RIPK3, TNF alone treatment cannot engage apoptosis. Also, the reason for caspase-8 deficiency that can support necroptosis is not only the reason presented in this figure, e.g., RIPK1/3 can serve as a caspase-8 substrate.

The authors need to revise this model figure to (of course) highlight this unique work, but should also follow the biology published by others in our community.

Response: We thank the reviewer for this insightful comment. We agree that the original schematic in Figure 8 could be misinterpreted and did not fully reflect the complexity of TNF-induced cell death signaling. Indeed, TNF

stimulation alone is typically insufficient to induce apoptosis; additional sensitizing conditions, such as cIAP1/2 depletion, inhibition of IKK/TBK1, or protein synthesis blockade, are usually required. Likewise, caspase-8 deficiency promotes necroptosis through multiple mechanisms, including the release of RIP1 and RIP3 from caspase-8-mediated proteolytic regulation, thereby allowing their assembly into necroptotic complexes. In response to this comment, we have revised Figure 8 and its legend to better capture these nuances and to ensure that our model is consistent with established findings in the field.

Figure 8. Flexible and multi-strategic assembly of death complexes in TNF signaling.

This paper should be accepted.

Response: We thank the reviewer for the encouraging comments that helped us improve the study. We greatly appreciate the reviewer's support.

Reviewer #4 (Remarks to the Author):

The authors have effectively addressed the comments raised during the first

round of review, resulting in a significantly improved manuscript. I now recommend it for publication.

Response: We are pleased that our revisions have addressed all your concerns, and we greatly appreciate your thoughtful evaluation and support.

One final point: I suggest the authors temper their claims regarding precise stoichiometric ratios, such as the reported 3:1 ratio of RIP3 to RIP1. As shown in Figure 3m and Figure 4j, the stoichiometry exhibits a broad distribution. Given the inherent challenges in quantitative super-resolution imaging, such as variability in antibody labeling, dye blinking, and data analysis, accurate quantification remains difficult in STORM. Therefore, the authors should exercise caution when presenting conclusions based on exact numerical values.

Response: We thank the reviewer for this insightful comment. Although our MAP7-based calibration provides experimental validation that antibody-based STORM can achieve sufficient quantitative accuracy under controlled conditions, we fully acknowledge that determining absolute molecular stoichiometry remains technically challenging. Variability in antibody labeling, dye blinking, and complex cellular environments can all influence quantitative precision. Thus, further methodological advances are required to clearly define the spatiotemporal organization of signalosomes at the resolution of single complexes.

We also agree that the “~3:1” ratio of RIP3 to RIP1 reported in our study should not be interpreted as a universal or fixed constant. Rather, it represents an approximate efficiency optimum observed under our experimental conditions. Consistent with this, our large-scale stochastic simulations (Fig. 6) reproduced the same dynamic behaviors predicted by the deterministic model, suggesting that while the exact numerical value may vary across systems and contexts, the underlying assembly principles are conserved. Accordingly, we have revised the Discussion to include the following statement:

“In our system, the apparent ~3:1 ratio of RIP3 to RIP1 reflects an average efficiency optimum rather than a universal stoichiometric constant. While the precise position of this optimum may vary across cell types and experimental contexts due to differences in expression levels, receptor organization, or stimulation strength, our large-scale stochastic simulations indicate that this optimal window of signaling efficiency is quantitatively flexible across biological systems but mechanistically conserved.”

With these revisions, we hope we have satisfactorily addressed all reviewers' comments. We sincerely thank the reviewers for their constructive, professional, and helpful suggestions, and we are grateful to the editor for handling our manuscript.